# THE PHASE TRANSITION PHENOMENON OF SHUFFLED REGRESSION

## ABSTRACT

We study the phase transition phenomenon inherent in the shuffled (permuted) regression problem, which has found numerous applications in databases, privacy, data analysis, etc. For the permuted regression task: $\mathbf{Y} = \mathbf{\Pi}^{\natural}\mathbf{X}\mathbf{B}^{\natural}$, the goal is to recover the permutation matrix $\mathbf{\Pi}^{\natural}$ as well as the coefficient matrix $\mathbf{B}^{\natural}$. It has been empirically observed in prior studies that when recovering $\mathbf{\Pi}^{\natural}$, there exists a phase transition phenomenon: the error rate drops to zero rapidly once the parameters reach certain thresholds. In this study, we aim to precisely identify the locations of the phase transition points by leveraging techniques from *message passing* (MP).

In our analysis, we first transform the permutation recovery problem into a probabilistic graphical model. Then, we leverage the analytical tools rooted in the message passing (MP) algorithm and derive an equation to track the convergence of the MP algorithm. By linking this equation to the branching random walk process, we are able to characterize the impact of the *signal-to-noise-ratio* (snr) on the permutation recovery. Depending on whether the signal is given or not, we separately investigate the oracle case and the non-oracle case. The bottleneck in identifying the phase transition regimes lies in deriving closed-form formulas for the corresponding critical points, but only in rare scenarios can one obtain such precise expressions. To tackle this challenge, we propose the Gaussian approximation method, which allows us to obtain the closed-form formulas in almost all scenarios. In the oracle case, our method can fairly accurately predict the phase transition snr. In the non-oracle case, our proposed algorithm can predict the maximum allowed number of permuted rows and uncover its dependency on the sample number.

Numerical experiments reveal that the observed phase transition points are well aligned with our theoretical predictions. Our study will motivate exploiting MP algorithms (and related techniques) as an effective tool for permuted regression problems, which have found applications in machine learning, privacy, and databases.

## 1 INTRODUCTION

In this paper, we consider the following permuted (shuffled) linear regression problem:

$$\mathbf{Y} = \mathbf{\Pi}^{\natural}\mathbf{X}\mathbf{B}^{\natural} + \sigma\mathbf{W}, \tag{1}$$

where $\mathbf{Y} \in \mathbb{R}^{n \times m}$ denotes the matrix of observations, $\mathbf{\Pi}^{\natural} \in \{0, 1\}^{n \times n}$ is the permutation matrix, $\mathbf{X} \in \mathbb{R}^{n \times p}$ is the design matrix, $\mathbf{B}^{\natural} \in \mathbb{R}^{p \times m}$ is the matrix of signals (regressors), $\mathbf{W} \in \mathbb{R}^{n \times m}$ denotes the additive noise matrix (with unit variance), and $\sigma^2$ is the noise variance. The task is to recover both the signal matrix $\mathbf{B}^{\natural}$ and the permutation matrix $\mathbf{\Pi}^{\natural}$. The research on this challenging permuted regression problem dates back at least to 1970s under the name "broken sample problem" (DeGroot et al., 1971; Goel, 1975; DeGroot & Goel, 1976; 1980; Bai & Hsing, 2005). Recent years have witnessed a revival of this problem due to its broad spectrum of applications in (e.g.,) privacy protection, data integration, etc. (Unnikrishnan et al., 2015; Pananjady et al., 2018; Slawski & Ben-David, 2019; Pananjady et al., 2017; Slawski et al., 2020; Zhang & Li, 2020).

Specifically, this paper will focus on studying the "phase transition" phenomenon in recovering the whole permutation matrix $\mathbf{\Pi}^{\natural}$: the error rate for the permutation recovery sharply drops to zero once the parameters reach certain thresholds. In particular, we leverage techniques in the *message passing* (MP) algorithm literature to identify the precise positions of the phase transition thresholds. The

bottleneck in identifying the phase transition regimes lies in deriving closed-form formulas for the corresponding critical points. This is a highly challenging task because only in rare scenarios can one obtain such precise expressions. To tackle the difficulty, we propose the Gaussian approximation method which allows us to obtain the closed-form formula in almost all scenarios. We should mention that, in previous studies (Slawski et al., 2020; Slawski & Ben-David, 2019; Pananjady et al., 2017; Zhang et al., 2022; Zhang & Li, 2020), this phase transition phenomenon was empirically observed.

**Related work.** The problem we study simultaneously touches two distinct areas of research: (A) permutation recovery, and (B) *message passing* (MP). In the literature of permuted linear regression, essentially all existing works used the same setting (1). Pananjady et al. (2018); Slawski & Ben-David (2019) consider the single observation model (i.e., $m = 1$) and prove that the *signal-to-noise-ratio* (snr) for the correct permutation recovery is $\mathbb{O}_{\mathrm{P}}(n^c)$, where $c > 0$ is some positive constant. Slawski et al. (2020); Zhang & Li (2020); Zhang et al. (2022) investigate the multiple observations model (i.e., $m > 1$) and suggest that the snr requirement can be significantly decreased, from $\mathbb{O}_{\mathrm{P}}(n^c)$ to $\mathbb{O}_{\mathrm{P}}(n^{c/m})$. In particular, Zhang & Li (2020) develop an estimator which we will leverage and analyze for studying the phase transition phenomenon. Compared with the above work, our analysis can identify the precise locations of the phase transition thresholds. In this February, there comes a paper (Lufkin et al., 2024) considering the same problem as ours but in a much simpler setting (single measurement with $m = 1$). Compared with their work, our framework can easily reproduce their predicted phase transition points, answer the questions they treat as open, and predict the phase transition points in a unified framework. A detailed discussion can be found in the main context.

Another line of related research comes from the field of statistical physics. For example, using the replica method, Mézard & Parisi (1985; 1986) study the *linear assignment problem* (LAP), i.e., $\min_{\mathbf{\Pi}} \sum_{i,j} \mathbf{\Pi}_{ij} \mathbf{E}_{ij}$ where $\mathbf{\Pi}$ denotes a permutation matrix and $\mathbf{E}_{ij}$ is i.i.d random variable uniformly distributed in $[0, 1]$. Martin et al. (2005) then generalize LAP to multi-index matching and presented an investigation based on MP algorithm. Recently, Caracciolo et al. (2017); Malatesta et al. (2019) extend the distribution of $\mathbf{E}_{ij}$ to a broader class. However, all the above works exhibit no phase transition. Chertkov et al. (2010) extend it to the particle tracking problem and observe a phase transition phenomenon. Later, Semerjian et al. (2020) modify it to fit the graph matching problem, which paves way for our work in studying the permuted linear regression problem.

**Our contributions.** We propose the first framework to identify the precise locations of phase transition thresholds associated with permuted linear regression. In the oracle case where $\mathbf{B}^{\natural}$ is known, our scheme is able to determine the phase transition snr. In the non-oracle case where $\mathbf{B}^{\natural}$ is not given, our method will predict the maximum allowed number of permuted rows and uncover its dependence on the ratio $p/n$. In our analysis, we identify the precise positions of the phase transition points in the large-system limit, e.g., $n, m, p$ all approach to infinity with $m/n \to \tau_m$, $p/n \to \tau_p$. Interestingly, numerical results well match predictions even when $n, m, p$ are in the hundreds.

Here, we would also like to briefly mention the technical challenges. Compared with the previous works (Mezard & Montanari, 2009; Talagrand, 2010; Linusson & Wästlund, 2004; Mézard & Parisi, 1987; 1986; Parisi & Ratiéville, 2002; Semerjian et al., 2020), where the edge weights are relatively simple, our edge weights usually involve high-order interactions across Gaussian random variables and are densely correlated. To tackle this issue, our proposed approximation method to compute the phase transition thresholds consists of three parts: 1) performing Gaussian approximation; 2) modifying the leave-one-out technique; and 3) performing size correction. A detailed explanation can be found in Section 4. Hopefully, our approximation method will serve independent technical interests for researchers in the machine learning community.

**Notations.** In this paper, $a \xrightarrow{\text{a.s.}} b$ denotes $a$ converges almost surely to $b$. We denote $f(n) \simeq g(n)$ when $\lim_{n \to \infty} f(n)/g(n) = 1$, and $f(n) = \mathbb{O}_{\mathrm{P}}(g(n))$ if the sequence $f(n)/g(n)$ is bounded in probability, and $f(n) = o_{\mathrm{P}}(g(n))$ if $f(n)/g(n)$ converges to zero in probability. The inner product between two vectors (resp. matrices) are denoted as $\langle \cdot, \cdot \rangle$. For two distributions $d_1$ and $d_2$, we write $d_1 \cong d_2$ if they are equal up to normalization. Moreover, $\mathcal{P}_n$ denotes the set of all possible permutation matrices: $\mathcal{P}_n \triangleq \{\mathbf{\Pi} \in \{0, 1\}^{n \times n}, \sum_i \mathbf{\Pi}_{ij} = 1, \sum_j \mathbf{\Pi}_{ij} = 1\}$. The *signal-to-noise-ratio* is $\mathrm{snr} = \frac{\|\mathbf{B}^{\natural}\|_{\mathrm{F}}^2}{m \cdot \sigma^2}$, where $\|\cdot\|_{\mathrm{F}}$ is the Frobenius norm and $\sigma^2$ is the variance of the sensing noise.

## 2    PERMUTATION RECOVERY USING THE MESSAGE PASSING ALGORITHM

Inspired by Mezard & Montanari (2009); Chertkov et al. (2010); Semerjian et al. (2020), we leverage tools from the statistical physics to identify the locations of the phase transition threshold. We start this section with a brief review of the *linear assignment problem* (LAP), which reads as

$$\widehat{\mathbf{\Pi}} = \ \mathrm{argmin}_{\mathbf{\Pi} \in \mathcal{P}_n} \ \langle \mathbf{\Pi}, \mathbf{E} \rangle , \tag{2}$$

where $\mathbf{E} \in \mathbb{R}^{n \times n}$ is a fixed matrix and $\mathcal{P}_n$ denotes the set of all possible permutation matrices.

In our work, we first establish the link between the LAP and the permuted linear recovery, to be more specific, formulating the permutation recovery of (1) in the form of (2). Next, we predict the phase transition points by studying the matrix $\mathbf{E}$, which is our major contribution.

We follow the approach in Mezard & Montanari (2009); Semerjian et al. (2020) and introduce a probability measure over the permutation matrix $\mathbf{\Pi}$, which is written as

$$\mu(\mathbf{\Pi}) = (1/z) \prod_i \mathbb{1}\big(1 - \sum_j \mathbf{\Pi}_{ij}\big) \prod_j \mathbb{1}\big(1 - \sum_i \mathbf{\Pi}_{ij}\big) \times \exp\Big( - \beta \sum_{i,j} \mathbf{\Pi}_{ij} \mathbf{E}_{ij}\Big), \tag{3}$$

where $\mathbb{1}(\cdot)$ is the indicator function, $Z$ is the normalization constant of the probability measure $\mu(\mathbf{\Pi})$, and $\beta > 0$ is an auxiliary parameter. It is easy to verify the following two properties, e.g., 1) ML estimator in (2) can be rewritten as $\widehat{\mathbf{\Pi}} = \mathrm{argmax}_{\mathbf{\Pi}}\mu(\mathbf{\Pi})$[1]; and 2) the probability measure $\mu(\mathbf{\Pi})$ concentrates on $\widehat{\mathbf{\Pi}}$ when letting $\beta \to \infty$.

Then, we study the impact of $\{\mathbf{E}_{ij}\}$ on the reconstructed permutation $\widehat{\mathbf{\Pi}}$ with the *message passing* (MP) algorithm. First, we associate a probabilistic graphical model with the probability measure defined in (3). Then, we rewrite the solution in (2) in the language of the MP algorithm. Finally, we derive an equation (7) to track the convergence of the MP algorithm. By exploiting relation of (7) to the *branching random walk* (BRW) process, we can identify the phase transition points corresponding to the LAP in (2).

### 2.1    CONSTRUCTION OF THE GRAPHICAL MODEL

First, we construct the factor graph associated with the probability measure in (3). Adopting the same strategy as in Chapter 16 of Mezard & Montanari (2009), we conduct the following operations, e.g., 1) associating each variable $\mathbf{\Pi}_{ij}$ a variable node $v_{ij}$; 2) associating the variable node $v_{ij}$ a function node representing the term $e^{-\beta \mathbf{\Pi}_{ij} \mathbf{E}_{ij}}$; and 3) linking each constraint $\sum_i \mathbf{\Pi}_{ij} = 1$ to a function node and similarly for the constraint $\sum_j \mathbf{\Pi}_{ij} = 1$. A graphical representation is available in Figure 1.

Now we briefly review the MP algorithm. Informally speaking, MP is a local algorithm to compute the marginal probabilities over the graphical model. In each iteration, the variable node $v$ transmits the message to its incident function node $f$ by multiplying all incoming messages except the message along the edge $(v, f)$. The function node $f$ transmits the message to its incident variable node $v$ by computing the weighted summary of all incoming messages except the message along the edge $(f, v)$. For a detailed introduction to MP, we refer readers to Kschischang et al. (2001), Chapter 16 in MacKay et al. (2003), and Chapter 14 in Mezard & Montanari (2009).

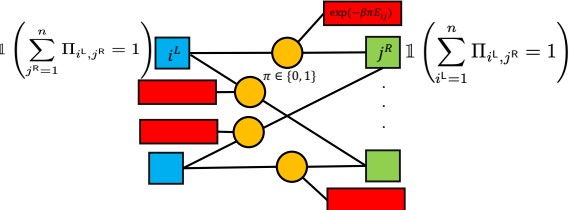

Figure 1: The constructed graphical model. Circle icons denote the variable nodes and square icons denote the function nodes: blue squares (green squares resp.) for the constraints on the rows (columns resp.) of $\mathbf{\Pi}$, and red squares for the function $e^{-\beta\pi \mathbf{E}_{ij}}$.

It is known that MP can obtain the exact marginals (Mezard & Montanari, 2009) for singly connected graphical models. For other types of graphs, however, whether MP can obtain the exact solution still remains an open problem (Cantwell & Newman, 2019; Kirkley et al., 2021). At the same time,

---

[1]Notice that the requirement $\mathbf{\Pi} \in \mathcal{P}_n$ is incorporated in $\mu(\mathbf{\Pi})$ implicitly and thus we do not need an explicit constraint.

numerical evidences have been witnessed to show that MP can yield meaningful results for graphs with loops; particular examples include applications in the coding theory (Chung, 2000; Richardson & Urbanke, 2001; 2008) and the LAP (which happens to be our case) (Mezard & Montanari, 2009; Chertkov et al., 2010; Caracciolo et al., 2017; Malatesta et al., 2019; Semerjian et al., 2020).

## 2.2 THE MESSAGE PASSING (MP) ALGORITHM

Next, we perform permutation recovery via MP. The following derivation follows the standard procedure, which can be found in the previous works (Mezard & Montanari, 2009; Semerjian et al., 2020). We denote the message flow from the node $i^{\mathsf{L}}$ to the variable node $(i^{\mathsf{L}}, j^{\mathsf{R}})$ as $\widehat{m}_{i^{\mathsf{L}} \to (i^{\mathsf{L}}, j^{\mathsf{R}})}(\cdot)$ and that from the edge $(i^{\mathsf{L}}, j^{\mathsf{R}})$ to node $i^{\mathsf{L}}$ as $m_{(i^{\mathsf{L}}, j^{\mathsf{R}}) \to i^{\mathsf{L}}}(\cdot)$. Similarly, we define $\widehat{m}_{j^{\mathsf{R}} \to (i^{\mathsf{L}}, j^{\mathsf{R}})}(\cdot)$ and $m_{(i^{\mathsf{L}}, j^{\mathsf{R}}) \to j^{\mathsf{R}}}(\cdot)$ as the message flow transmitted between the functional node $j^{\mathsf{R}}$ and the variable node $(i^{\mathsf{L}}, j^{\mathsf{R}})$. Here the superscripts L and R are used to indicate the positions of the node (left and right, respectively). Roughly speaking, these transmitted messages can be viewed as (unnormalized) conditional probability $\mathbb{P}(\Pi_{i,j} = \{0, 1\} | (\cdot))$ with the joint PDF being defined in (3). The message transmission process is to iteratively compute these conditional probabilities.

First, we consider the message flows transmitted between the functional node $i^{\mathsf{L}}$ and the variable node $(i^{\mathsf{L}}, j^{\mathsf{R}})$, which are written as

$$m_{(i^{\mathsf{L}}, j^{\mathsf{R}}) \to i^{\mathsf{L}}}(\pi) \cong \widehat{m}_{j^{\mathsf{R}} \to (i^{\mathsf{L}}, j^{\mathsf{R}})}(\pi) e^{-\beta \pi \mathbf{E}_{i^{\mathsf{L}}, j^{\mathsf{R}}}},$$

$$\widehat{m}_{i^{\mathsf{L}} \to (i^{\mathsf{L}}, j^{\mathsf{R}})}(\pi) \cong \sum_{\pi_{i^{\mathsf{L}}, k^{\mathsf{R}}}} \prod_{k^{\mathsf{R}} \neq j^{\mathsf{R}}} \widehat{m}_{k^{\mathsf{R}} \to (i^{\mathsf{L}}, k^{\mathsf{R}})}(\pi_{i^{\mathsf{L}}, k^{\mathsf{R}}}) \times e^{-\beta \pi_{i^{\mathsf{L}}, k^{\mathsf{R}}} \mathbf{E}_{i^{\mathsf{L}}, k^{\mathsf{R}}}} \mathbb{1}(\pi + \sum_{k} \pi_{i^{\mathsf{L}}, k^{\mathsf{R}}} = 1), \quad (4)$$

where $\pi \in \{0, 1\}$ is a binary value. Similarly, we can write the message flows between the functional node $j^{\mathsf{R}}$ and the variable node $(i^{\mathsf{L}}, j^{\mathsf{R}})$, which are denoted as $m_{(i^{\mathsf{L}}, j^{\mathsf{R}}) \to j^{\mathsf{R}}}(\pi)$ and $\widehat{m}_{j^{\mathsf{R}} \to (i^{\mathsf{L}}, j^{\mathsf{R}})}(\pi)$, respectively. With the parametrization approach, we define

$$h_{i^{\mathsf{L}} \to (i^{\mathsf{L}}, j^{\mathsf{R}})} \triangleq \frac{1}{\beta} \log \frac{\widehat{m}_{i^{\mathsf{L}} \to (i^{\mathsf{L}}, j^{\mathsf{R}})}(1)}{\widehat{m}_{i^{\mathsf{L}} \to (i^{\mathsf{L}}, j^{\mathsf{R}})}(0)}, \quad h_{j^{\mathsf{R}} \to (i^{\mathsf{L}}, j^{\mathsf{R}})} \triangleq \frac{1}{\beta} \log \frac{\widehat{m}_{j^{\mathsf{R}} \to (i^{\mathsf{L}}, j^{\mathsf{R}})}(1)}{\widehat{m}_{j^{\mathsf{R}} \to (i^{\mathsf{L}}, j^{\mathsf{R}})}(0)}.$$

Following the routine derivations in MP, we get the edge selection criteria, i.e., we pick $\widehat{\pi}(i^{\mathsf{L}}) = j^{\mathsf{R}}$ if

$$h_{i^{\mathsf{L}} \to (i^{\mathsf{L}}, j^{\mathsf{R}})} + h_{j^{\mathsf{R}} \to (i^{\mathsf{L}}, j^{\mathsf{R}})} > \mathbf{E}_{i^{\mathsf{L}}, j^{\mathsf{R}}}; \tag{5}$$

otherwise, we have $\widehat{\pi}(i^{\mathsf{L}}) \neq j^{\mathsf{R}}$. Due to the fact that $\mu(\mathbf{\Pi})$ concentrates on $\widehat{\mathbf{\Pi}}$ when $\beta$ is sufficiently large, we can thus rewrite the MP update equation as

$$h_{i^{\mathsf{L}} \to (i^{\mathsf{L}}, j^{\mathsf{R}})} = \min_{k^{\mathsf{R}} \neq j^{\mathsf{R}}} \mathbf{E}_{i^{\mathsf{L}}, k^{\mathsf{R}}} - h_{k^{\mathsf{R}} \to (i^{\mathsf{L}}, k^{\mathsf{R}})}, \quad h_{j^{\mathsf{R}} \to (i^{\mathsf{L}}, j^{\mathsf{R}})} = \min_{k^{\mathsf{L}} \neq i^{\mathsf{L}}} \mathbf{E}_{k^{\mathsf{L}}, j^{\mathsf{R}}} - h_{k^{\mathsf{L}} \to (k^{\mathsf{L}}, j^{\mathsf{R}})}, \tag{6}$$

which is attained by letting $\beta \to \infty$.

## 2.3 IDENTIFICATION OF THE PHASE TRANSITION THRESHOLD

To identify the phase transition phenomenon inherent in the MP update equation (6), we follow the strategy in Semerjian et al. (2020) and divide all edges $(i^{\mathsf{L}}, j^{\mathsf{R}})$ into two categories according to whether the edge $(i^{\mathsf{L}}, j^{\mathsf{R}})$ corresponds to the ground-truth permutation matrix $\mathbf{\Pi}^{\natural}$ or not. Within each category, we assume the edges' weights and the message flows along them can be represented by independently identically distributed random variables.

For the edge $(i^{\mathsf{L}}, \pi^{\natural}(i^{\mathsf{L}}))$ for the ground-truth correspondence, we represent the random variable associated with the weight $\mathbf{E}_{ij}$ as $\Omega$. The random variable for the message flow along this edge is denoted $H$ (for both $h_{i^{\mathsf{L}} \to (i^{\mathsf{L}}, j^{\mathsf{R}})}$ and $h_{j^{\mathsf{R}} \to (i^{\mathsf{L}}, j^{\mathsf{R}})}$). For the rest of edges $(i^{\mathsf{L}}, j^{\mathsf{R}})$ $(j^{\mathsf{R}} \neq \pi^{\natural}(i^{\mathsf{L}}))$, we define the corresponding random variables for the edge weight and message flow as $\widehat{\Omega}$ and $\widehat{H}$, respectively. Then, we can rewrite (6) as

$$\widehat{H}^{(t+1)} = \min\left(\Omega - H^{(t)}, H^{'(t)}\right), \quad H^{(t+1)} = \min_{1 \leq i \leq n-1} \widehat{\Omega}_i - \widehat{H}_i^{(t)}, \tag{7}$$

where $(\cdot)^{(t)}$ denotes the update in the $t$-th iteration, $H^{'}$ is an independent copy of $H$, $\{H_i^{(t)}\}_{1 \leq i \leq n-1}$ and $\{\widehat{\Omega}_i\}_{1 \leq i \leq n-1}$ denote the i.i.d. copies of random variables $H_{(\cdot)}^{(t)}$ and $\widehat{\Omega}_{(\cdot)}$. This equation (7) can

be viewed as the analogous version of the *density evolution* and *state evolution*, which are used to analyze the convergence of the message passing and approximate message passing algorithm, respectively (Chung, 2000; Richardson & Urbanke, 2001; 2008; Donoho et al., 2009; Maleki, 2010; Bayati & Montanari, 2011; Rangan, 2011).

**Remark 1.** *We conjecture that the distribution difference in the edges' weights is a necessary component in capturing the phase transition. On one hand, according to Mézard & Parisi (1986; 1987); Parisi & Ratiéville (2002); Linusson & Wästlund (2004); Mezard & Montanari (2009); Talagrand (2010), there is no phase transition phenomenon in LAP if the edges' weights, i.e., $\mathbf{E}_{ij}$, are assumed to be i.i.d uniformly distributed in $[0, 1]$. On the other hand, Semerjian et al. (2020) show a phase transition phenomenon when assuming the weights $\mathbf{E}_{ij}$ follow different distributions among the edges associated with the ground-truth correspondence $(i^{\mathsf{L}}, \pi^\natural(i^{\mathsf{L}}))$ and the rest edges.*

**Relation to *branching random walk* (BRW) process.** Conditional on the event that the permutation can be perfectly reconstructed, i.e., $H + H' > \Omega$ as in (5), we can simplify (7) as

$$H^{(t+1)} = \min_{1 \leq i \leq n-1} H_i^{(t)} + \Xi_i, \tag{8}$$

where $\Xi$ is defined as the difference between $\widehat{\Omega}$ and $\Omega$, which is written as $\Xi \triangleq \widehat{\Omega} - \Omega$, and $\{H_i^{(t)}\}_{1 \leq i \leq n-1}$ and $\{\Xi_i\}_{1 \leq i \leq n-1}$ denote the i.i.d. copies of random variables $H_{(\cdot)}^{(t)}$ and $\Xi_{(\cdot)}$.

**Remark 2.** *We briefly explain why the phase transition points predicted by (7) correspond to the full permutation recovery (i.e., evaluating the performance in terms of $\mathbb{P}(\widehat{\mathbf{\Pi}} \neq \mathbf{\Pi}^\natural)$). This is because that we regard message flows $h_{i^{\mathsf{L}} \to (i^{\mathsf{L}}, j^{\mathsf{R}})}$ and $h_{j^{\mathsf{R}} \to (i^{\mathsf{L}}, j^{\mathsf{R}})}$ i.i.d. samples from certain distributions (represented by the random variable $H$) in the derivation. In other words, we track the behaviors of all message flows when studying the evolution behavior of the random variable $H$.*

*Hence, we can claim that all correspondence between all pairs is correct if we find the correct recovery can be obtained for an arbitrary sample $H$. On the other hand, we can expect some pairs with wrong correspondence if $H$ leads to incorrect recovery. [2] Thus, we can predict the phase transition points based on the convergence of (7).*

Adopting the same viewpoint of Semerjian et al. (2020), we treat (8) as a *branching random walk* (BRW) process, which enjoys the following property.

**Theorem 1** (Hammersley (1974); Kingman (1975); Semerjian et al. (2020)). *Consider the recursive distributional equation $K^{(t+1)} = \min_{1 \leq i \leq n} K_i^{(t)} + \Xi_i$, where $K_i^{(t)}$ and $\Xi_i$ are i.i.d copies of random variables $K_{(\cdot)}^{(t)}$ and $\Xi_{(\cdot)}$, we have $\frac{K^{(t+1)}}{t} \xrightarrow{\text{a.s.}} -\inf_{\theta > 0} \frac{1}{\theta} \log \left[ \sum_{i=1}^{n} \mathbb{E} e^{-\theta \Xi_i} \right]$, conditional on the event that $\lim_{t \to \infty} K^{(t)} \neq \infty$.*

With Theorem 1, we can compute phase transition point for the correct (full) permutation recovery, i.e., $H + H' > \Omega$, by letting $\inf_{\theta > 0} \frac{1}{\theta} \log \left[ \sum_{i=1}^{n} \mathbb{E} e^{-\theta \Xi_i} \right] = 0$, since otherwise the condition in (5) will be violated (see a detailed explanation in Appendix). In practice, directly computing the infimum of $\inf_{\theta > 0} \frac{1}{\theta} \log \left[ \sum_{i=1}^{n} \mathbb{E} e^{-\theta \Xi_i} \right]$ is only possible for limited scenarios. In the next section, we propose an approximate computation method for the phase transition points, which is capable of covering a broader class of scenarios.

## 3 Analysis of the Phase Transition Points

Recall that, in this paper, we consider the following linear regression problem with permuted labels

$$\mathbf{Y} = \mathbf{\Pi}^\natural \mathbf{X} \mathbf{B}^\natural + \sigma \mathbf{W},$$

where $\mathbf{Y} \in \mathbb{R}^{n \times m}$ represents the matrix of observations, $\mathbf{\Pi}^\natural \in \mathcal{P}_n$ denotes the permutation matrix to be reconstructed, $\mathbf{X} \in \mathbb{R}^{n \times p}$ is the sensing matrix with each entry $\mathbf{X}_{ij}$ following the i.i.d standard normal distribution, $\mathbf{B}^\natural \in \mathbb{R}^{p \times m}$ is the matrix of signals, and $\mathbf{W} \in \mathbb{R}^{n \times m}$ represents the additive noise matrix and its entries $\mathbf{W}_{ij}$ are i.i.d standard normal random variables. In addition, we denote $h$ as the number of permuted rows corresponding to the permutation matrix $\mathbf{\Pi}^\natural$.

---

[2]It's noteworthy that the fact $H$ leads to incorrect recovery does not mean the reconstructed correspondences are simultaneously incorrect. Numerical experiments also confirm this claim.

In this work, we focus on studying the "phase transition" phenomenon in recovering $\mathbf{\Pi}^\natural$ from the pair $(\mathbf{Y}, \mathbf{X})$. That is, the error rate for the permutation recovery sharply drops to zero once certain parameters reach the thresholds. In particular, our analysis will identify the precise positions of the phase transition points in the large-system limit, i.e., $n$, $m$, $p$, and $h$ all approach to infinity with $m/n \to \tau_m$, $p/n \to \tau_p$, $h/n \to \tau_h$. We will separately study the phase transition phenomenon in 1) the oracle case where $\mathbf{B}^\natural$ is given as a prior, and 2) the non-oracle case where $\mathbf{B}^\natural$ is unknown.

In this section, we consider the oracle scenario, as a warm-up example. To reconstruct the permutation matrix $\mathbf{\Pi}^\natural$, we adopt the following *maximum-likelihood* (ML) estimator:

$$\widehat{\mathbf{\Pi}}^{\text{oracle}} = \text{argmin}_{\mathbf{\Pi} \in \mathcal{P}_n} \left\langle \mathbf{\Pi}, -\mathbf{Y}\mathbf{B}^{\natural\top}\mathbf{X}^\top \right\rangle. \tag{9}$$

Denoting the variable $\mathbf{E}_{ij}^{\text{oracle}}$ as $-\mathbf{X}_{\pi^\natural(i)}^\top \mathbf{B}^\natural\mathbf{B}^{\natural\top}\mathbf{X}_j - \sigma \mathbf{W}_i^\top \mathbf{B}^{\natural\top}\mathbf{X}_j$, $(1 \leq i, j \leq n)$, we can transform the objective function in (9) as the canonical form of LAP, i.e., $\sum_{i,j} \mathbf{\Pi}_{ij}\mathbf{E}_{ij}^{\text{oracle}}$.

### 3.1 The phase transition threshold for the oracle case

In the oracle case where $\mathbf{B}^\natural$ is known, we define the following random variable $\Xi$:

$$\Xi = \boldsymbol{x}^\top \mathbf{B}^\natural\mathbf{B}^{\natural\top}(\boldsymbol{x} - \boldsymbol{y}) + \sigma \boldsymbol{w}\mathbf{B}^{\natural\top}(\boldsymbol{x} - \boldsymbol{y}), \tag{10}$$

where $\boldsymbol{x}$ and $\boldsymbol{y}$ follow the distribution $\mathsf{N}(\mathbf{0}, \mathbf{I}_{p\times p})$, and $\boldsymbol{w}$ follows the distribution $\mathsf{N}(\mathbf{0}, \mathbf{I}_{m\times m})$.

**Assumption 1.** *We ignore the weak correlation across the $\mathbf{E}_{ij}^{\text{oracle}}$ and view the corresponding $\Xi_i$ as i.i.d. copies of* (10)*.*

Numerical experiments show that we can safely adopt Assumption 1 without much sacrifice in the prediction accuracy, see Table 1 and 2. Recalling Theorem 1, we predict the critical points by letting

$$\inf_{\theta>0} {}^1/\theta \cdot \log\left(\sum_{i=1}^n \mathbb{E}e^{-\theta\Xi_i}\right) = \inf_{\theta>0} {}^1/\theta \cdot \left(\log n + \log \mathbb{E}e^{-\theta\Xi}\right) = 0. \tag{11}$$

The computation procedure consists of two stages:

- **Stage I.** We compute the optimal $\theta_*$, which is written as $\theta_* = \text{argmin}_{\theta>0} {}^1/\theta \cdot \left(\log n + \log \mathbb{E}e^{-\theta\Xi_i}\right)$.
- **Stage II.** We plug the optimal $\theta^*$ into (11) and obtain the phase transition snr accordingly.

The following context illustrates the computation details.

**Stage I: Determine $\theta_*$.** The key in determining $\theta_*$ lies in the computation of $\mathbb{E}e^{-\theta\Xi}$, which is summarized in the following proposition.

**Lemma 1.** *For the random variable $\Xi$ defined in* (10)*, we can write its expectation as*

$$\mathbb{E}e^{-\theta\Xi} = \prod_{i=1}^{\text{rank}(\mathbf{B}^\natural)} \left[1 + 2\theta\lambda_i^2 - \theta^2\lambda_i^2\left(\lambda_i^2 + 2\sigma^2\right)\right]^{-\frac{1}{2}}, \tag{12}$$

*provided that*

$$\theta^2\sigma^2\lambda_i^2 < 1 \text{ and } \theta^2\lambda_i^2\left(\lambda_i^2 + 2\sigma^2\right) \leq 1 + 2\theta\lambda_i^2 \tag{13}$$

*hold for all singular values $\lambda_i$ of $\mathbf{B}^\natural$, $1 \leq i \leq \text{rank}(\mathbf{B}^\natural)$.*

**Remark 3.** *When the conditions in* (13) *is violated, we have the expectation $\mathbb{E}e^{-\theta\Xi}$ diverge to infinity, which suggests the optimal $\theta_*$ for $\inf_{\theta>0} \log\left(n\cdot\mathbb{E}e^{-\theta\Xi}\right)/\theta$ cannot be achieved.*

With (12), we can compute the optimal $\theta_*$ by setting the gradient $\frac{\partial\left[\log(n\cdot\mathbb{E}e^{-\theta\Xi})/\theta\right]}{\partial\theta} = 0$. However, a closed-form of the exact solution for $\theta^*$ is out of reach. As a mitigation, we resort to approximating $\log\mathbb{E}e^{-\theta\Xi}$ by its lower-bound, which reads as

$$\log\mathbb{E}e^{-\theta\Xi} \geq \frac{\theta^2}{2}\left(\left\|\mathbf{B}^{\natural\top}\mathbf{B}^\natural\right\|_{\text{F}}^2 + 2\sigma^2\left\|\mathbf{B}^\natural\right\|_{\text{F}}^2\right) - \theta\left\|\mathbf{B}^\natural\right\|_{\text{F}}^2.$$

The corresponding minimum value $\widetilde{\theta}_*$ is thus obtained by minimizing the lower-bound, which is written as $\widetilde{\theta}_* = 2\log n / \left(\left\|\mathbf{B}^{\natural\top}\mathbf{B}^\natural\right\|_{\text{F}}^2 + 2\sigma^2\left\|\mathbf{B}^\natural\right\|_{\text{F}}^2\right)$.

**Stage II: Compute the phase transition** snr. We predict the phase transition point $\mathsf{snr}_{\text{oracle}}$ by letting the lower bound being zero, which can be written as

$$\frac{\log n}{\theta^*} - \left\|\mathbf{B}^\natural\right\|_{\text{F}}^2 + \frac{\theta^*}{2}\left(\left\|\mathbf{B}^{\natural\top}\mathbf{B}^\natural\right\|_{\text{F}}^2 + 2\sigma^2\left\|\mathbf{B}^\natural\right\|_{\text{F}}^2\right) = 0.$$

With standard algebraic manipulations, we have

**Proposition 1.** *The predicted phase transition for the oracle case in* (9) *can be computed as*

$$2(\log n)\mathsf{snr}_{\text{oracle}} \cdot \left\|\mathbf{B}^{\natural\top}/\|\mathbf{B}^\natural\|_{\text{F}} \cdot \mathbf{B}^\natural/\|\mathbf{B}^\natural\|_{\text{F}}\right\|_{\text{F}}^2 + 4\log n/m = \mathsf{snr}_{\text{oracle}}. \tag{14}$$

To evaluate the accuracy of our predicted phase transition threshold, we compare the predicted values with the numerical values (c.f. Section A). The results are shown in Table 1, from which we can conclude the phase transition threshold snr can be predicted to a good extent. In addition, we observe that the gap between the theoretical values and the numerical values keeps shrinking as $m$ increases.

Table 1: Comparison between the predicted value of the phase transition threshold $\mathsf{snr}_{\text{oracle}}$ in Proposition 1 and its simulated value when $n = 500$. **P** denotes the predicted value while **S** denotes the simulated value (i.e., mean $\pm$ std). **S** corresponds to the snr when the error rate drops below 0.05. A detailed description of the numerical method can be found in the appendix (code also included).

| $m$ | 20 | 30 | 40 | 50 | 60 | 70 |
|---|---|---|---|---|---|---|
| **P** | 3.283 | 1.415 | 0.902 | 0.662 | 0.523 | 0.432 |
| **S** | $2.529 \pm 0.079$ | $1.290 \pm 0.054$ | $0.872 \pm 0.034$ | $0.649 \pm 0.012$ | $0.515 \pm 0.016$ | $0.429 \pm 0.015$ |

| $m$ | 100 | 110 | 120 | 130 | 140 | 150 |
|---|---|---|---|---|---|---|
| **P** | 0.284 | 0.255 | 0.231 | 0.211 | 0.195 | 0.181 |
| **S** | $0.282 \pm 0.008$ | $0.256 \pm 0.006$ | $0.232 \pm 0.006$ | $0.212 \pm 0.004$ | $0.196 \pm 0.006$ | $0.183 \pm 0.005$ |

### 3.2 GAUSSIAN APPROXIMATION OF THE PHASE TRANSITION THRESHOLD

From the above analysis, we can see that deriving a closed-form expression of the infimum value $\theta$ of $\log(n\mathbb{E}e^{-\theta\Xi})/\theta$ can be difficult. In fact, in certain scenarios, even obtaining a closed-form expression of $\mathbb{E}e^{-\theta\Xi}$ is difficult. To handle such challenge, we propose to approximate random variable $\Xi$ by a Gaussian $\mathsf{N}(\mathbb{E}\Xi, \text{Var}\Xi)$, namely,

$$\mathbb{E}e^{-\theta\Xi} \approx \exp\left(-\theta\mathbb{E}\Xi + \frac{\theta^2}{2}\text{Var}\Xi\right). \tag{15}$$

With this approximation, we can express $\theta_* \triangleq \inf \log(n \cdot \mathbb{E}e^{-\theta\Xi})/\theta$ in a closed form, which is $\sqrt{2\log n/\text{Var}\Xi}$.

**Theorem 2.** *For the random variable $\Xi$ defined in* (10), *its mean and variance can be computed as*

$$\mathbb{E}\Xi = \left\|\mathbf{B}^\natural\right\|_{\text{F}}^2, \quad \text{Var}\Xi = 3\left\|\mathbf{B}^\natural\mathbf{B}^{\natural\top}\right\|_{\text{F}}^2 + 2\sigma^2\left\|\mathbf{B}^\natural\right\|_{\text{F}}^2. \tag{16}$$

Then, we can predict the phase transition point as follows.

**Proposition 2.** *With Gaussian approximation, we can predict the critical point corresponding to the phase transition in* (11) *as*

$$2(\log n) \cdot \text{Var}\Xi = \left(\mathbb{E}\Xi\right)^2, \tag{17}$$

*where $\mathbb{E}\Xi$ and $\text{Var}\Xi$ can be found in Theorem 2.*

**Example 1** (Scaled identity matrix). *We consider the scenario where $\mathbf{B}^\natural = \lambda\mathbf{I}_{m\times m}$. Then, we have $\mathbf{B}^\natural/\|\mathbf{B}^\natural\|_{\text{F}} = m^{-1/2}\mathbf{I}$. The phase transition threshold $\mathsf{snr}_{\text{oracle}}$ in* (14) *is then $4\log n/(m-2\log n)$, and the phase transition threshold $\widetilde{\mathsf{snr}}_{\text{oracle}}$ in* (17) *as $4\log n/(m-6\log n)$. This solution is almost identical to* (14) *in the limit as $\mathsf{snr}_{\text{oracle}} \approx \widetilde{\mathsf{snr}}_{\text{oracle}} \approx 4\log n/m \simeq n^{\frac{4}{m}} - 1$.*

Moreover, we should mention that 1) our approximation method applies to a general matrix $\mathbf{B}^\natural$, not limited to a scaled identity matrix; and 2) our approximation method can also predict the phase transition thresholds to a good extent when the entries $\mathbf{X}_{ij}$ are sub-Gaussian. The numerical experiments are given in Table 2, from which we can conclude that the predicted values are well aligned with the simulation results.

**Remark 4.** *In addition, we want to discuss one newly-released paper (Lufkin et al., 2024), which studies the same topic but in a much simpler setting (i.e., single measurement with $m = 1$). Compared with their work, our framework can easily produce results that the rigorous method in Lufkin et al. (2024) regards as an open problem. For example, the results (on the oracle case) which we treat as a warm-up example, are unsolved and specifically mentioned in the last paragraph. In particular, our framework can derive their proposed conjectured phase transition $\mathsf{snr} = n^{4/m}$. Moreover, our framework can also tell when the conjecture holds, i.e., $\mathbf{B}^\natural$ is an identity matrix, and our predictions' accuracy has been extensively verified by numerical experiments.*

*Other advantages of our work include 1) our ability to more accurately pinpoint phase transition points (their work can only obtain the lower bound of the phase transition point, while our work can predict the precise location); 2) our applicability to a wider array of cases (we cover the case when $m > 1$ while their method only works for $m = 1$); and 3) our consolidation into a more cohesive framework to predict the phase transition point.*

Table 2: Comparison between the predicted value of the phase transition threshold $\widetilde{\mathsf{snr}}_{\text{oracle}}$ in Proposition 2 and its simulated value when $n = 600$. In (**Case 1**), half of singular values are with $\lambda$ and the other half are with $\lambda/2$; while in (**Case 2**), half of the singular values are with $\lambda$ and the other half are with $^{(3 \cdot \lambda)}/4$. **Gauss** refers to $\mathbf{X}_{ij} \overset{\text{i.i.d}}{\sim} \mathsf{N}(0, 1)$ and **Unif** refers to $\mathbf{X}_{ij} \overset{\text{i.i.d}}{\sim} \mathsf{Unif}[-1, 1]$. **P** denotes the predicted value and **S** denotes the simulated value (i.e., mean $\pm$ std). **S** corresponds to the snr when the error rate drops below $0.05$. Averaged over 20 repetitions.

| $m$ | | 100 | 110 | 120 | 130 | 140 | 150 |
|---|---|---|---|---|---|---|---|
| (**Case 1**) **P** | | 0.297 | 0.266 | 0.241 | 0.220 | 0.203 | 0.188 |
| (**Gauss**) **S** | | $0.307 \pm 0.009$ | $0.275 \pm 0.005$ | $0.246 \pm 0.006$ | $0.227 \pm 0.007$ | $0.210 \pm 0.005$ | $0.194 \pm 0.004$ |
| (**Unif**) **S** | | $0.294 \pm 0.008$ | $0.266 \pm 0.005$ | $0.239 \pm 0.008$ | $0.216 \pm 0.004$ | $0.201 \pm 0.005$ | $0.189 \pm 0.006$ |
| | | | | | | | |
| (**Case 2**) **P** | | 0.310 | 0.276 | 0.249 | 0.227 | 0.209 | 0.193 |
| (**Gauss**) **S** | | $0.294 \pm 0.008$ | $0.266 \pm 0.006$ | $0.241 \pm 0.005$ | $0.220 \pm 0.004$ | $0.204 \pm 0.006$ | $0.190 \pm 0.003$ |
| (**Unif**) **S** | | $0.287 \pm 0.007$ | $0.255 \pm .0043$ | $0.234 \pm 0.007$ | $0.213 \pm 0.005$ | $0.197 \pm 0.003$ | $0.185 \pm 0.005$ |

## 4 EXTENSION TO NON-ORACLE CASE

Having analyzed the oracle case in the previous section, we now extend the analysis to the non-oracle case, where the value of $\mathbf{B}^\natural$ is not given. Different from the oracle case, the ML estimator reduces to a *quadratic assignment problem* (QAP) as opposed to LAP. As a mitigation, we adopt the estimator in Zhang & Li (2020), which reconstructs the permutation matrix within the LAP framework, i.e.,

$$\widehat{\mathbf{\Pi}}^{\text{non-oracle}} = \operatorname{argmin}_{\mathbf{\Pi} \in \mathcal{P}_n} \left\langle \mathbf{\Pi}, -\mathbf{Y}\mathbf{Y}^\top \mathbf{X}\mathbf{X}^\top \right\rangle. \tag{18}$$

We expect this estimator can yield good insights of the permuted linear regression since 1) this estimator can reach the statistical optimality in a broad range of parameters; and 2) estimator exhibits a phase transition phenomenon, a similar pattern as the oracle case. The technical details of the above claims can be found in Zhang & Li (2020).

Following the same procedure as in Section 3, we identify the phase transition threshold snr with Theorem 1. First, we rewrite the random variable $\Xi$ as

$$\Xi \cong \mathbf{Y}_i \mathbf{Y}^\top \mathbf{X} \left( \mathbf{X}_{\pi^\natural(i)} - \mathbf{X}_j \right)^\top, \tag{19}$$

where $i$ and $j$ are uniformly distributed among the set $\{1, 2, \cdots, n\}$. Afterwards, we adopt the Gaussian approximation scheme illustrated in Subsection 3.2 and determine the phase transition points by first computing $\mathbb{E}\Xi$ and $\mathrm{Var}\Xi$, respectively.

**Theorem 3.** *For the random variable $\Xi$ defined in (19), its mean $\mathbb{E}\Xi$ and variance $\mathrm{Var}\Xi$ are*

$$\mathbb{E}\Xi \simeq n \left(1 - \tau_h\right) \left[ \left(1 + \tau_p\right) \left\|\left|\mathbf{B}^\natural\right|\right\|_{\mathrm{F}}^2 + m\tau_p\sigma^2 \right],$$

$$\mathrm{Var}\Xi \simeq n^2 \tau_h \left(1 - \tau_h\right) \tau_p^2 \left[ \left\|\left|\mathbf{B}^\natural\right|\right\|_{\mathrm{F}}^2 + m\sigma^2 \right]^2 + n^2 \left[ 2\tau_p + 3\left(1 - \tau_h\right)^2 \right] \left\|\left|\mathbf{B}^{\natural\top}\mathbf{B}^\natural\right|\right\|_{\mathrm{F}}^2$$

$$+ n^2 \left[ 6\tau_p \left(1 - \tau_h\right)^2 + \left(3 - \tau_h\right) \tau_p^2 \right] \left\|\left|\mathbf{B}^{\natural\top}\mathbf{B}^\natural\right|\right\|_{\mathrm{F}}^2,$$

*respectively, where the definitions of $\tau_p$ and $\tau_h$ can be found in Section 3.*

The proof of Theorem 3 is quite complicated, involving Wick's theorem, Stein's lemma, the conditional technique, and the leave-one-out technique, etc. We defer the technical details to Section C.

### 4.1 An illustrating example

Afterwards, we predict the phase transition points. Unlike the oracle case, we notice the edge weights $\mathbf{E}_{ij}$ are strongly correlated, especially when $j = \pi^\natural(j)$, which corresponds to the non-permuted rows. To factor out these dependencies, we only take the permuted rows into account and correct the sample size from $n$ to $\tau_h n$.

**Proposition 3.** *The predicted* snr$_{\text{non-oracle}}$ *for the non-oracle case in* (18) *can be computed by solving*

$$2\log(n\tau_h)\mathrm{Var}\Xi = (\mathbb{E}\Xi)^2,$$

*where* $\mathbb{E}\Xi$ *and* $\mathrm{Var}\Xi$ *are in Theorem 3.*

To illustrate the prediction accuracy, we consider the case where $\mathbf{B}^\natural$'s singular values are of the same order, i.e., $\frac{\lambda_i(\mathbf{B}^\natural)}{\lambda_j(\mathbf{B}^\natural)} = O(1)$, $1 \le i, j \le m$, where $\lambda_i(\cdot)$ denotes the $i$-th singular value. Then, we obtain the snr$_{\text{non-oracle}}$, which is written as

$$\mathsf{snr}_{\text{non-oracle}} \approx \eta_1/\eta_2. \tag{20}$$

Here, $\eta_1$ and $\eta_2$ are defined as

$$\eta_1 \triangleq 2\tau_h\tau_p^2 \log(n\tau_h) - \tau_p(\tau_p+1)(1-\tau_h) + \tau_p\sqrt{2(1-\tau_h)\tau_h \cdot \log(n\tau_h)},$$

$$\eta_2 \triangleq (1-\tau_h)(\tau_p+1)^2 - 2\tau_h\tau_p^2\log(n\tau_h).$$

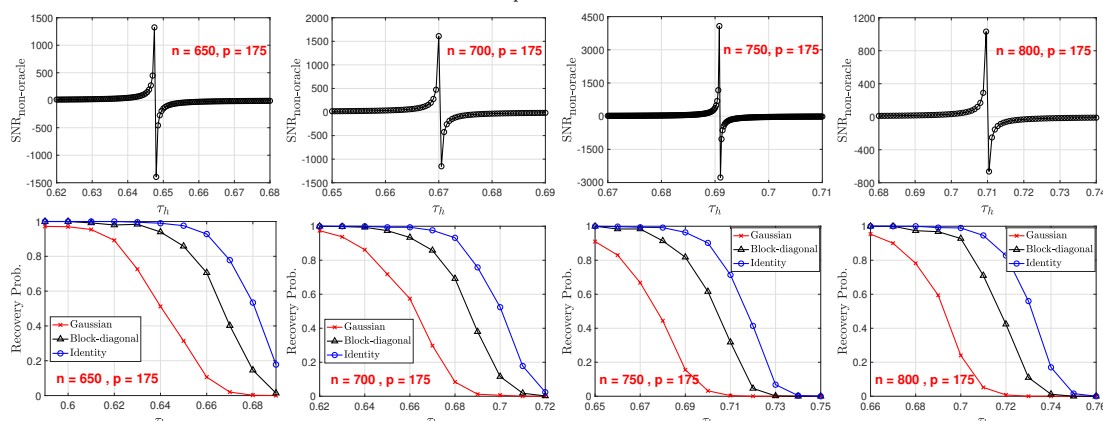

Figure 2: (**Upper panel**) Predicted phase transition points snr$_{\text{non-oralce}}$. (**Lower panel**) Plot of the recovery rate under the noiseless setting, i.e., snr $= \infty$. **Gaussian**: $\mathbf{B}^\natural_{ij} \overset{\text{i.i.d}}{\sim} \mathsf{N}(0,1)$; **Identity**: $\mathbf{B}^\natural = \mathbf{I}_{p\times p}$; **Block-diagonal**: $\mathbf{B}^\natural = \mathrm{diag}\{1,\cdots,1,0.5,\cdots,0.5\}$. We observe that the correct recovery rates drop sharply within the regions of our predicted value.

Note that the predicted snr$_{\text{non-oracle}}$ varies for different $\tau_h$ and $\tau_p$. Viewing snr$_{\text{non-oracle}}$ as a function of $\tau_h$, we observe a singularity point of $\tau_h$, which corresponds to the case when $\eta_2 = 0$. This suggests a potential phase transition phenomenon w.r.t. $\tau_h$. To validate the predicted phenomenon, we consider the noiseless case, i.e., snr $= \infty$, and reconstruct the permutation matrix $\mathbf{\Pi}^\natural$ with (2). Numerical experiments in Figure 2 confirm our prediction.

Due to the space limit, this section only presents a glimpse of our results in the non-oracle case. The technical details along with the additional numerical experiments can be found in Section C.

## 5 Conclusions

This is the first work that can identify the precise location of phase transition thresholds of permuted linear regressions. For the oracle case where the signal $\mathbf{B}^\natural$ is given as a prior, our analysis can predict the phase transition threshold snr$_{\text{oracle}}$ to a good extent. For the non-oracle case where $\mathbf{B}^\natural$ is not given, we modified the leave-one-out technique to approximately compute the phase critical snr$_{\text{non-oracle}}$ value for the phase transition, as the precise computation becomes significantly complicated as the high-order interaction between Gaussian random variables is involved. Moreover, we associated the singularity point in snr$_{\text{non-oracle}}$ with a phase transition point w.r.t the allowed number of permuted rows. Moreover, we present numerous numerical experiments to confirm the accuracy of our theoretical predictions.

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
