{1}\left(1 - \sum_j \mathbf{\Pi}_{ij}\right) \prod_j \mathbb{1}\left(1 - \sum_i \mathbf{\Pi}_{ij}\right) \times \exp\left(-\beta \sum_{i,j} \mathbf{\Pi}_{ij} \mathbf{E}_{ij}\right), \tag{3}$$

where $\mathbb{1}(\cdot)$ is the indicator function, $Z$ is the normalization constant of the probability measure $\mu(\mathbf{\Pi})$, and $\beta > 0$ is an auxiliary parameter. It is easy to verify the following two properties, e.g., 1) ML estimator in (2) can be rewritten as $\widehat{\mathbf{\Pi}} = \mathrm{argmax}_{\mathbf{\Pi}} \mu(\mathbf{\Pi})$[1]; and 2) the probability measure $\mu(\mathbf{\Pi})$ concentrates on $\widehat{\mathbf{\Pi}}$ when letting $\beta \to \infty$.

Then, we study the impact of $\{\mathbf{E}_{ij}\}$ on the reconstructed permutation $\widehat{\mathbf{\Pi}}$ with the *message passing* (MP) algorithm. First, we associate a probabilistic graphical model with the probability measure defined in (3). Then, we rewrite the solution in (2) in the language of the MP algorithm. Finally, we derive an equation (7) to track the convergence of the MP algorithm. By exploiting relation of (7) to the *branching random walk* (BRW) process, we can identify the phase transition points corresponding to the LAP in (2).

### 2.1 CONSTRUCTION OF THE GRAPHICAL MODEL

First, we construct the factor graph associated with the probability measure in (3). Adopting the same strategy as in Chapter 16 of Mezard & Montanari (2009), we conduct the following operations, e.g., 1) associating each variable $\mathbf{\Pi}_{ij}$ a variable node $v_{ij}$; 2) associating the variable node $v_{ij}$ a function node representing the term $e^{-\beta \mathbf{\Pi}_{ij} \mathbf{E}_{ij}}$; and 3) linking each constraint $\sum_i \mathbf{\Pi}_{ij} = 1$ to a function node and similarly for the constraint $\sum_j \mathbf{\Pi}_{ij} = 1$. A graphical representation is available in Figure 1.

Now we briefly review the MP algorithm. Informally speaking, MP is a local algorithm to compute the marginal probabilities over the graphical model. In each iteration, the variable node $v$ transmits the message to its incident function node $f$ by multiplying all incoming messages except the message along the edge $(v, f)$. The function node $f$ transmits the message to its incident variable node $v$ by computing the weighted summary of all incoming messages except the message along the edge $(f, v)$. For a detailed introduction to MP, we refer readers to Kschischang et al. (2001), Chapter 16 in MacKay et al. (2003), and Chapter 14 in Mezard & Montanari (2009).

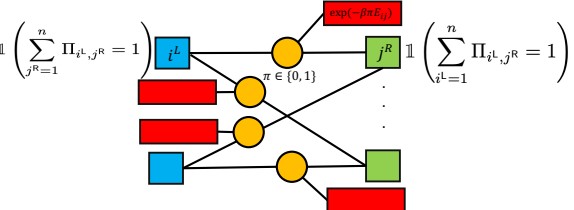

Figure 1: The constructed graphical model. Circle icons denote the variable nodes and square icons denote the function nodes: blue squares (green squares resp.) for the constraints on the rows (columns resp.) of $\mathbf{\Pi}$, and red squares for the function $e^{-\beta \pi \mathbf{E}_{ij}}$.

It is known that MP can obtain the exact marginals (Mezard & Montanari, 2009) for singly connected graphical models. For other types of graphs, however, whether MP can obtain the exact solution still remains an open problem (Cantwell & Newman, 2019; Kirkley et al., 2021). At the same time,

---

[1] Notice that the requirement $\mathbf{\Pi} \in \mathcal{P}_n$ is incorporated in $\mu(\mathbf{\Pi})$ implicitly and thus we do not need an explicit constraint.

numerical evidences have been witnessed to show that MP can yield meaningful results for graphs with loops; particular examples include applications in the coding theory (Chung, 2000; Richardson & Urbanke, 2001; 2008) and the LAP (which happens to be our case) (Mezard & Montanari, 2009; Chertkov et al., 2010; Caracciolo et al., 2017; Malatesta et al., 2019; Semerjian et al., 2020).

## 2.2 THE MESSAGE PASSING (MP) ALGORITHM

Next, we perform permutation recovery via MP. The following derivation follows the standard procedure, which can be found in the previous works (Mezard & Montanari, 2009; Semerjian et al., 2020). We denote the message flow from the node $i^L$ to the variable node $(i^L, j^R)$ as $\widehat{m}_{i^L \to (i^L, j^R)}(\cdot)$ and that from the edge $(i^L, j^R)$ to node $i^L$ as $m_{(i^L, j^R) \to i^L}(\cdot)$. Similarly, we define $\widehat{m}_{j^R \to (i^L, j^R)}(\cdot)$ and $m_{(i^L, j^R) \to j^R}(\cdot)$ as the message flow transmitted between the functional node $j^R$ and the variable node $(i^L, j^R)$. Here the superscripts L and R are used to indicate the positions of the node (left and right, respectively). Roughly speaking, these transmitted messages can be viewed as (unnormalized) conditional probability $\mathbb{P}(\Pi_{i,j} = \{0, 1\}|(\cdot))$ with the joint PDF being defined in (3). The message transmission process is to iteratively compute these conditional probabilities.

First, we consider the message flows transmitted between the functional node $i^L$ and the variable node $(i^L, j^R)$, which are written as

$$m_{(i^L, j^R) \to i^L}(\pi) \cong \widehat{m}_{j^R \to (i^L, j^R)}(\pi) e^{-\beta \pi \mathbf{E}_{i^L, j^R}},$$

$$\widehat{m}_{i^L \to (i^L, j^R)}(\pi) \cong \sum_{\pi_{i^L, k^R}} \prod_{k^R \neq j^R} \widehat{m}_{k^R \to (i^L, k^R)}(\pi_{i^L, k^R}) \times e^{-\beta \pi_{i^L, k^R} \mathbf{E}_{i^L, k^R}} \mathbb{1}(\pi + \sum_k \pi_{i^L, k^R} = 1), \quad (4)$$

where $\pi \in \{0, 1\}$ is a binary value. Similarly, we can write the message flows between the functional node $j^R$ and the variable node $(i^L, j^R)$, which are denoted as $m_{(i^L, j^R) \to j^R}(\pi)$ and $\widehat{m}_{j^R \to (i^L, j^R)}(\pi)$, respectively. With the parametrization approach, we define

$$h_{i^L \to (i^L, j^R)} \triangleq \frac{1}{\beta} \log \frac{\widehat{m}_{i^L \to (i^L, j^R)}(1)}{\widehat{m}_{i^L \to (i^L, j^R)}(0)}, \quad h_{j^R \to (i^L, j^R)} \triangleq \frac{1}{\beta} \log \frac{\widehat{m}_{j^R \to (i^L, j^R)}(1)}{\widehat{m}_{j^R \to (i^L, j^R)}(0)}.$$

Following the routine derivations in MP, we get the edge selection criteria, i.e., we pick $\widehat{\pi}(i^L) = j^R$ if

$$h_{i^L \to (i^L, j^R)} + h_{j^R \to (i^L, j^R)} > \mathbf{E}_{i^L, j^R}; \quad (5)$$

otherwise, we have $\widehat{\pi}(i^L) \neq j^R$. Due to the fact that $\mu(\mathbf{\Pi})$ concentrates on $\widehat{\mathbf{\Pi}}$ when $\beta$ is sufficiently large, we can thus rewrite the MP update equation as

$$h_{i^L \to (i^L, j^R)} = \min_{k^R \neq j^R} \mathbf{E}_{i^L, k^R} - h_{k^R \to (i^L, k^R)}, \quad h_{j^R \to (i^L, j^R)} = \min_{k^L \neq i^L} \mathbf{E}_{k^L, j^R} - h_{k^L \to (k^L, j^R)}, \quad (6)$$

which is attained by letting $\beta \to \infty$.

## 2.3 IDENTIFICATION OF THE PHASE TRANSITION THRESHOLD

To identify the phase transition phenomenon inherent in the MP update equation (6), we follow the strategy in Semerjian et al. (2020) and divide all edges $(i^L, j^R)$ into two categories according to whether the edge $(i^L, j^R)$ corresponds to the ground-truth permutation matrix $\mathbf{\Pi}^\natural$ or not. Within each category, we assume the edges' weights and the message flows along them can be represented by independently identically distributed random variables.

For the edge $(i^L, \pi^\natural(i^L))$ for the ground-truth correspondence, we represent the random variable associated with the weight $\mathbf{E}_{ij}$ as $\Omega$. The random variable for the message flow along this edge is denoted $H$ (for both $h_{i^L \to (i^L, j^R)}$ and $h_{j^R \to (i^L, j^R)}$). For the rest of edges $(i^L, j^R)$ $(j^R \neq \pi^\natural(i^

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

## A    ANALYSIS OF ORACLE CASE: PROOF OF LEMMA 1

*Proof.* Denote the singular values of $\mathbf{B}^\natural$ as $\{\lambda_i\}_{i=1}^{\mathrm{rank}(\mathbf{B}^\natural)}$. We exploit the rotation invariance property of Gaussian random variables; and have $\boldsymbol{\Xi}$ be identically distributed as

$$\boldsymbol{\Xi} = \sum_{i=1}^{\mathrm{rank}(\mathbf{B}^\natural)} \lambda_i^2 x_i \left(x_i - y_i\right) + \sigma \sum_{i=1}^{\mathrm{rank}(\mathbf{B}^\natural)} \lambda_i w_i \left(x_i - y_i\right).$$

Due to the independence across $\boldsymbol{w}$, $\boldsymbol{x}$, and $\boldsymbol{y}$, we have

$$\mathbb{E}e^{-\theta\boldsymbol{\Xi}} = \prod_{i=1}^{\mathrm{rank}(\mathbf{B}^\natural)} \mathbb{E}_{x,y,w} \exp\left[-\theta\lambda_i^2 x\left(x-y\right) - \theta\sigma\lambda_i w\left(x-y\right)\right]$$

$$= \prod_{i=1}^{\mathrm{rank}(\mathbf{B}^\natural)} \mathbb{E}_{x,y} \exp\left(\frac{\theta\lambda_i^2(x-y)\left(\theta\sigma^2(x-y)-2x\right)}{2}\right)$$

$$\overset{①}{=} \prod_{i=1}^{\mathrm{rank}(\mathbf{B}^\natural)} \mathbb{E}_x \frac{\exp\left(\frac{\theta\lambda_i^2 x^2\left(\theta\left(\lambda_i^2+\sigma^2\right)-2\right)}{2-2\theta^2\lambda_i^2\sigma^2}\right)}{\sqrt{1-\theta^2\lambda_i^2\sigma^2}}$$

$$\overset{②}{=} \prod_{i=1}^{\mathrm{rank}(\mathbf{B}^\natural)} \left(1 + 2\theta\lambda_i^2 - \theta^2\lambda_i^2\left(\lambda_i^2 + 2\sigma^2\right)\right)^{-\frac{1}{2}},$$

where in ① we use the fact $\theta^2\sigma^2\lambda_i^2 < 1$ and in ② we use the fact $\theta^2\lambda_i^2\left(\lambda_i^2 + 2\sigma^2\right) \le 1 + 2\theta\lambda_i^2$. $\qquad\square$

## B    NUMERICAL METHODS FOR PHASE TRANSITION POINTS

We present the numerical method to compute the phase transition points in Table 1 and 2. To begin with, we notice that the correct recovery rate is in monotonic non-decreasing relation with the snr, that is, the error rate $\mathbb{P}(\widehat{\Pi} \neq \Pi^{\natural})$ for a larger snr is at least equal to, if not less than, that for a smaller snr. Thus, we propose a binary-search-based method to compute the phase transition points.

For each snr, we run (2) for 100 times and calculate the error rate of permutation recovery (full permutation recovery, namely, $\mathbb{P}(\widehat{\Pi} \neq \Pi^{\natural})$). If the error rate is below 0.05, we regard snr as above the phase transition point and try a smaller value. Otherwise, we will try a larger value. The detailed description is in Algorithm 1.

For each parameter setting, we run 20 times of Algorithm 1. Then, we estimate its mean and the standard deviation from these estimated phase transition points.

**Complexity analysis**. For a given precision threshold $\varepsilon$, each iteration (Line 3 to Line 14 in Algorithm 1) takes $O(\log \frac{1}{\varepsilon})$ rounds to converge and it runs the permutation recovery algorithm 100 times in each round. That means, we run $20 \times 100 \times \log_2(10^4)$ ($\approx 26575$) rounds of experiments for each parameter.

---

**Algorithm 1** Numerical method to compute the phase transition points.

---

1: **Initialization.** Set the initial search range for snr as $[l, r]$. Define the precision threshold $\varepsilon$.
2:
3: **while** $|l - r| > \varepsilon$ **do**
4:     Set $\mathsf{snr}_{\text{middle}} = \frac{l+r}{2}$.
5:     Given $\mathsf{snr}_{\text{middle}}$, we run (2) for 100 times.
6:     Compute the error rate of full permutation recovery, namely, $\mathbb{P}(\widehat{\Pi} \neq \Pi^{\natural})$.
7:
8:     **if** the error rate is below $0.05$ **then**
9:         $r \to \mathsf{snr}_{\text{middle}} - \varepsilon$,   # we have $\mathsf{snr}_{\text{middle}}$ be greater than the phase transition point
10:    **else**
11:        $l \to \mathsf{snr}_{\text{middle}} + \varepsilon$.   # we have $\mathsf{snr}_{\text{middle}}$ be no greater than the phase transition point
12:    **end if**
13:
14: **end while**
15:
16: **Output.** Return the phase transition point $\mathsf{snr}_{\text{middle}}$.

---

# C    ANALYSIS OF THE NON-ORACLE CASE

This section presents the technical details in analyzing the non-oracle case.

## C.1    ADDITIONAL NUMERICAL RESULTS

We consider the same settings as in Subsection 4.1. Here, we present additional numerical results to evaluate the prediction accuracy of our method.

### C.1.1    VERIFICATION OF PHASE TRANSITION POINTS

For the predicted phase transition $\mathsf{snr}_{\text{non-oracle}}$, we notice an increasing gap between the predicted value and the simulated value, unlike in the oracle case. This might be caused by the strong correlation across the edge weights $\{\mathbf{E}_{ij}\}_{1 \leq i,j \leq n}$, or due to the error with the approximation relation $\mathbb{E}e^{-\theta\Xi} \approx \mathbb{E}\exp\left(\theta\mathbb{E}\Xi - \theta^2 \text{Var}\Xi/2\right)$.

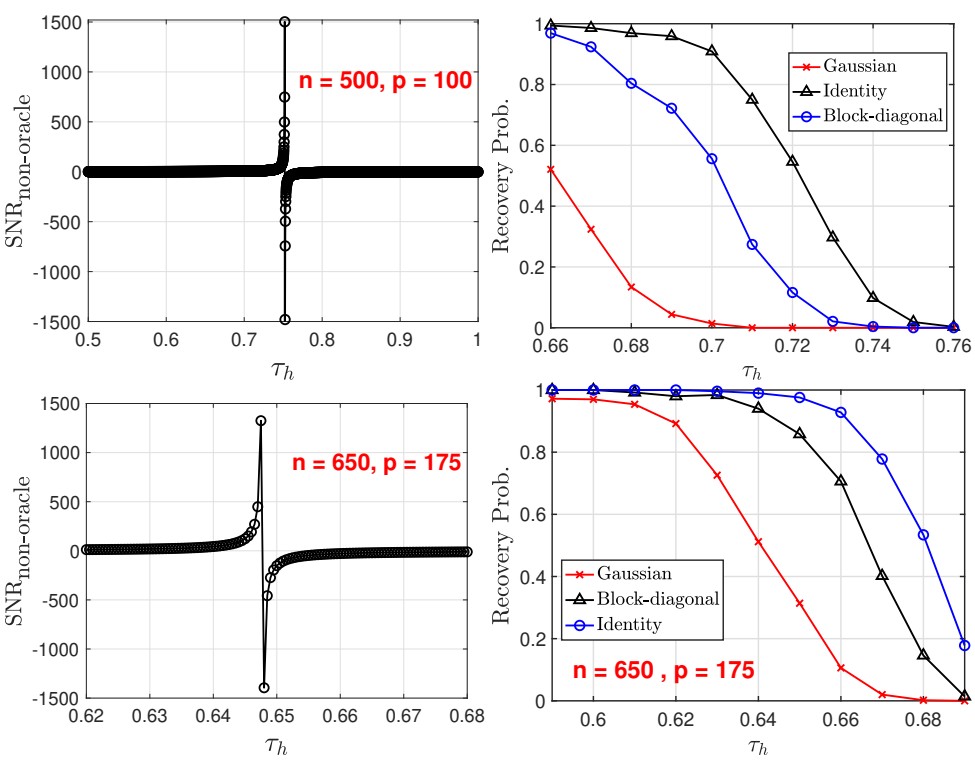

Figure 3:    (**Left panel**) Predicted $\mathsf{snr}_{\text{non-oralce}}$. (**Right panel**) Plot of recovery rate under the noiseless setting, i.e., $\mathsf{snr} = \infty$. **Gaussian**: $\mathbf{B}^{\natural}_{ij} \overset{\text{i.i.d}}{\sim} \mathsf{N}(0,1)$; **Identity**: $\mathbf{B}^{\natural} = \mathbf{I}_{p \times p}$; **Block-diagonal**: $\mathbf{B}^{\natural} = \text{diag}\left\{1, \cdots, 1, 0.5, \cdots, 0.5\right\}$.

Additional experiments are available in Table 3, from which we conclude the solution (20) can predict the phase transition point w.r.t. $\tau_h$ to a good extent.

Table 3: Comparison between the predicted value of the phase transition threshold $\tau_h$ and its simulated value when $n = 500$. **P** denotes the **predicted value** while **S** denotes the **simulated value**. **S** corresponds to the $\tau_h$ when the error rate drops below 0.05. We adopt a similar algorithm as in Algorithm 1. The only difference is we replace (2) in Line 5 with (18).

| $p$ | 75 | 100 | 125 | 150 | 175 | 200 |
|---|---|---|---|---|---|---|
| **P** | 0.82 | 0.73 | 0.68 | 0.62 | 0.56 | 0.52 |
| **S** | 0.77 | 0.75 | 0.7 | 0.66 | 0.61 | 0.57 |

### C.1.2 IMPACT OF $n$ ON THE PHASE TRANSITION POINT

We study the impact of $n$ on $\tau_h$. The numerical experiment is shown in the top row of Figure 3, from which we can see the predicted phase transition $\tau_h$ matches to a good extent to the numerical experiments. Then, we fix the $p$ and study the impact of $n$ on $\tau_h$. We observe that the phase transition $\tau_h$ increases together with the sample number $n$, which is also captured by our formula in (20).

### C.1.3 LIMITS OF $\tau_h$

We consider the limiting behavior of $\tau_h$ when $\tau_p$ approaches 0, or equivalently, $p = o_{\mathrm{P}}(n)$. We can simplify $\mathbb{E}\Xi$ and $\mathrm{Var}\Xi$ in Theorem 3 as

$$\mathbb{E}\Xi \simeq n(1 - \tau_h) \left\| \mathbf{B}^{\natural} \right\|_{\mathrm{F}}^2,$$

$$\mathrm{Var}\Xi \simeq 3n^2 (1 - \tau_h)^2 \left\| \mathbf{B}^{\natural\top} \mathbf{B}^{\natural} \right\|_{\mathrm{F}}^2.$$

We notice that the singularity point in (20) disappears. In other words, we can have the correct permutation matrix $\mathbf{\Pi}^{\natural}$ even when $h \approx n$. This is (partly) verified by Figure 4, from which we observe that the phase transition point w.r.t. $\tau_h$ approaches to one, or equivalently, $h$ approaches $n$, as $\tau_p$ decreases to zero.

### C.2 ANALYSIS OF NON-ORACLE CASE: PROOF OF THEOREM 3

This subsection presents the computational details of Theorem 3. To begin with, we decompose the random variable $\Xi$ as

$$\Xi = \Xi_1 + \sigma(\Xi_2 + \Xi_3) + \sigma^2 \Xi_4, \tag{21}$$

where $\Xi_i$ $(1 \leq i \leq 4)$ are respectively defined as

$$\Xi_1 \triangleq \mathbf{X}_{\pi^{\natural}(i)}^{\top} \mathbf{B}^{\natural} \mathbf{B}^{\natural\top} \mathbf{X}^{\top} \mathbf{\Pi}^{\natural\top} \mathbf{X} (\mathbf{X}_{\pi^{\natural}(i)} - \mathbf{X}_j),$$

$$\Xi_2 \triangleq \mathbf{X}_{\pi^{\natural}(i)}^{\top} \mathbf{B}^{\natural} \mathbf{W}^{\top} \mathbf{X} (\mathbf{X}_{\pi^{\natural}(i)} - \mathbf{X}_j),$$

$$\Xi_3 \triangleq \mathbf{W}_i^{\top} \mathbf{B}^{\natural\top} \mathbf{X}^{\top} \mathbf{\Pi}^{\natural\top} \mathbf{X} (\mathbf{X}_{\pi^{\natural}(i)} - \mathbf{X}_j),$$

$$\Xi_4 \triangleq \mathbf{W}_i^{\top} \mathbf{W}^{\top} \mathbf{X} (\mathbf{X}_{\pi^{\natural}(i)} - \mathbf{X}_j).$$

Unlike the oracle case, obtaining a closed-form expression of $\mathbb{E}e^{-\theta\Xi}$ would be too difficult. Hence, we adopt the Gaussian approximation method as presented in Section 3.2. The task then transforms to computing the expectation and variance of $\Xi$. Before delving into the technical details, we give a glimpse of our proof strategy.

**Computation of the mean $\mathbb{E}\Xi$.** For the computation of the mean $\mathbb{E}\Xi$, we can verify that $\mathbb{E}\Xi_2$ and $\mathbb{E}\Xi_3$ are both zero, due to the independence between $\mathbf{X}$ and $\mathbf{W}$. For $\mathbb{E}\Xi_1$ and $\mathbb{E}\Xi_4$, we adopt Wick's theorem to obtain

$$\mathbb{E}\Xi_1 = n(1 - \tau_h)(1 + \tau_p)[1 + o_{\mathrm{P}}(1)] \left\| \mathbf{B}^{\natural} \right\|_{\mathrm{F}}^2,$$

$$\mathbb{E}\Xi_4 = nm\tau_p(1 - \tau_h)[1 + o_{\mathrm{P}}(1)].$$

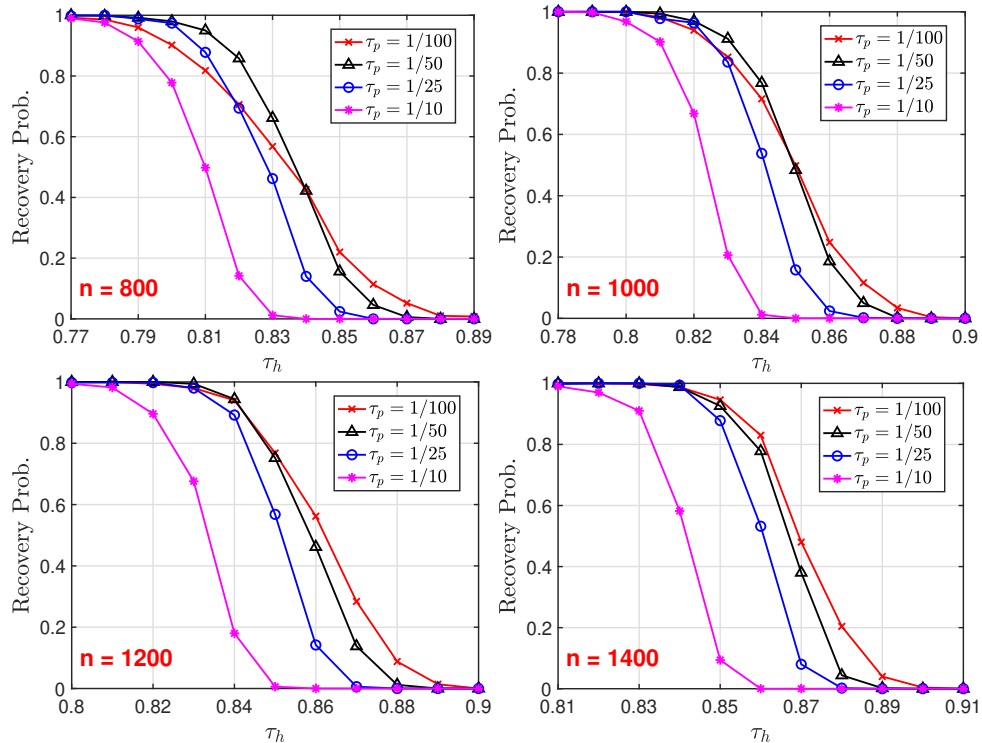

Figure 4: Plot of correct recovery rate w.r.t. $\tau_h$. We consider the noiseless scenario (i.e., snr $= \infty$) and pick $n = \{800, 1000, 1200, 1400\}$.

**Computation of the variance Var$\Xi$.**   Since Var$\Xi = \mathbb{E}\Xi^2 - (\mathbb{E}\Xi)^2$, we just need to compute $\mathbb{E}\Xi^2$, which can be expanded into the following six terms

$$\mathbb{E}\Xi^2 = \mathbb{E}\Xi_1^2 + \sigma^2\mathbb{E}\Xi_2^2 + \sigma^2\mathbb{E}\Xi_3^2 + \sigma^4\mathbb{E}\Xi_4^2 + 2\sigma^2\mathbb{E}\Xi_1\Xi_4 + 2\sigma^2\mathbb{E}\Xi_2\Xi_3.$$

The computation of above terms turns out to be quite complex due to the high order Gaussian random variables. For example, the term $\mathbb{E}\Xi_1^2$ involves the eighth-order Gaussian moments, the terms $\mathbb{E}\Xi_2^2, \mathbb{E}\Xi_3^2, \mathbb{E}\Xi_1\Xi_4$ and $\mathbb{E}\Xi_2\Xi_3$ all involve the sixth-order Gaussian variables, etc. To handle the difficulties in computing $\mathbb{E}\Xi^2$, we propose the following computation procedure, which can be roughly divided into 3 phases.

- **Phase I: Leave-one-out decomposition.** The major technical difficulty comes from the correlation between the product $\mathbf{X}^\top\mathbf{\Pi}^\natural\mathbf{X}$ and the difference $\mathbf{X}_{\pi^\natural(i)} - \mathbf{X}_j$. We decouple this correlation by first rewriting the matrix $\mathbf{X}^\top\mathbf{\Pi}^\natural\mathbf{X}$ as the sum $\sum_\ell \mathbf{X}_\ell\mathbf{X}_{\pi^\natural(\ell)}^\top$. Then we collect all terms $\mathbf{X}_\ell\mathbf{X}_{\pi^\natural(\ell)}^\top$ independent of $\mathbf{X}_{\pi^\natural(i)}$ and $\mathbf{X}_j$ in the matrix $\mathbf{\Sigma}$ and leave the remaining terms to the matrix $\mathbf{\Delta}$, i.e., $\mathbf{\Delta} \triangleq \mathbf{X}^\top\mathbf{\Pi}^\natural\mathbf{X} - \mathbf{\Sigma}$. This decomposition shares the same spirit as the leave-one-out technique (Karoui, 2013; Bai & Silverstein, 2010; Karoui, 2018; Sur et al., 2019). Then, we divide all terms in $\mathbb{E}\Xi^2$ into 3 categories: 1) terms only containing matrix $\mathbf{\Sigma}$; 2) terms containing both $\mathbf{\Sigma}$ and $\mathbf{\Delta}$; and 3) terms only containing $\mathbf{\Delta}$.
- **Phase II: Conditional technique.** Concerning the terms in the first two categories, which covers the majority of terms, we can exploit the independence among the rows in the sensing matrix $\mathbf{X}$. With the conditional technique, we can reduce the order of Gaussian moments by separately taking the expectation w.r.t $\mathbf{\Sigma}$ and w.r.t vectors $\mathbf{X}_{\pi^\natural(i)}$ and $\mathbf{X}_j$.
- **Phase III: Direct computation.** For the few terms in the third category (i.e., terms only containing $\mathbf{\Delta}$), we compute the high-order Gaussian moments by exhausting all terms and iterative applying of Wick's Theorem and Stein's Lemma, which can reduce the higher-order Gaussian moments to lower-orders.

The computation details are attached as follows.

### C.2.1 NOTATIONS

Note that our analysis can involve the terms containing $(\mathbf{X}_{\pi^\natural}(i) - \mathbf{X}_j)$ and $\mathbf{X}^\top \mathbf{\Pi}^\natural \mathbf{X}$ simultaneously. To decouple the dependence between $(\mathbf{X}_{\pi^\natural}(i) - \mathbf{X}_j)$ and $\mathbf{X}^\top \mathbf{\Pi}^\natural \mathbf{X}$, we first rewrite the matrix $\mathbf{X}^\top \mathbf{\Pi}^\natural \mathbf{X}$ as the sum $\sum_\ell \mathbf{X}_\ell \mathbf{X}_{\pi^\natural(\ell)}^\top$ and then collect all terms $\mathbf{X}_\ell \mathbf{X}_{\pi^\natural(\ell)}^\top$ independent of $\mathbf{X}_{\pi^\natural(i)}$ and $\mathbf{X}_j$ in the matrix $\mathbf{\Sigma}$, which is written as

$$\mathbf{\Sigma} \triangleq \sum_{\ell, \pi^\natural(\ell) \neq \pi^\natural(i), j} \mathbf{X}_\ell \mathbf{X}_{\pi^\natural(\ell)}^\top. \tag{22}$$

The rest terms are then put in the matrix $\mathbf{\Delta}$ such that $\mathbf{X}^\top \mathbf{\Pi}^\natural \mathbf{X} = \mathbf{\Sigma} + \mathbf{\Delta}$. Note that the expression of $\mathbf{\Delta}$ varies under different cases such that

- **Case $(s, s)$:** $i = \pi^\natural(i)$ **and** $j = \pi^\natural(j)$. We have

$$\mathbf{\Delta} = \mathbf{\Delta}^{(s,s)} = \mathbf{X}_i \mathbf{X}_i^\top + \mathbf{X}_j \mathbf{X}_j^\top. \tag{23}$$

- **Case $(s, d)$:** $i = \pi^\natural(i)$ **and** $j \neq \pi^\natural(j)$. We have

$$\mathbf{\Delta} = \mathbf{\Delta}^{(s,d)} = \mathbf{X}_i \mathbf{X}_i^\top + \mathbf{X}_j \mathbf{X}_{\pi^\natural(j)}^\top + \mathbf{X}_{\pi^{\natural-1}(j)} \mathbf{X}_j^\top. \tag{24}$$

- **Case $(d, s)$:** $i \neq \pi^\natural(i)$ **and** $j = \pi^\natural(j)$. We have

$$\mathbf{\Delta} = \mathbf{\Delta}^{(d,s)} = \mathbf{X}_i \mathbf{X}_{\pi^\natural(i)}^\top + \mathbf{X}_{\pi^\natural(i)} \mathbf{X}_{\pi^{\natural 2}(i)}^\top + \mathbf{X}_j \mathbf{X}_j^\top. \tag{25}$$

- **Case $(d, d)$:** $i \neq \pi^\natural(i)$ **and** $j \neq \pi^\natural(j)$. We have

$$\mathbf{\Delta} = \mathbf{\Delta}^{(d,d)} = \mathbf{X}_i \mathbf{X}_{\pi^\natural(i)}^\top + \mathbf{X}_{\pi^\natural(i)} \mathbf{X}_{\pi^{\natural 2}(i)}^\top + \mathbf{X}_j \mathbf{X}_{\pi^\natural(j)}^\top + \mathbf{X}_{\pi^{\natural-1}(j)} \mathbf{X}_j^\top. \tag{26}$$

In addition, we define the matrix $\mathbf{M}$ as $\mathbf{B}^\natural \mathbf{B}^{\natural\top}$, and define the index sets $\mathcal{S}, \mathcal{D}$, and $\mathcal{D}_{\text{pair}}$ as

$$\mathcal{S} \triangleq \left\{ \ell \mid \ell \neq i \text{ or } j, \ \ell = \pi^\natural(\ell) \right\}, \tag{27}$$

$$\mathcal{D} \triangleq \left\{ \ell \mid \ell, \pi^\natural(\ell) \neq i \text{ or } j, \ \ell \neq \pi^\natural(\ell) \right\}, \tag{28}$$

$$\mathcal{D}_{\text{pair}} \triangleq \left\{ (\ell_1, \ell_2) : \ell_1 = \pi^\natural(\ell_2), \ell_2 = \pi^\natural(\ell 1), \ell_1, \ell_2 \in \mathcal{D} \right\}, \tag{29}$$

respectively.

### C.2.2 MAIN COMPUTATION

In this case, we can write $\Xi$ as

$$\Xi = \underbrace{\mathbf{X}_{\pi^\natural(i)}^\top \mathbf{B}^\natural \mathbf{B}^{\natural\top} \mathbf{X}^\top \mathbf{\Pi}^{\natural\top} \mathbf{X} \left[ \mathbf{X}_{\pi^\natural(i)} - \mathbf{X}_j \right]}_{\triangleq \Xi_1} + \sigma \underbrace{\mathbf{X}_{\pi^\natural(i)}^\top \mathbf{B}^\natural \mathbf{W}^\top \mathbf{X} \left[ \mathbf{X}_{\pi^\natural(i)} - \mathbf{X}_j \right]}_{\triangleq \Xi_2}$$

$$+ \sigma \underbrace{\mathbf{W}_i^\top \mathbf{B}^{\natural\top} \mathbf{X}^\top \mathbf{\Pi}^{\natural\top} \mathbf{X} \left[ \mathbf{X}_{\pi^\natural(i)} - \mathbf{X}_j \right]}_{\triangleq \Xi_3} + \sigma^2 \underbrace{\mathbf{W}_i^\top \mathbf{W}^\top \mathbf{X} \left[ \mathbf{X}_{\pi^\natural(i)} - \mathbf{X}_j \right]}_{\triangleq \Xi_4}.$$

The following context separately computes its expectation $\mathbb{E}\Xi$ and its variance $\text{Var}\Xi$.

**Expectation.** We can easily verify that both $\mathbb{E}\Xi_2$ and $\mathbb{E}\Xi_3$ are zero. Then our goal turns to calculating the expectation of $\mathbb{E}\Xi_1$ and $\mathbb{E}\Xi_4$. First, we have

$$\mathbb{E}\Xi_1 = \mathbb{E} \sum_{\ell = \pi^\natural(\ell)} \mathbf{X}_{\pi^\natural(i)}^\top \mathbf{M} \mathbf{X}_\ell \mathbf{X}_\ell^\top \mathbf{X}_{\pi^\natural(i)} - \mathbb{E} \sum_\ell \mathbf{X}_{\pi^\natural(i)}^\top \mathbf{M} \mathbf{X}_{\pi^\natural(\ell)} \mathbf{X}_\ell^\top \mathbf{X}_j.$$

With Lemma 16 and Lemma 17, we conclude

$$\mathbb{E}\Xi_1 = (n - h) \text{Tr}(\mathbf{M}) + (p + 1) \mathbb{E} \mathbb{1}_{i = \pi^\natural(i)} \text{Tr}(\mathbf{M}) - \left( p \mathbb{E} \mathbb{1}_{i=j} + \mathbb{E} \mathbb{1}_{j = \pi^{\natural 2}(i)} \right) \text{Tr}(\mathbf{M})$$

$$= (n + p - h - hp/n) [1 + o(1)] \text{Tr}(\mathbf{M}). \tag{30}$$

Meanwhile, we have

$$\mathbb{E}\Xi_4 = \mathbb{E}\left[\mathbf{W}_i^\top \mathbf{W}_1 \cdots \mathbf{W}_i^\top \mathbf{W}_i \cdots \mathbf{W}_i^\top \mathbf{W}_n\right] \mathbf{X} \left(\mathbf{X}_{\pi^\natural(i)} - \mathbf{X}_j\right)$$

$$= m\mathbb{E}\mathbf{X}_i^\top \left(\mathbf{X}_{\pi^\natural(i)} - \mathbf{X}_j\right) = mp\left(\mathbb{E}\mathbb{1}_{i=\pi^\natural(i)} - \mathbb{E}\mathbb{1}_{i=j}\right) = \frac{mp(n-h)\sigma^2}{n}\left[1 + o(1)\right]. \quad (31)$$

Combining (30) and (31) and neglecting the $o(1)$ terms yields

$$\mathbb{E}\Xi \approx (n+p)\left(1 - h/n\right)\left\|\!\left\|\mathbf{B}^\natural\right\|\!\right\|_{\mathrm{F}}^2 + \frac{mp(n-h)\sigma^2}{n}.$$

**Variance.** Then we study the variance of $\Xi$. With the relation $\mathrm{Var}(\Xi) = \mathbb{E}\Xi^2 - (\mathbb{E}\Xi)^2$, our goal reduces to computing $\mathbb{E}\Xi^2$, which can be written as

$$\mathbb{E}\Xi^2 = \mathbb{E}\Xi_1^2 + \sigma^2\mathbb{E}\Xi_2^2 + \sigma^2\mathbb{E}\Xi_3^2 + \sigma^4\mathbb{E}\Xi_4^2 + 2\sigma^2\mathbb{E}\Xi_1\Xi_4 + 2\sigma^2\mathbb{E}\Xi_2\Xi_3.$$

The following context separately computes each terms

$$\mathbb{E}\Xi_1^2 \approx (n-h)^2\left(1 + \frac{2p}{n} + \frac{p^2}{n(n-h)}\right)\left[\mathrm{Tr}(\mathbf{M})\right]^2$$

$$+ n^2\left[\frac{2p}{n} + 3\left(1 - \frac{h}{n}\right)^2 + \frac{6(n-h)^2p}{n^3} + \frac{(3n-h)p^2}{n^3}\right]\mathrm{Tr}(\mathbf{MM}),$$

$$\mathbb{E}\Xi_2^2 \approx 2np\left(1 + p/n\right)\mathrm{Tr}(\mathbf{M}),$$

$$\mathbb{E}\Xi_3^2 \approx 2n^2\left(\frac{p}{n} + \left(1 - \frac{h}{n}\right)^2 + \frac{p^2}{n^2} + \frac{4p(n-h)^2}{n^3}\right)\mathrm{Tr}(\mathbf{M}),$$

$$\mathbb{E}\Xi_4^2 \approx \frac{(n-h)m^2p^2}{n},$$

$$\mathbb{E}\Xi_1\Xi_4 \approx \frac{mp(n-h)(n+p-h)}{n}\mathrm{Tr}(\mathbf{M}),$$

$$\mathbb{E}\Xi_2\Xi_3 \approx \frac{p(n-h)(n+p-h)}{n}\mathrm{Tr}(\mathbf{M}).$$

The detailed computation is attached as follows.

**Lemma 2.** *We have*

$$\mathbb{E}\Xi_1^2 = (n-h)^2\left(1 + \frac{2p}{n} + \frac{p^2}{n(n-h)} + o(1)\right)\left[\mathrm{Tr}(\mathbf{M})\right]^2$$

$$+ n^2\left[\frac{2p}{n} + 3\left(1 - \frac{h}{n}\right)^2 + \frac{6(n-h)^2p}{n^3} + \frac{(3n-h)p^2}{n^3} + o(1)\right]\mathrm{Tr}(\mathbf{MM}),$$

*where $\Xi_1$ is defined in (21).*

*Proof.* We begin the proof by decomposing $\Xi_1^2$ as

$$\mathbb{E}\Xi_1^2 = \mathbb{E}\underbrace{\left(\mathbf{X}_{\pi^\natural(i)} - \mathbf{X}_j\right)^\top \boldsymbol{\Sigma}\mathbf{M}\mathbf{X}_{\pi^\natural(i)}\mathbf{X}_{\pi^\natural(i)}^\top \mathbf{M}\boldsymbol{\Sigma}^\top \left(\mathbf{X}_{\pi^\natural(i)} - \mathbf{X}_j\right)}_{\Lambda_1}$$

$$+ 2\mathbb{E}\underbrace{\left(\mathbf{X}_{\pi^\natural(i)} - \mathbf{X}_j\right)^\top \boldsymbol{\Sigma}\mathbf{M}\mathbf{X}_{\pi^\natural(i)}\mathbf{X}_{\pi^\natural(i)}^\top \mathbf{M}\boldsymbol{\Delta}^\top \left(\mathbf{X}_{\pi^\natural(i)} - \mathbf{X}_j\right)}_{\Lambda_2}$$

$$+ \mathbb{E}\underbrace{\left(\mathbf{X}_{\pi^\natural(i)} - \mathbf{X}_j\right)^\top \boldsymbol{\Delta}\mathbf{M}\mathbf{X}_{\pi^\natural(i)}\mathbf{X}_{\pi^\natural(i)}^\top \mathbf{M}\boldsymbol{\Delta}^\top \left(\mathbf{X}_{\pi^\natural(i)} - \mathbf{X}_j\right)}_{\Lambda_3},$$

and separately bound each term as in Lemma 3, Lemma 4, and Lemma 5.

$\square$

**Lemma 3.** *We have*

$$\mathbb{E}\left(\mathbf{X}_{\pi^\natural(i)} - \mathbf{X}_j\right)^\top \mathbf{\Sigma M X}_{\pi^\natural(i)} \mathbf{X}_{\pi^\natural(i)}^\top \mathbf{M \Sigma}^\top \left(\mathbf{X}_{\pi^\natural(i)} - \mathbf{X}_j\right)$$

$$= (n-h)^2 \left(1 + o(1)\right) \left[\mathrm{Tr}(\mathbf{M})\right]^2 + n^2 \left[\frac{2p}{n} + 3\left(1 - \frac{h}{n}\right)^2 + o(1)\right] \mathrm{Tr}(\mathbf{MM}).$$

*Proof.* Due to the independence among different rows of the sensing matrix $\mathbf{X}$, we condition on $\mathbf{\Sigma}$ and take expectation w.r.t. $\mathbf{X}_{\pi^\natural(i)}$ and $\mathbf{X}_j$, which leads to

$$\mathbb{E}\Lambda_1 = \mathbb{E}\underbrace{\mathbf{X}_{\pi^\natural(i)}^\top \mathbf{\Sigma M X}_{\pi^\natural(i)} \mathbf{X}_{\pi^\natural(i)}^\top \mathbf{M \Sigma}^\top \mathbf{X}_{\pi^\natural(i)}}_{\Lambda_{1,1}} + \mathbb{E}\underbrace{\mathbf{X}_j^\top \mathbf{\Sigma M X}_{\pi^\natural(i)} \mathbf{X}_{\pi^\natural(i)}^\top \mathbf{M \Sigma}^\top \mathbf{X}_j}_{\Lambda_{1,2}}.$$

For $\mathbb{E}\Lambda_{1,1}$, we obtain

$$\mathbb{E}\Lambda_{1,1} \overset{①}{=} \mathbb{E}\left[\mathrm{Tr}\left(\mathbf{\Sigma M}\right)\mathrm{Tr}\left(\mathbf{\Sigma M}\right)\right] + \mathbb{E}\,\mathrm{Tr}\left(\mathbf{\Sigma MM}^\top\mathbf{\Sigma}^\top\right) + \mathbb{E}\,\mathrm{Tr}\left(\mathbf{\Sigma M\Sigma M}\right)$$

$$\overset{②}{=} (n-h)^2 \left[1 + o(1)\right] \left[\mathrm{Tr}(\mathbf{M})\right]^2 + n^2 \left[\frac{p}{n} + 2\left(1 - \frac{h}{n}\right)^2 + o(1)\right] \mathrm{Tr}(\mathbf{MM}),$$

where ① is due to (65), and ② is due to Lemma 13, Lemma 14, and Lemma 15. As for $\mathbb{E}\Lambda_{1,2}$, we have

$$\mathbb{E}\Lambda_{1,2} = \mathbb{E}\,\mathrm{Tr}\left(\mathbf{\Sigma MM}^\top\mathbf{\Sigma}^\top\right) = n^2 \left[\frac{p}{n} + \left(1 - \frac{h}{n}\right)^2 + o(1)\right] \mathrm{Tr}\left(\mathbf{M}^\top\mathbf{M}\right),$$

and hence complete the proof. □

**Lemma 4.** *We have*

$$\mathbb{E}\left(\mathbf{X}_{\pi^\natural(i)} - \mathbf{X}_j\right)^\top \mathbf{\Sigma M X}_{\pi^\natural(i)} \mathbf{X}_{\pi^\natural(i)}^\top \mathbf{M\Delta}^\top \left(\mathbf{X}_{\pi^\natural(i)} - \mathbf{X}_j\right) \approx \frac{(n-h)^2 p}{n}\left[\left(\mathrm{Tr}(\mathbf{M})\right)^2 + 3\,\mathrm{Tr}(\mathbf{MM})\right].$$

*Proof.* Similar as above, we first expand $\Lambda_2$ as

$$\mathbb{E}\Lambda_2 = (n-h)\mathbb{E}\underbrace{\mathbf{X}_{\pi^\natural(i)}^\top \mathbf{M X}_{\pi^\natural(i)} \mathbf{X}_{\pi^\natural(i)}^\top \mathbf{M\Delta}^\top \mathbf{X}_{\pi^\natural(i)}}_{\Lambda_{2,1}} + (n-h)\mathbb{E}\underbrace{\mathbf{X}_j^\top \mathbf{M X}_{\pi^\natural(i)} \mathbf{X}_{\pi^\natural(i)}^\top \mathbf{M\Delta}^\top \mathbf{X}_j}_{\Lambda_{2,2}}$$

$$- (n-h)\mathbb{E}\underbrace{\mathbf{X}_{\pi^\natural(i)}^\top \mathbf{M X}_{\pi^\natural(i)} \mathbf{X}_{\pi^\natural(i)}^\top \mathbf{M\Delta}^\top \mathbf{X}_j}_{\Lambda_{2,3}} - (n-h)\mathbb{E}\underbrace{\mathbf{X}_j^\top \mathbf{M X}_{\pi^\natural(i)} \mathbf{X}_{\pi^\natural(i)}^\top \mathbf{M\Delta}^\top \mathbf{X}_{\pi^\natural(i)}}_{\Lambda_{2,4}}.$$

**Case $(s,s)$: $i = \pi^\natural(i)$ and $j = \pi^\natural(j)$.** We first compute $\Lambda_{2,1}$ as

$$\mathbb{E}\Lambda_{2,1} = \underbrace{\mathbb{E}\mathbf{X}_i^\top \mathbf{M X}_i \mathbf{X}_i^\top \mathbf{M X}_i \mathbf{X}_i^\top \mathbf{X}_i}_{\mathbb{E}\|\mathbf{X}_i\|_2^2 \left(\mathbf{X}_i^\top \mathbf{M X}_i\right)^2} + \underbrace{\mathbb{E}\mathbf{X}_i^\top \mathbf{M X}_i \mathbf{X}_i^\top \mathbf{M X}_j \mathbf{X}_j^\top \mathbf{X}_i}_{\mathbb{E}\left(\mathbf{X}_i^\top \mathbf{M X}_i\right)^2}$$

$$= (p+5)\left[\left(\mathrm{Tr}(\mathbf{M})\right)^2 + \mathrm{Tr}(\mathbf{MM}) + \mathrm{Tr}\left(\mathbf{M}^\top\mathbf{M}\right)\right].$$

We consider $\Lambda_{2,2}$ as

$$\mathbb{E}\Lambda_{2,2} = \underbrace{\mathbb{E}\left(\mathbf{X}_i^\top \mathbf{M X}_i \mathbf{X}_i^\top \mathbf{M X}_i\right)}_{\mathbb{E}\left(\mathbf{X}_i^\top \mathbf{M X}_i\right)^2} + \underbrace{\mathbb{E}\left(\mathbf{X}_j^\top \mathbf{MM X}_j \mathbf{X}_j^\top \mathbf{X}_j\right)}_{\mathbb{E}\|\mathbf{X}_j\|_2^2 \mathbf{X}_j^\top \mathbf{MM X}_j}$$

$$= \left(\mathrm{Tr}(\mathbf{M})\right)^2 + \mathrm{Tr}(\mathbf{MM}) + \mathrm{Tr}\left(\mathbf{M}^\top\mathbf{M}\right) + (p+2)\,\mathrm{Tr}\left(\mathbf{MM}\right).$$

As for $\Lambda_{2,3}$ and $\Lambda_{2,4}$, we can verify that they are both zero, which gives

$$\mathbb{E}\Lambda_2 = (n-h)p\left[1 + o(1)\right]\left[\mathrm{Tr}(\mathbf{M})\right]^2 + 3(n-h)p\left[1 + o(1)\right]\mathrm{Tr}(\mathbf{MM}). \tag{32}$$

**Case $(s,d)$: $i = \pi^\natural(i)$ and $j \neq \pi^\natural(j)$.** We can compute $\Lambda_{2,1}$ as

$$\mathbb{E}\Lambda_{2,1} = \mathbb{E}\mathbf{X}_i^\top \mathbf{M}\mathbf{X}_i \mathbf{X}_i^\top \mathbf{M}\mathbf{X}_i \mathbf{X}_i^\top \mathbf{X}_i + \underbrace{\mathbb{E}\mathbf{X}_{\pi^\natural(i)}^\top \mathbf{M}\mathbf{X}_{\pi^\natural(i)}\mathbf{X}_{\pi^\natural(i)}^\top \mathbf{M}\mathbf{X}_{\pi^\natural(j)}\mathbf{X}_j^\top \mathbf{X}_{\pi^\natural(i)}}_{0}$$

$$+ \underbrace{\mathbb{E}\mathbf{X}_{\pi^\natural(i)}^\top \mathbf{M}\mathbf{X}_{\pi^\natural(i)}\mathbf{X}_{\pi^\natural(i)}^\top \mathbf{M}\mathbf{X}_j\mathbf{X}_{\pi^{\natural-1}(j)}^\top \mathbf{X}_{\pi^\natural(i)}}_{0}$$

$$= (p+4)\left[(\mathrm{Tr}(\mathbf{M}))^2 + \mathrm{Tr}(\mathbf{M}\mathbf{M}) + \mathrm{Tr}\left(\mathbf{M}^\top \mathbf{M}\right)\right].$$

We consider $\Lambda_{2,2}$ as

$$\mathbb{E}\Lambda_{2,2} = \mathbb{E}\mathbf{X}_j^\top \mathbf{M}\mathbf{X}_i \mathbf{X}_i^\top \mathbf{M}\mathbf{X}_i \mathbf{X}_i^\top \mathbf{X}_j + \underbrace{\mathbb{E}\mathbf{X}_j^\top \mathbf{M}\mathbf{X}_{\pi^\natural(i)}\mathbf{X}_{\pi^\natural(i)}^\top \mathbf{M}\mathbf{X}_{\pi^\natural(j)}\mathbf{X}_j^\top \mathbf{X}_j}_{0}$$

$$+ \underbrace{\mathbb{E}\mathbf{X}_j^\top \mathbf{M}\mathbf{X}_{\pi^\natural(i)}\mathbf{X}_{\pi^\natural(i)}^\top \mathbf{M}\mathbf{X}_j\mathbf{X}_{\pi^{\natural-1}(j)}^\top \mathbf{X}_j}_{0}$$

$$= \mathbb{E}\left(\mathbf{X}_i^\top \mathbf{M}\mathbf{X}_i\right)^2 = (\mathrm{Tr}(\mathbf{M}))^2 + \mathrm{Tr}(\mathbf{M}\mathbf{M}) + \mathrm{Tr}\left(\mathbf{M}^\top \mathbf{M}\right).$$

Similarly, we can verify that both $\mathbb{E}\Lambda_{2,3}$ and $\mathbb{E}\Lambda_{2,4}$ are zero and hence have

$$\mathbb{E}\Lambda_2 = (n-h)p\left[1+o(1)\right]\left[\mathrm{Tr}(\mathbf{M})\right]^2 + 2(n-h)p\left[1+o(1)\right]\mathrm{Tr}(\mathbf{M}\mathbf{M}). \tag{33}$$

**Case $(d,s)$: $i \neq \pi^\natural(i)$ and $j = \pi^\natural(j)$.** We compute $\Lambda_{2,1}$ as

$$\mathbb{E}\Lambda_{2,1} = \underbrace{\mathbb{E}\mathbf{X}_{\pi^\natural(i)}^\top \mathbf{M}\mathbf{X}_{\pi^\natural(i)}\mathbf{X}_{\pi^\natural(i)}^\top \mathbf{M}\mathbf{X}_{\pi^\natural(i)}\mathbf{X}_i^\top \mathbf{X}_{\pi^\natural(i)}}_{0} + \underbrace{\mathbb{E}\mathbf{X}_{\pi^\natural(i)}^\top \mathbf{M}\mathbf{X}_{\pi^\natural(i)}\mathbf{X}_{\pi^\natural(i)}^\top \mathbf{M}\mathbf{X}_{\pi^{\natural 2}(i)}\mathbf{X}_{\pi^\natural(i)}^\top \mathbf{X}_{\pi^\natural(i)}}_{0}$$

$$+ \underbrace{\mathbb{E}\mathbf{X}_{\pi^\natural(i)}^\top \mathbf{M}\mathbf{X}_{\pi^\natural(i)}\mathbf{X}_{\pi^\natural(i)}^\top \mathbf{M}\mathbf{X}_j\mathbf{X}_j^\top \mathbf{X}_{\pi^\natural(i)}}_{\mathbb{E}\mathbf{X}_{\pi^\natural(i)}^\top \mathbf{M}\mathbf{X}_{\pi^\natural(i)}\mathbf{X}_{\pi^\natural(i)}^\top \mathbf{M}\mathbf{X}_{\pi^\natural(i)}} = (\mathrm{Tr}(\mathbf{M}))^2 + \mathrm{Tr}(\mathbf{M}\mathbf{M}) + \mathrm{Tr}\left(\mathbf{M}^\top \mathbf{M}\right).$$

We consider $\Lambda_{2,2}$ as

$$\mathbb{E}\Lambda_{2,2} = \underbrace{\mathbb{E}\mathbf{X}_j^\top \mathbf{M}\mathbf{X}_{\pi^\natural(i)}\mathbf{X}_{\pi^\natural(i)}^\top \mathbf{M}\mathbf{X}_{\pi^\natural(i)}\mathbf{X}_i^\top \mathbf{X}_j}_{0} + \underbrace{\mathbb{E}\mathbf{X}_j^\top \mathbf{M}\mathbf{X}_{\pi^\natural(i)}\mathbf{X}_{\pi^\natural(i)}^\top \mathbf{M}\mathbf{X}_{\pi^{\natural 2}(i)}\mathbf{X}_{\pi^\natural(i)}^\top \mathbf{X}_j}_{0}$$

$$+ \underbrace{\mathbb{E}\mathbf{X}_j^\top \mathbf{M}\mathbf{X}_{\pi^\natural(i)}\mathbf{X}_{\pi^\natural(i)}^\top \mathbf{M}\mathbf{X}_j\mathbf{X}_j^\top \mathbf{X}_j}_{\mathbb{E}\|\mathbf{X}_j\|_2^2\mathbf{X}_j^\top \mathbf{M}\mathbf{M}\mathbf{X}_j} = (p+2)\,\mathrm{Tr}(\mathbf{M}\mathbf{M}).$$

As for $\mathbb{E}\Lambda_{2,3}$ and $\Lambda_{2,4}$, we can follow the same strategy and prove they are both zero, which yields

$$\mathbb{E}\Lambda_2 = (n-h)\left[\mathrm{Tr}(\mathbf{M})\right]^2 + (n-h)p\left[1+o(1)\right]\mathrm{Tr}(\mathbf{M}\mathbf{M}). \tag{34}$$

**Case $(d,d)$: $i \neq \pi^\natural(i)$ and $j \neq \pi^\natural(j)$.** Contrary to the previous cases, we have $\mathbb{E}\Lambda_{2,1}$ and $\mathbb{E}\Lambda_{2,2}$ to be zero in this case rather than $\mathbb{E}\Lambda_{2,3}$ and $\mathbb{E}\Lambda_{2,4}$.

Hence our focus turns to the calculation of $\mathbb{E}\Lambda_{2,3}$ and that of $\mathbb{E}\Lambda_{2,4}$. For $\Lambda_{2,3}$, we have

$$\mathbb{E}\Lambda_{2,3} = \underbrace{\mathbb{E}\mathbf{X}_{\pi^\natural(i)}^\top \mathbf{M}\mathbf{X}_{\pi^\natural(i)}\mathbf{X}_{\pi^\natural(i)}^\top \mathbf{M}\mathbf{X}_{\pi^\natural(i)}\mathbf{X}_i^\top \mathbf{X}_j}_{p\mathbb{1}_{i=j}\mathbb{E}\mathbf{X}_{\pi^\natural(i)}^\top \mathbf{M}\mathbf{X}_{\pi^\natural(i)}\mathbf{X}_{\pi^\natural(i)}^\top \mathbf{M}\mathbf{X}_{\pi^\natural(i)}} + \underbrace{\mathbb{E}\mathbf{X}_{\pi^\natural(i)}^\top \mathbf{M}\mathbf{X}_{\pi^\natural(i)}\mathbf{X}_{\pi^\natural(i)}^\top \mathbf{M}\mathbf{X}_{\pi^{\natural 2}(i)}\mathbf{X}_{\pi^\natural(i)}^\top \mathbf{X}_j}_{\mathbb{1}_{j=\pi^{\natural 2}(i)}\mathbb{E}\mathbf{X}_{\pi^\natural(i)}^\top \mathbf{M}\mathbf{X}_{\pi^\natural(i)}\mathbf{X}_{\pi^\natural(i)}^\top \mathbf{M}\mathbf{X}_{\pi^\natural(i)}}$$

$$+ \underbrace{\mathbb{E}\mathbf{X}_{\pi^\natural(i)}^\top \mathbf{M}\mathbf{X}_{\pi^\natural(i)}\mathbf{X}_{\pi^\natural(i)}^\top \mathbf{M}\mathbf{X}_{\pi^\natural(j)}\mathbf{X}_j^\top \mathbf{X}_j}_{p\mathbb{1}_{i=j}\mathbb{E}\mathbf{X}_{\pi^\natural(i)}^\top \mathbf{M}\mathbf{X}_{\pi^\natural(i)}\mathbf{X}_{\pi^\natural(i)}^\top \mathbf{M}\mathbf{X}_{\pi^\natural(i)}} + \underbrace{\mathbb{E}\mathbf{X}_{\pi^\natural(i)}^\top \mathbf{M}\mathbf{X}_{\pi^\natural(i)}\mathbf{X}_{\pi^\natural(i)}^\top \mathbf{M}\mathbf{X}_j\mathbf{X}_{\pi^{\natural-1}(j)}^\top \mathbf{X}_j}_{\mathbb{1}_{j=\pi^{\natural 2}(i)}\mathbb{E}\mathbf{X}_{\pi^\natural(i)}^\top \mathbf{M}\mathbf{X}_{\pi^\natural(i)}\mathbf{X}_{\pi^\natural(i)}^\top \mathbf{M}\mathbf{X}_{\pi^\natural(i)}}$$

$$= 2\left(p\mathbb{1}_{i=j} + \mathbb{1}_{j=\pi^{\natural 2}(i)}\right)\left[(\mathrm{Tr}(\mathbf{M}))^2 + \mathrm{Tr}(\mathbf{M}\mathbf{M}) + \mathrm{Tr}\left(\mathbf{M}^\top \mathbf{M}\right)\right].$$

Then we turn to the calculation of $\mathbb{E}\Lambda_{2,4}$, which proceeds as

$$
\mathbb{E}\Lambda_{2,4} = \underbrace{\mathbb{E}\mathbf{X}_j^\top \mathbf{M}\mathbf{X}_{\pi^\natural(i)}\mathbf{X}_{\pi^\natural(i)}^\top \mathbf{M}\mathbf{X}_{\pi^\natural(i)}\mathbf{X}_i^\top \mathbf{X}_{\pi^\natural(i)}}_{\mathbb{1}(i=j)\mathbb{E}\mathbf{X}_{\pi^\natural(i)}^\top \mathbf{M}\mathbf{X}_{\pi^\natural(i)}\mathbf{X}_{\pi^\natural(i)}^\top \mathbf{M}\mathbf{X}_{\pi^\natural(i)}} + \underbrace{\mathbb{E}\mathbf{X}_j^\top \mathbf{M}\mathbf{X}_{\pi^\natural(i)}\mathbf{X}_{\pi^\natural(i)}^\top \mathbf{M}\mathbf{X}_{\pi^{\natural 2}(i)}\mathbf{X}_{\pi^\natural(i)}^\top \mathbf{X}_{\pi^\natural(i)}}_{\mathbb{1}_{j=\pi^{\natural 2}(i)}\mathbb{E}\left\|\mathbf{X}_{\pi^\natural(i)}\right\|_2^2 \mathbf{X}_{\pi^\natural(i)}^\top \mathbf{M}\mathbf{M}\mathbf{X}_{\pi^\natural(i)}}
$$

$$
+ \underbrace{\mathbb{E}\mathbf{X}_j^\top \mathbf{M}\mathbf{X}_{\pi^\natural(i)}\mathbf{X}_{\pi^\natural(i)}^\top \mathbf{M}\mathbf{X}_{\pi^\natural(j)}\mathbf{X}_j^\top \mathbf{X}_{\pi^\natural(i)}}_{\mathbb{1}(i=j)\mathbb{E}\mathbf{X}_{\pi^\natural(i)}^\top \mathbf{M}\mathbf{X}_{\pi^\natural(i)}\mathbf{X}_{\pi^\natural(i)}^\top \mathbf{M}\mathbf{X}_{\pi^\natural(i)}} + \underbrace{\mathbb{E}\mathbf{X}_j^\top \mathbf{M}\mathbf{X}_{\pi^\natural(i)}\mathbf{X}_{\pi^\natural(i)}^\top \mathbf{M}\mathbf{X}_j\mathbf{X}_{\pi^{\natural-1}(j)}^\top \mathbf{X}_{\pi^\natural(i)}}_{\mathbb{1}_{j=\pi^{\natural 2}(i)}\mathbb{E}\left\|\mathbf{X}_{\pi^\natural(i)}\right\|_2^2 \mathbf{X}_{\pi^\natural(i)}^\top \mathbf{M}\mathbf{M}\mathbf{X}_{\pi^\natural(i)}}
$$

$$
= 2\mathbb{1}_{j=\pi^{\natural 2}(i)}(p+2)\operatorname{Tr}(\mathbf{M}\mathbf{M}) + 2\mathbb{1}(i=j)\left[(\operatorname{Tr}(\mathbf{M}))^2 + \operatorname{Tr}(\mathbf{M}\mathbf{M}) + \operatorname{Tr}\left(\mathbf{M}^\top \mathbf{M}\right)\right].
$$

Then we conclude

$$
\mathbb{E}\Lambda_2 = -2(n-h)\left[(p+1)\mathbb{1}_{i=j} + \mathbb{1}_{j=\pi^{\natural 2}(i)}\right]\left[(\operatorname{Tr}(\mathbf{M}))^2 + \operatorname{Tr}(\mathbf{M}\mathbf{M}) + \operatorname{Tr}\left(\mathbf{M}^\top \mathbf{M}\right)\right]
$$
$$
-2(n-h)(p+2)\mathbb{1}_{j=\pi^{\natural 2}(i)}\operatorname{Tr}(\mathbf{M}\mathbf{M}). \tag{35}
$$

The proof is thus completed by combining (32), (33), (34), and (35).

$\square$

**Lemma 5.** *We have*

$$
\mathbb{E}\left(\mathbf{X}_{\pi^\natural(i)} - \mathbf{X}_j\right)^\top \mathbf{\Delta}\mathbf{M}\mathbf{X}_{\pi^\natural(i)}\mathbf{X}_{\pi^\natural(i)}^\top \mathbf{M}\mathbf{\Delta}^\top\left(\mathbf{X}_{\pi^\natural(i)} - \mathbf{X}_j\right)
$$
$$
= \left(1 - \frac{h}{n} + o(1)\right)p^2\left[\operatorname{Tr}(\mathbf{M})\right]^2 + \left(3 - \frac{h}{n} + o(1)\right)p^2\operatorname{Tr}(\mathbf{M}\mathbf{M}).
$$

*Proof.* We begin the proof by expanding $\Lambda_3$ as

$$
\mathbb{E}\Lambda_3 = \mathbb{E}\underbrace{\mathbf{X}_{\pi^\natural(i)}^\top \mathbf{\Delta}\mathbf{M}\mathbf{X}_{\pi^\natural(i)}\mathbf{X}_{\pi^\natural(i)}^\top \mathbf{M}\mathbf{\Delta}^\top \mathbf{X}_{\pi^\natural(i)}}_{\Lambda_{3,1}} + \mathbb{E}\underbrace{\mathbf{X}_j^\top \mathbf{\Delta}\mathbf{M}\mathbf{X}_{\pi^\natural(i)}\mathbf{X}_{\pi^\natural(i)}^\top \mathbf{M}\mathbf{\Delta}^\top \mathbf{X}_j}_{\Lambda_{3,2}}
$$
$$
- \mathbb{E}\underbrace{\mathbf{X}_{\pi^\natural(i)}^\top \mathbf{\Delta}\mathbf{M}\mathbf{X}_{\pi^\natural(i)}\mathbf{X}_{\pi^\natural(i)}^\top \mathbf{M}\mathbf{\Delta}^\top \mathbf{X}_j}_{\Lambda_{3,3}} - \mathbb{E}\underbrace{\mathbf{X}_j^\top \mathbf{\Delta}\mathbf{M}\mathbf{X}_{\pi^\natural(i)}\mathbf{X}_{\pi^\natural(i)}^\top \mathbf{M}\mathbf{\Delta}^\top \mathbf{X}_{\pi^\natural(i)}}_{\Lambda_{3,4}}.
$$

**Case $(s,s)$: $i = \pi^\natural(i)$ and $j = \pi^\natural(j)$.** First we compute $\Lambda_{3,1}$ as

$$
\mathbb{E}\Lambda_{3,1} = \underbrace{\mathbb{E}\mathbf{X}_i^\top \mathbf{X}_i\mathbf{X}_i^\top \mathbf{M}\mathbf{X}_i\mathbf{X}_i^\top \mathbf{M}\mathbf{X}_i\mathbf{X}_i^\top \mathbf{X}_i}_{\mathbb{E}\|\mathbf{X}_i\|_2^4\left(\mathbf{X}_i^\top \mathbf{M}\mathbf{X}_i\right)^2} + \underbrace{\mathbb{E}\mathbf{X}_i^\top \mathbf{X}_i\mathbf{X}_i^\top \mathbf{M}\mathbf{X}_i\mathbf{X}_i^\top \mathbf{M}\mathbf{X}_j\mathbf{X}_j^\top \mathbf{X}_i}_{\mathbb{E}\|\mathbf{X}_i\|_2^2\left(\mathbf{X}_i^\top \mathbf{M}\mathbf{X}_i\right)^2}
$$
$$
+ \underbrace{\mathbb{E}\mathbf{X}_i^\top \mathbf{X}_j\mathbf{X}_j^\top \mathbf{M}\mathbf{X}_i\mathbf{X}_i^\top \mathbf{M}\mathbf{X}_i\mathbf{X}_i^\top \mathbf{X}_i}_{\mathbb{E}\|\mathbf{X}_i\|_2^2\left(\mathbf{X}_i^\top \mathbf{M}\mathbf{X}_i\right)^2} + \underbrace{\mathbb{E}\mathbf{X}_i^\top \mathbf{X}_j\mathbf{X}_j^\top \mathbf{M}\mathbf{X}_i\mathbf{X}_i^\top \mathbf{M}\mathbf{X}_j\mathbf{X}_j^\top \mathbf{X}_i}_{\mathbb{E}\left(\mathbf{X}_j^\top \mathbf{X}_i\right)^2\mathbf{X}_j^\top \mathbf{M}\mathbf{X}_i\mathbf{X}_i^\top \mathbf{M}\mathbf{X}_j}
$$
$$
= (p+4)(p+8)\left[(\operatorname{Tr}(\mathbf{M}))^2 + 2\operatorname{Tr}(\mathbf{M}\mathbf{M})\right] + 2(\operatorname{Tr}(\mathbf{M}))^2 + (p+6)\operatorname{Tr}(\mathbf{M}\mathbf{M}).
$$

Then, we consider $\Lambda_{3,2}$ as

$$
\mathbb{E}\Lambda_{3,2} = \underbrace{\mathbb{E}\mathbf{X}_j^\top \mathbf{X}_i\mathbf{X}_i^\top \mathbf{M}\mathbf{X}_i\mathbf{X}_i^\top \mathbf{M}\mathbf{X}_i\mathbf{X}_i^\top \mathbf{X}_j}_{\mathbb{E}\|\mathbf{X}_i\|_2^2\left(\mathbf{X}_i^\top \mathbf{M}\mathbf{X}_i\right)^2} + \underbrace{\mathbb{E}\mathbf{X}_j^\top \mathbf{X}_i\mathbf{X}_i^\top \mathbf{M}\mathbf{X}_i\mathbf{X}_i^\top \mathbf{M}\mathbf{X}_j\mathbf{X}_j^\top \mathbf{X}_j}_{(p+2)\mathbb{E}\left(\mathbf{X}_i^\top \mathbf{M}\mathbf{X}_i\right)^2}
$$
$$
+ \underbrace{\mathbb{E}\mathbf{X}_j^\top \mathbf{X}_j\mathbf{X}_j^\top \mathbf{M}\mathbf{X}_i\mathbf{X}_i^\top \mathbf{M}\mathbf{X}_i\mathbf{X}_i^\top \mathbf{X}_j}_{(p+2)\mathbb{E}\left(\mathbf{X}_i^\top \mathbf{M}\mathbf{X}_i\right)^2} + \underbrace{\mathbb{E}\mathbf{X}_j^\top \mathbf{X}_j\mathbf{X}_j^\top \mathbf{M}\mathbf{X}_i\mathbf{X}_i^\top \mathbf{M}\mathbf{X}_j\mathbf{X}_j^\top \mathbf{X}_j}_{\mathbb{E}\|\mathbf{X}_j\|_2^4\mathbf{X}_j^\top \mathbf{M}\mathbf{M}\mathbf{X}_j}
$$
$$
= (3p+8)\left[(\operatorname{Tr}(\mathbf{M}))^2 + \operatorname{Tr}(\mathbf{M}\mathbf{M}) + \operatorname{Tr}\left(\mathbf{M}^\top \mathbf{M}\right)\right] + (p+2)(p+4)\operatorname{Tr}(\mathbf{M}\mathbf{M}).
$$

In addition, we can verify that $\mathbb{E}\Lambda_{3,3}$ and $\mathbb{E}\Lambda_{3,4}$ are both zero. Hence we conclude

$$
\mathbb{E}\Lambda_3 = \left(p^2 + 15p + 42\right)\left[\operatorname{Tr}(\mathbf{M})\right]^2 + \left(3p^2 + 37p + 94\right)\operatorname{Tr}(\mathbf{M}\mathbf{M}). \tag{36}
$$

**Case $(s,d)$: $i = \pi^\natural(i)$ and $j \neq \pi^\natural(j)$.** We can compute $\Lambda_{3,1}$ as

$$\mathbb{E}\Lambda_{3,1} = \underbrace{\mathbb{E}\mathbf{X}_i^\top\mathbf{X}_i\mathbf{X}_i^\top\mathbf{M}\mathbf{X}_i\mathbf{X}_i^\top\mathbf{M}\mathbf{X}_i\mathbf{X}_i^\top\mathbf{X}_i}_{\mathbb{E}\|\mathbf{X}_i\|_2^4(\mathbf{X}_i^\top\mathbf{M}\mathbf{X}_i)^2} + \underbrace{\mathbb{E}\mathbf{X}_i^\top\mathbf{X}_i\mathbf{X}_i^\top\mathbf{M}\mathbf{X}_i\mathbf{X}_i^\top\mathbf{M}\mathbf{X}_{\pi^\natural(j)}\mathbf{X}_j^\top\mathbf{X}_i}_{0}$$

$$+ \underbrace{\mathbb{E}\mathbf{X}_i^\top\mathbf{X}_i\mathbf{X}_i^\top\mathbf{M}\mathbf{X}_i\mathbf{X}_i^\top\mathbf{M}\mathbf{X}_j\mathbf{X}_{\pi^{\natural-1}(j)}^\top\mathbf{X}_i}_{0} + \underbrace{\mathbb{E}\mathbf{X}_i^\top\mathbf{X}_j\mathbf{X}_{\pi^\natural(j)}^\top\mathbf{M}\mathbf{X}_i\mathbf{X}_i^\top\mathbf{M}\mathbf{X}_i\mathbf{X}_i^\top\mathbf{X}_i}_{0}$$

$$+ \underbrace{\mathbb{E}\mathbf{X}_i^\top\mathbf{X}_j\mathbf{X}_{\pi^\natural(j)}^\top\mathbf{M}\mathbf{X}_i\mathbf{X}_i^\top\mathbf{M}\mathbf{X}_{\pi^\natural(j)}\mathbf{X}_j^\top\mathbf{X}_i}_{\mathbb{E}\|\mathbf{X}_i\|_2^2\mathbf{X}_i^\top\mathbf{M}\mathbf{M}\mathbf{X}_i} + \underbrace{\mathbb{E}\mathbf{X}_i^\top\mathbf{X}_j\mathbf{X}_{\pi^\natural(j)}^\top\mathbf{M}\mathbf{X}_i\mathbf{X}_i^\top\mathbf{M}\mathbf{X}_j\mathbf{X}_{\pi^{\natural-1}(j)}^\top\mathbf{X}_i}_{\mathbb{1}_{j=\pi^{\natural2}(j)}\mathbb{E}(\mathbf{X}_i^\top\mathbf{M}\mathbf{X}_i)^2}$$

$$+ \underbrace{\mathbb{E}\mathbf{X}_i^\top\mathbf{X}_{\pi^{\natural-1}(j)}\mathbf{X}_j^\top\mathbf{M}\mathbf{X}_i\mathbf{X}_i^\top\mathbf{M}\mathbf{X}_i\mathbf{X}_i^\top\mathbf{X}_i}_{0} + \underbrace{\mathbb{E}\mathbf{X}_i^\top\mathbf{X}_{\pi^{\natural-1}(j)}\mathbf{X}_j^\top\mathbf{M}\mathbf{X}_i\mathbf{X}_i^\top\mathbf{M}\mathbf{X}_{\pi^\natural(j)}\mathbf{X}_j^\top\mathbf{X}_i}_{\mathbb{1}_{j=\pi^{\natural2}(j)}\mathbb{E}(\mathbf{X}_i^\top\mathbf{M}\mathbf{X}_i)^2}$$

$$+ \underbrace{\mathbb{E}\mathbf{X}_i^\top\mathbf{X}_{\pi^{\natural-1}(j)}\mathbf{X}_j^\top\mathbf{M}\mathbf{X}_i\mathbf{X}_i^\top\mathbf{M}\mathbf{X}_j\mathbf{X}_{\pi^{\natural-1}(j)}^\top\mathbf{X}_i}_{\mathbb{E}\|\mathbf{X}_i\|_2^2\mathbf{X}_i^\top\mathbf{M}\mathbf{M}\mathbf{X}_i}$$

$$= \left(p^2 + 10p + 24 + 2\mathbb{1}_{j=\pi^{\natural2}(j)}\right)\left[(\mathrm{Tr}(\mathbf{M}))^2 + 2\,\mathrm{Tr}(\mathbf{M}\mathbf{M})\right] + 2(p+2)\,\mathrm{Tr}(\mathbf{M}\mathbf{M}).$$

We consider $\Lambda_{3,2}$ as

$$\mathbb{E}\Lambda_{3,2} = \underbrace{\mathbb{E}\mathbf{X}_j^\top\mathbf{X}_i\mathbf{X}_i^\top\mathbf{M}\mathbf{X}_i\mathbf{X}_i^\top\mathbf{M}\mathbf{X}_i\mathbf{X}_i^\top\mathbf{X}_j}_{\mathbb{E}\|\mathbf{X}_i\|_2^2(\mathbf{X}_i^\top\mathbf{M}\mathbf{X}_i)^2} + \underbrace{\mathbb{E}\mathbf{X}_j^\top\mathbf{X}_i\mathbf{X}_i^\top\mathbf{M}\mathbf{X}_i\mathbf{X}_i^\top\mathbf{M}\mathbf{X}_{\pi^\natural(j)}\mathbf{X}_j^\top\mathbf{X}_j}_{0}$$

$$+ \underbrace{\mathbb{E}\mathbf{X}_j^\top\mathbf{X}_i\mathbf{X}_i^\top\mathbf{M}\mathbf{X}_i\mathbf{X}_i^\top\mathbf{M}\mathbf{X}_j\mathbf{X}_{\pi^{\natural-1}(j)}^\top\mathbf{X}_j}_{0}$$

$$+ \underbrace{\mathbb{E}\mathbf{X}_j^\top\mathbf{X}_j\mathbf{X}_{\pi^\natural(j)}^\top\mathbf{M}\mathbf{X}_i\mathbf{X}_i^\top\mathbf{M}\mathbf{X}_i\mathbf{X}_i^\top\mathbf{X}_j}_{0} + \underbrace{\mathbb{E}\mathbf{X}_j^\top\mathbf{X}_j\mathbf{X}_{\pi^\natural(j)}^\top\mathbf{M}\mathbf{X}_i\mathbf{X}_i^\top\mathbf{M}\mathbf{X}_{\pi^\natural(j)}\mathbf{X}_j^\top\mathbf{X}_j}_{\mathbb{E}\|\mathbf{X}_i\|_2^4\,\mathrm{Tr}(\mathbf{M}\mathbf{M})}$$

$$+ \underbrace{\mathbb{E}\mathbf{X}_j^\top\mathbf{X}_j\mathbf{X}_{\pi^\natural(j)}^\top\mathbf{M}\mathbf{X}_i\mathbf{X}_i^\top\mathbf{M}\mathbf{X}_j\mathbf{X}_{\pi^{\natural-1}(j)}^\top\mathbf{X}_j}_{\mathbb{1}_{j=\pi^{\natural2}(j)}\mathbb{E}\|\mathbf{X}_j\|_2^2\mathbf{X}_j^\top\mathbf{M}\mathbf{M}\mathbf{X}_j} + \underbrace{\mathbb{E}\mathbf{X}_j^\top\mathbf{X}_{\pi^{\natural-1}(j)}\mathbf{X}_j^\top\mathbf{M}\mathbf{X}_i\mathbf{X}_i^\top\mathbf{M}\mathbf{X}_i\mathbf{X}_i^\top\mathbf{X}_j}_{0}$$

$$+ \underbrace{\mathbb{E}\mathbf{X}_j^\top\mathbf{X}_{\pi^{\natural-1}(j)}\mathbf{X}_j^\top\mathbf{M}\mathbf{X}_i\mathbf{X}_i^\top\mathbf{M}\mathbf{X}_{\pi^\natural(j)}\mathbf{X}_j^\top\mathbf{X}_j}_{\mathbb{1}_{j=\pi^{\natural2}(j)}\mathbb{E}\|\mathbf{X}_j\|_2^2\mathbf{X}_j^\top\mathbf{M}\mathbf{M}\mathbf{X}_j} + \underbrace{\mathbb{E}\mathbf{X}_j^\top\mathbf{X}_{\pi^{\natural-1}(j)}\mathbf{X}_j^\top\mathbf{M}\mathbf{X}_i\mathbf{X}_i^\top\mathbf{M}\mathbf{X}_j\mathbf{X}_{\pi^{\natural-1}(j)}^\top\mathbf{X}_j}_{\mathbb{E}\|\mathbf{X}_j\|_2^2\mathbf{X}_j^\top\mathbf{M}\mathbf{M}\mathbf{X}_j}$$

$$= (p+4)\left[(\mathrm{Tr}(\mathbf{M}))^2 + 2\,\mathrm{Tr}(\mathbf{M}\mathbf{M})\right] + (p+2)\left(p+1+2\mathbb{1}_{j=\pi^{\natural2}(j)}\right)\mathrm{Tr}(\mathbf{M}\mathbf{M}).$$

As for $\Lambda_{3,3}$ and $\Lambda_{3,4}$, we can prove that they are both zero in this case. Hence we conclude

$$\mathbb{E}\Lambda_3 = \left(p^2 + 11p + 28 + 2\mathbb{1}_{j=\pi^{\natural2}(j)}\right)(\mathrm{Tr}(\mathbf{M}))^2$$
$$+ \left[3p^2 + 27p + 62 + 2(p+4)\,\mathbb{1}_{j=\pi^{\natural2}(j)}\right]\mathrm{Tr}(\mathbf{M}\mathbf{M}). \tag{37}$$

**Case $(d,s)$: $i \neq \pi^\natural(i)$ and $j = \pi^\natural(j)$.** We consider the term $\Lambda_{3,1}$ as

$$\mathbb{E}\Lambda_{3,1} = \underbrace{\mathbb{E}\mathbf{X}_{\pi^\natural(i)}^\top\mathbf{X}_i\mathbf{X}_{\pi^\natural(i)}^\top\mathbf{M}\mathbf{X}_{\pi^\natural(i)}\mathbf{X}_{\pi^\natural(i)}^\top\mathbf{M}\mathbf{X}_{\pi^\natural(i)}\mathbf{X}_i^\top\mathbf{X}_{\pi^\natural(i)}}_{\mathbb{E}\left\|\mathbf{X}_{\pi^\natural(i)}\right\|_F^2\left(\mathbf{X}_{\pi^\natural(i)}^\top\mathbf{M}\mathbf{X}_{\pi^\natural(i)}\right)^2}$$

$$+ \underbrace{\mathbb{E}\mathbf{X}_{\pi^\natural(i)}^\top\mathbf{X}_i\mathbf{X}_{\pi^\natural(i)}^\top\mathbf{M}\mathbf{X}_{\pi^\natural(i)}\mathbf{X}_{\pi^\natural(i)}^\top\mathbf{M}\mathbf{X}_{\pi^{\natural2}(i)}\mathbf{X}_{\pi^\natural(i)}^\top\mathbf{X}_{\pi^\natural(i)}}_{\mathbb{1}_{i=\pi^{\natural2}(i)}\mathbb{E}\left\|\mathbf{X}_{\pi^\natural(i)}\right\|_F^2\left(\mathbf{X}_{\pi^\natural(i)}^\top\mathbf{M}\mathbf{X}_{\pi^\natural(i)}\right)^2}$$

$$+ \underbrace{\mathbb{E}\mathbf{X}_{\pi^\natural(i)}^\top\mathbf{X}_i\mathbf{X}_{\pi^\natural(i)}^\top\mathbf{M}\mathbf{X}_{\pi^\natural(i)}\mathbf{X}_{\pi^\natural(i)}^\top\mathbf{M}\mathbf{X}_j\mathbf{X}_j^\top\mathbf{X}_{\pi^\natural(i)}}_{0}$$

$$+ \underbrace{\mathbb{E}\mathbf{X}_{\pi^\natural(i)}^\top\mathbf{X}_{\pi^\natural(i)}\mathbf{X}_{\pi^{\natural2}(i)}^\top\mathbf{M}\mathbf{X}_{\pi^\natural(i)}\mathbf{X}_{\pi^\natural(i)}^\top\mathbf{M}\mathbf{X}_{\pi^\natural(i)}\mathbf{X}_i^\top\mathbf{X}_{\pi^\natural(i)}}_{\mathbb{1}_{i=\pi^{\natural2}(i)}\mathbb{E}\left\|\mathbf{X}_{\pi^\natural(i)}\right\|_F^2\left(\mathbf{X}_{\pi^\natural(i)}^\top\mathbf{M}\mathbf{X}_{\pi^\natural(i)}\right)^2}$$

$$+ \underbrace{\mathbb{E}\mathbf{X}_{\pi^\natural(i)}^\top \mathbf{X}_{\pi^\natural(i)} \mathbf{X}_{\pi^{\natural 2}(i)}^\top \mathbf{M}\mathbf{X}_{\pi^\natural(i)} \mathbf{X}_{\pi^\natural(i)}^\top \mathbf{M}\mathbf{X}_{\pi^{\natural 2}(i)} \mathbf{X}_{\pi^\natural(i)}^\top \mathbf{X}_{\pi^\natural(i)}}_{\mathbb{E}\|\mathbf{X}_i\|_2^4 \mathbf{X}_i^\top \mathbf{M}\mathbf{M}\mathbf{X}_i}$$

$$+ \underbrace{\mathbb{E}\mathbf{X}_{\pi^\natural(i)}^\top \mathbf{X}_{\pi^\natural(i)} \mathbf{X}_{\pi^{\natural 2}(i)}^\top \mathbf{M}\mathbf{X}_{\pi^\natural(i)} \mathbf{X}_{\pi^\natural(i)}^\top \mathbf{M}\mathbf{X}_j \mathbf{X}_j^\top \mathbf{X}_{\pi^\natural(i)}}_{0}$$

$$+ \underbrace{\mathbb{E}\mathbf{X}_{\pi^\natural(i)}^\top \mathbf{X}_j \mathbf{X}_j^\top \mathbf{M}\mathbf{X}_{\pi^\natural(i)} \mathbf{X}_{\pi^\natural(i)}^\top \mathbf{M}\mathbf{X}_{\pi^\natural(i)} \mathbf{X}_i^\top \mathbf{X}_{\pi^\natural(i)}}_{0}$$

$$+ \underbrace{\mathbb{E}\mathbf{X}_{\pi^\natural(i)}^\top \mathbf{X}_j \mathbf{X}_j^\top \mathbf{M}\mathbf{X}_{\pi^\natural(i)} \mathbf{X}_{\pi^\natural(i)}^\top \mathbf{M}\mathbf{X}_{\pi^{\natural 2}(i)} \mathbf{X}_{\pi^\natural(i)}^\top \mathbf{X}_{\pi^\natural(i)}}_{0}$$

$$+ \underbrace{\mathbb{E}\mathbf{X}_{\pi^\natural(i)}^\top \mathbf{X}_j \mathbf{X}_j^\top \mathbf{M}\mathbf{X}_{\pi^\natural(i)} \mathbf{X}_{\pi^\natural(i)}^\top \mathbf{M}\mathbf{X}_j \mathbf{X}_j^\top \mathbf{X}_{\pi^\natural(i)}}_{\mathbb{E}(\mathbf{X}_i^\top \mathbf{X}_j)^2 \mathbf{X}_j^\top \mathbf{M}\mathbf{X}_i \mathbf{X}_i^\top \mathbf{M}\mathbf{X}_j}$$

$$= (p+6)(\operatorname{Tr}(\mathbf{M}))^2 + (p^2 + 9p + 22)\operatorname{Tr}(\mathbf{M}\mathbf{M}) + 2\mathbb{1}_{i=\pi^{\natural 2}(i)}(p+4)\left[(\operatorname{Tr}(\mathbf{M}))^2 + 2\operatorname{Tr}(\mathbf{M}\mathbf{M})\right].$$

We consider the term $\Lambda_{3,2}$ as

$$\mathbb{E}\Lambda_{3,2} = \underbrace{\mathbb{E}\mathbf{X}_j^\top \mathbf{X}_i \mathbf{X}_{\pi^\natural(i)}^\top \mathbf{M}\mathbf{X}_{\pi^\natural(i)} \mathbf{X}_{\pi^\natural(i)}^\top \mathbf{M}\mathbf{X}_{\pi^\natural(i)} \mathbf{X}_i^\top \mathbf{X}_j}_{p \times \mathbb{E}\left(\mathbf{X}_{\pi^\natural(i)}^\top \mathbf{M}\mathbf{X}_{\pi^\natural(i)}\right)^2}$$

$$+ \underbrace{\mathbb{E}\mathbf{X}_j^\top \mathbf{X}_i \mathbf{X}_{\pi^\natural(i)}^\top \mathbf{M}\mathbf{X}_{\pi^\natural(i)} \mathbf{X}_{\pi^\natural(i)}^\top \mathbf{M}\mathbf{X}_{\pi^{\natural 2}(i)} \mathbf{X}_{\pi^\natural(i)}^\top \mathbf{X}_j}_{\mathbb{1}_{i=\pi^{\natural 2}(i)} \mathbb{E}\left(\mathbf{X}_i^\top \mathbf{M}\mathbf{X}_i\right)^2}$$

$$+ \underbrace{\mathbb{E}\mathbf{X}_j^\top \mathbf{X}_i \mathbf{X}_{\pi^\natural(i)}^\top \mathbf{M}\mathbf{X}_{\pi^\natural(i)} \mathbf{X}_{\pi^\natural(i)}^\top \mathbf{M}\mathbf{X}_j \mathbf{X}_j^\top \mathbf{X}_j}_{0}$$

$$+ \underbrace{\mathbb{E}\mathbf{X}_j^\top \mathbf{X}_{\pi^\natural(i)} \mathbf{X}_{\pi^{\natural 2}(i)}^\top \mathbf{M}\mathbf{X}_{\pi^\natural(i)} \mathbf{X}_{\pi^\natural(i)}^\top \mathbf{M}\mathbf{X}_{\pi^\natural(i)} \mathbf{X}_i^\top \mathbf{X}_j}_{\mathbb{1}_{i=\pi^{\natural 2}(i)} \mathbb{E}\left(\mathbf{X}_i^\top \mathbf{M}\mathbf{X}_i\right)^2}$$

$$+ \underbrace{\mathbb{E}\mathbf{X}_j^\top \mathbf{X}_{\pi^\natural(i)} \mathbf{X}_{\pi^{\natural 2}(i)}^\top \mathbf{M}\mathbf{X}_{\pi^\natural(i)} \mathbf{X}_{\pi^\natural(i)}^\top \mathbf{M}\mathbf{X}_{\pi^{\natural 2}(i)} \mathbf{X}_{\pi^\natural(i)}^\top \mathbf{X}_j}_{\mathbb{E}\|\mathbf{X}_i\|_2^2 \mathbf{X}_i^\top \mathbf{M}\mathbf{M}\mathbf{X}_i}$$

$$+ \underbrace{\mathbb{E}\mathbf{X}_j^\top \mathbf{X}_{\pi^\natural(i)} \mathbf{X}_{\pi^{\natural 2}(i)}^\top \mathbf{M}\mathbf{X}_{\pi^\natural(i)} \mathbf{X}_{\pi^\natural(i)}^\top \mathbf{M}\mathbf{X}_j \mathbf{X}_j^\top \mathbf{X}_j}_{0}$$

$$+ \underbrace{\mathbb{E}\mathbf{X}_j^\top \mathbf{X}_j \mathbf{X}_j^\top \mathbf{M}\mathbf{X}_{\pi^\natural(i)} \mathbf{X}_{\pi^\natural(i)}^\top \mathbf{M}\mathbf{X}_{\pi^\natural(i)} \mathbf{X}_i^\top \mathbf{X}_j}_{0}$$

$$+ \underbrace{\mathbb{E}\mathbf{X}_j^\top \mathbf{X}_j \mathbf{X}_j^\top \mathbf{M}\mathbf{X}_{\pi^\natural(i)} \mathbf{X}_{\pi^\natural(i)}^\top \mathbf{M}\mathbf{X}_{\pi^{\natural 2}(i)} \mathbf{X}_{\pi^\natural(i)}^\top \mathbf{X}_j}_{0}$$

$$+ \underbrace{\mathbb{E}\mathbf{X}_j^\top \mathbf{X}_j \mathbf{X}_j^\top \mathbf{M}\mathbf{X}_{\pi^\natural(i)} \mathbf{X}_{\pi^\natural(i)}^\top \mathbf{M}\mathbf{X}_j \mathbf{X}_j^\top \mathbf{X}_j}_{\mathbb{E}\|\mathbf{X}_j\|_F^4 \times \mathbf{X}_j^\top \mathbf{M}\mathbf{M}\mathbf{X}_j}$$

$$= p(\operatorname{Tr}(\mathbf{M}))^2 + (p^2 + 9p + 10)\operatorname{Tr}(\mathbf{M}\mathbf{M}) + 2\mathbb{1}_{i=\pi^{\natural 2}(i)}\left[(\operatorname{Tr}(\mathbf{M}))^2 + 2\operatorname{Tr}(\mathbf{M}\mathbf{M})\right].$$

As for $\Lambda_{3,3}$ and $\Lambda_{3,4}$, easily we can verify they are both zero and hence

$$\mathbb{E}\Lambda_3 = 2(p+3)(\operatorname{Tr}(\mathbf{M}))^2 + 2(p^2 + 9p + 16)\operatorname{Tr}(\mathbf{M}\mathbf{M})$$
$$+ 2\mathbb{1}_{i=\pi^{\natural 2}(i)}(p+5)\left[(\operatorname{Tr}(\mathbf{M}))^2 + 2\operatorname{Tr}(\mathbf{M}\mathbf{M})\right]. \tag{38}$$

**Case $(d,d)$: $i \neq \pi^\natural(i)$ and $j \neq \pi^\natural(j)$.** First, We compute $\mathbb{E}\Lambda_{3,1}$ as

$$\mathbb{E}\Lambda_{3,1} = \underbrace{\mathbb{E}\mathbf{X}_{\pi^\natural(i)}^\top \mathbf{X}_i \mathbf{X}_{\pi^\natural(i)}^\top \mathbf{M}\mathbf{X}_{\pi^\natural(i)} \mathbf{X}_{\pi^\natural(i)}^\top \mathbf{M}\mathbf{X}_{\pi^\natural(i)} \mathbf{X}_i^\top \mathbf{X}_{\pi^\natural(i)}}_{\mathbb{E}\left\|\mathbf{X}_{\pi^\natural(i)}\right\|_F^2 \left(\mathbf{X}_{\pi^\natural(i)}^\top \mathbf{M}\mathbf{X}_{\pi^\natural(i)}\right)^2}$$

$$+ \underbrace{\mathbb{E}\mathbf{X}_{\pi^\natural(i)}^\top \mathbf{X}_i \mathbf{X}_{\pi^\natural(i)}^\top \mathbf{M} \mathbf{X}_{\pi^\natural(i)} \mathbf{X}_{\pi^\natural(i)}^\top \mathbf{M} \mathbf{X}_{\pi^{\natural 2}(i)} \mathbf{X}_{\pi^\natural(i)}^\top \mathbf{X}_{\pi^\natural(i)}}_{\mathbb{1}_{i=\pi^{\natural 2}(i)} \mathbb{E}\left\|\mathbf{X}_{\pi^\natural(i)}\right\|_{\mathrm{F}}^2 \left(\mathbf{X}_{\pi^\natural(i)}^\top \mathbf{M} \mathbf{X}_{\pi^\natural(i)}\right)^2}$$

$$+ \underbrace{\mathbb{E}\mathbf{X}_{\pi^\natural(i)}^\top \mathbf{X}_i \mathbf{X}_{\pi^\natural(i)}^\top \mathbf{M} \mathbf{X}_{\pi^\natural(i)} \mathbf{X}_{\pi^\natural(i)}^\top \mathbf{M} \mathbf{X}_{\pi^\natural(j)} \mathbf{X}_j^\top \mathbf{X}_{\pi^\natural(i)}}_{\mathbb{1}_{i=j} \mathbb{E}\left\|\mathbf{X}_{\pi^\natural(i)}\right\|_{\mathrm{F}}^2 \left(\mathbf{X}_{\pi^\natural(i)}^\top \mathbf{M} \mathbf{X}_{\pi^\natural(i)}\right)^2}$$

$$+ \underbrace{\mathbb{E}\mathbf{X}_{\pi^\natural(i)}^\top \mathbf{X}_i \mathbf{X}_{\pi^\natural(i)}^\top \mathbf{M} \mathbf{X}_{\pi^\natural(i)} \mathbf{X}_{\pi^\natural(i)}^\top \mathbf{M} \mathbf{X}_j \mathbf{X}_{\pi^{\natural-1}(j)}^\top \mathbf{X}_{\pi^\natural(i)}}_{\mathbb{1}_{j=i} \mathbb{1}_{j=\pi^{\natural 2}(i)} \mathbb{E}\left\|\mathbf{X}_{\pi^\natural(i)}\right\|_{\mathrm{F}}^2 \left(\mathbf{X}_{\pi^\natural(i)}^\top \mathbf{M} \mathbf{X}_{\pi^\natural(i)}\right)^2}$$

$$+ \underbrace{\mathbb{E}\mathbf{X}_{\pi^\natural(i)}^\top \mathbf{X}_{\pi^\natural(i)} \mathbf{X}_{\pi^{\natural 2}(i)}^\top \mathbf{M} \mathbf{X}_{\pi^\natural(i)} \mathbf{X}_{\pi^\natural(i)}^\top \mathbf{M} \mathbf{X}_{\pi^\natural(i)} \mathbf{X}_i^\top \mathbf{X}_{\pi^\natural(i)}}_{\mathbb{1}_{i=\pi^{\natural 2}(i)} \mathbb{E}\left\|\mathbf{X}_{\pi^\natural(i)}\right\|_{\mathrm{F}}^2 \left(\mathbf{X}_{\pi^\natural(i)}^\top \mathbf{M} \mathbf{X}_{\pi^\natural(i)}\right)^2}$$

$$+ \underbrace{\mathbb{E}\mathbf{X}_{\pi^\natural(i)}^\top \mathbf{X}_{\pi^\natural(i)} \mathbf{X}_{\pi^{\natural 2}(i)}^\top \mathbf{M} \mathbf{X}_{\pi^\natural(i)} \mathbf{X}_{\pi^\natural(i)}^\top \mathbf{M} \mathbf{X}_{\pi^{\natural 2}(i)} \mathbf{X}_{\pi^\natural(i)}^\top \mathbf{X}_{\pi^\natural(i)}}_{\mathbb{E}\left\|\mathbf{X}_{\pi^\natural(i)}\right\|_{\mathrm{F}}^4 \left(\mathbf{X}_{\pi^\natural(i)}^\top \mathbf{M} \mathbf{M} \mathbf{X}_{\pi^\natural(i)}\right)}$$

$$+ \underbrace{\mathbb{E}\mathbf{X}_{\pi^\natural(i)}^\top \mathbf{X}_{\pi^\natural(i)} \mathbf{X}_{\pi^{\natural 2}(i)}^\top \mathbf{M} \mathbf{X}_{\pi^\natural(i)} \mathbf{X}_{\pi^\natural(i)}^\top \mathbf{M} \mathbf{X}_{\pi^\natural(j)} \mathbf{X}_j^\top \mathbf{X}_{\pi^\natural(i)}}_{\mathbb{1}_{j=i} \mathbb{1}_{j=\pi^{\natural 2}(i)} \mathbb{E}\left\|\mathbf{X}_{\pi^\natural(i)}\right\|_{\mathrm{F}}^2 \left(\mathbf{X}_{\pi^\natural(i)}^\top \mathbf{M} \mathbf{X}_{\pi^\natural(i)}\right)^2}$$

$$+ \underbrace{\mathbb{E}\mathbf{X}_{\pi^\natural(i)}^\top \mathbf{X}_{\pi^\natural(i)} \mathbf{X}_{\pi^{\natural 2}(i)}^\top \mathbf{M} \mathbf{X}_{\pi^\natural(i)} \mathbf{X}_{\pi^\natural(i)}^\top \mathbf{M} \mathbf{X}_j \mathbf{X}_{\pi^{\natural-1}(j)}^\top \mathbf{X}_{\pi^\natural(i)}}_{\mathbb{1}_{j=\pi^{\natural 2}(i)} \mathbb{E}\left\|\mathbf{X}_{\pi^\natural(i)}\right\|_{\mathrm{F}}^4 \left(\mathbf{X}_{\pi^\natural(i)}^\top \mathbf{M} \mathbf{M} \mathbf{X}_{\pi^\natural(i)}\right)}$$

$$+ \underbrace{\mathbb{E}\mathbf{X}_{\pi^\natural(i)}^\top \mathbf{X}_j \mathbf{X}_{\pi^\natural(j)}^\top \mathbf{M} \mathbf{X}_{\pi^\natural(i)} \mathbf{X}_{\pi^\natural(i)}^\top \mathbf{M} \mathbf{X}_{\pi^\natural(i)} \mathbf{X}_i^\top \mathbf{X}_{\pi^\natural(i)}}_{\mathbb{1}_{i=j} \mathbb{E}\left\|\mathbf{X}_{\pi^\natural(i)}\right\|_{\mathrm{F}}^2 \left(\mathbf{X}_{\pi^\natural(i)}^\top \mathbf{M} \mathbf{X}_{\pi^\natural(i)}\right)^2}$$

$$+ \underbrace{\mathbb{E}\mathbf{X}_{\pi^\natural(i)}^\top \mathbf{X}_j \mathbf{X}_{\pi^\natural(j)}^\top \mathbf{M} \mathbf{X}_{\pi^\natural(i)} \mathbf{X}_{\pi^\natural(i)}^\top \mathbf{M} \mathbf{X}_{\pi^{\natural 2}(i)} \mathbf{X}_{\pi^\natural(i)}^\top \mathbf{X}_{\pi^\natural(i)}}_{\mathbb{1}_{j=i} \mathbb{1}_{j=\pi^{\natural 2}(i)} \mathbb{E}\left\|\mathbf{X}_{\pi^\natural(i)}\right\|_{\mathrm{F}}^2 \left(\mathbf{X}_{\pi^\natural(i)}^\top \mathbf{M} \mathbf{X}_{\pi^\natural(i)}\right)^2}$$

$$+ \underbrace{\mathbb{E}\mathbf{X}_{\pi^\natural(i)}^\top \mathbf{X}_j \mathbf{X}_{\pi^\natural(j)}^\top \mathbf{M} \mathbf{X}_{\pi^\natural(i)} \mathbf{X}_{\pi^\natural(i)}^\top \mathbf{M} \mathbf{X}_{\pi^\natural(j)} \mathbf{X}_j^\top \mathbf{X}_{\pi^\natural(i)}}_{\mathbb{1}_{i=j} \mathbb{E}\left\|\mathbf{X}_{\pi^\natural(i)}\right\|_{\mathrm{F}}^2 \left(\mathbf{X}_{\pi^\natural(i)}^\top \mathbf{M} \mathbf{X}_{\pi^\natural(i)}\right)^2 + \mathbb{1}_{i\neq j} \mathbb{E}\left\|\mathbf{X}_{\pi^\natural(i)}\right\|_{\mathrm{F}}^2 \mathbf{X}_{\pi^\natural(i)}^\top \mathbf{M} \mathbf{M} \mathbf{X}_{\pi^\natural(i)}}$$

$$+ \underbrace{\mathbb{E}\mathbf{X}_{\pi^\natural(i)}^\top \mathbf{X}_j \mathbf{X}_{\pi^\natural(j)}^\top \mathbf{M} \mathbf{X}_{\pi^\natural(i)} \mathbf{X}_{\pi^\natural(i)}^\top \mathbf{M} \mathbf{X}_j \mathbf{X}_{\pi^{\natural-1}(j)}^\top \mathbf{X}_{\pi^\natural(i)}}_{\mathbb{1}_{j=\pi^{\natural 2}(j)} \left[\mathbb{1}_{i=j} \mathbb{E}\left\|\mathbf{X}_{\pi^\natural(i)}\right\|_{\mathrm{F}}^2 \left(\mathbf{X}_{\pi^\natural(i)}^\top \mathbf{M} \mathbf{X}_{\pi^\natural(i)}\right)^2 + \mathbb{1}_{i\neq j} \mathbb{E}\left(\mathbf{X}_{\pi^\natural(i)}^\top \mathbf{M} \mathbf{X}_{\pi^\natural(i)}\right)^2\right]}$$

$$+ \underbrace{\mathbb{E}\mathbf{X}_{\pi^\natural(i)}^\top \mathbf{X}_{\pi^{\natural-1}(j)} \mathbf{X}_j^\top \mathbf{M} \mathbf{X}_{\pi^\natural(i)} \mathbf{X}_{\pi^\natural(i)}^\top \mathbf{M} \mathbf{X}_{\pi^\natural(i)} \mathbf{X}_i^\top \mathbf{X}_{\pi^\natural(i)}}_{\mathbb{1}_{i=j} \mathbb{1}_{j=\pi^{\natural 2}(i)} \mathbb{E}\left\|\mathbf{X}_{\pi^\natural(i)}\right\|_{\mathrm{F}}^2 \left(\mathbf{X}_{\pi^\natural(i)}^\top \mathbf{M} \mathbf{X}_{\pi^\natural(i)}\right)^2}$$

$$+ \underbrace{\mathbb{E}\mathbf{X}_{\pi^\natural(i)}^\top \mathbf{X}_{\pi^{\natural-1}(j)} \mathbf{X}_j^\top \mathbf{M} \mathbf{X}_{\pi^\natural(i)} \mathbf{X}_{\pi^\natural(i)}^\top \mathbf{M} \mathbf{X}_{\pi^{\natural 2}(i)} \mathbf{X}_{\pi^\natural(i)}^\top \mathbf{X}_{\pi^\natural(i)}}_{\mathbb{1}_{j=\pi^{\natural 2}(i)} \mathbb{E}\left\|\mathbf{X}_{\pi^\natural(i)}\right\|_{\mathrm{F}}^4 \left(\mathbf{X}_{\pi^\natural(i)}^\top \mathbf{M} \mathbf{M} \mathbf{X}_{\pi^\natural(i)}\right)}$$

$$+ \underbrace{\mathbb{E}\mathbf{X}_{\pi^\natural(i)}^\top \mathbf{X}_{\pi^{\natural-1}(j)} \mathbf{X}_j^\top \mathbf{M} \mathbf{X}_{\pi^\natural(i)} \mathbf{X}_{\pi^\natural(i)}^\top \mathbf{M} \mathbf{X}_{\pi^\natural(j)} \mathbf{X}_j^\top \mathbf{X}_{\pi^\natural(i)}}_{\mathbb{1}_{j=\pi^{\natural 2}(j)} \left[\mathbb{1}_{i=j} \mathbb{E}\left\|\mathbf{X}_{\pi^\natural(i)}\right\|_{\mathrm{F}}^2 \left(\mathbf{X}_{\pi^\natural(i)}^\top \mathbf{M} \mathbf{X}_{\pi^\natural(i)}\right)^2 + \mathbb{1}_{i\neq j} \mathbb{E}\left(\mathbf{X}_{\pi^\natural(i)}^\top \mathbf{M} \mathbf{X}_{\pi^\natural(i)}\right)^2\right]}$$

$$+ \underbrace{\mathbb{E}\mathbf{X}_{\pi^\natural(i)}^\top \mathbf{X}_{\pi^{\natural-1}(j)} \mathbf{X}_j^\top \mathbf{M} \mathbf{X}_{\pi^\natural(i)} \mathbf{X}_{\pi^\natural(i)}^\top \mathbf{M} \mathbf{X}_j \mathbf{X}_{\pi^{\natural-1}(j)}^\top \mathbf{X}_{\pi^\natural(i)}}_{\mathbb{1}_{j=\pi^{\natural 2}(i)} \mathbb{E}\left\|\mathbf{X}_{\pi^\natural(i)}\right\|_{\mathrm{F}}^4 \left(\mathbf{X}_{\pi^\natural(i)}^\top \mathbf{M} \mathbf{M} \mathbf{X}_{\pi^\natural(i)}\right) + \mathbb{1}_{j\neq \pi^{\natural 2}(i)} \mathbb{E}\left\|\mathbf{X}_{\pi^\natural(i)}\right\|_{\mathrm{F}}^2 \left(\mathbf{X}_{\pi^\natural(i)}^\top \mathbf{M} \mathbf{M} \mathbf{X}_{\pi^\natural(i)}\right)}$$

$$= \left(1 + 2\mathbb{1}_{i=\pi^{\natural 2}(i)} + 3\mathbb{1}_{i=j} + 6\mathbb{1}_{i=j}\mathbb{1}_{j=\pi^{\natural 2}(i)}\right)(p+4)\left[(\mathrm{Tr}(\mathbf{M}))^2 + 2\mathrm{Tr}(\mathbf{M}\mathbf{M})\right]$$

$$+ \left(1 + 3\mathbb{1}_{j=\pi^{\natural 2}(i)}\right)(p+2)(p+4)\mathrm{Tr}(\mathbf{M}\mathbf{M})$$

$$+ 2\mathbb{1}_{j=\pi^{\natural 2}(j)}\mathbb{1}_{i\neq j}\left[(\mathrm{Tr}(\mathbf{M}))^2 + 2\,\mathrm{Tr}(\mathbf{M}\mathbf{M})\right]$$

$$+ \mathbb{1}_{j\neq\pi^{\natural 2}(i)}(p+2)\,\mathrm{Tr}(\mathbf{M}\mathbf{M}) + \mathbb{1}_{i\neq j}(p+2)\,\mathrm{Tr}(\mathbf{M}\mathbf{M}). \tag{39}$$

Then we calculate $\mathbb{E}\Lambda_{3,2}$ as

$$\mathbb{E}\Lambda_{3,2} = \underbrace{\mathbb{E}\mathbf{X}_j^\top\mathbf{X}_i\mathbf{X}_{\pi^\natural(i)}^\top\mathbf{M}\mathbf{X}_{\pi^\natural(i)}\mathbf{X}_{\pi^\natural(i)}^\top\mathbf{M}\mathbf{X}_{\pi^\natural(i)}\mathbf{X}_i^\top\mathbf{X}_j}_{\mathbb{1}_{i=j}\mathbb{E}\|\mathbf{X}_i\|_\mathrm{F}^4\left(\mathbf{X}_{\pi^\natural(i)}^\top\mathbf{M}\mathbf{X}_{\pi^\natural(i)}\right)^2 + p\mathbb{1}_{i\neq j}\mathbb{E}\left(\mathbf{X}_{\pi^\natural(i)}^\top\mathbf{M}\mathbf{X}_{\pi^\natural(i)}\right)^2}$$

$$+ \underbrace{\mathbb{E}\mathbf{X}_j^\top\mathbf{X}_i\mathbf{X}_{\pi^\natural(i)}^\top\mathbf{M}\mathbf{X}_{\pi^\natural(i)}\mathbf{X}_{\pi^\natural(i)}^\top\mathbf{M}\mathbf{X}_{\pi^{\natural 2}(i)}\mathbf{X}_{\pi^\natural(i)}^\top\mathbf{X}_j}_{\mathbb{1}_{i=\pi^{\natural 2}(i)}\left[\mathbb{1}_{i=j}\mathbb{E}\mathbf{X}_i^\top\mathbf{X}_i\mathbf{X}_{\pi^\natural(i)}^\top\mathbf{M}\mathbf{X}_{\pi^\natural(i)}\mathbf{X}_{\pi^\natural(i)}^\top\mathbf{M}\mathbf{X}_i\mathbf{X}_i^\top\mathbf{X}_{\pi^\natural(i)} + \mathbb{1}_{i\neq j}\mathbb{E}\left(\mathbf{X}_{\pi^\natural(i)}^\top\mathbf{M}\mathbf{X}_{\pi^\natural(i)}\right)^2\right]}$$

$$+ \underbrace{\mathbb{E}\mathbf{X}_j^\top\mathbf{X}_i\mathbf{X}_{\pi^\natural(i)}^\top\mathbf{M}\mathbf{X}_{\pi^\natural(i)}\mathbf{X}_{\pi^\natural(i)}^\top\mathbf{M}\mathbf{X}_{\pi^\natural(j)}\mathbf{X}_j^\top\mathbf{X}_j}_{\mathbb{1}_{i=j}\mathbb{E}\|\mathbf{X}_i\|_\mathrm{F}^4\left(\mathbf{X}_{\pi^\natural(i)}^\top\mathbf{M}\mathbf{X}_{\pi^\natural(i)}\right)^2}$$

$$+ \underbrace{\mathbb{E}\mathbf{X}_j^\top\mathbf{X}_i\mathbf{X}_{\pi^\natural(i)}^\top\mathbf{M}\mathbf{X}_{\pi^\natural(i)}\mathbf{X}_{\pi^\natural(i)}^\top\mathbf{M}\mathbf{X}_j\mathbf{X}_{\pi^{\natural-1}(j)}^\top\mathbf{X}_j}_{\mathbb{1}_{i=j}\mathbb{1}_{i=\pi^{\natural 2}(i)}\mathbb{E}\mathbf{X}_i^\top\mathbf{X}_i\mathbf{X}_{\pi^\natural(i)}^\top\mathbf{M}\mathbf{X}_{\pi^\natural(i)}\mathbf{X}_{\pi^\natural(i)}^\top\mathbf{M}\mathbf{X}_i\mathbf{X}_i^\top\mathbf{X}_{\pi^\natural(i)}}$$

$$+ \underbrace{\mathbb{E}\mathbf{X}_j^\top\mathbf{X}_{\pi^\natural(i)}\mathbf{X}_{\pi^{\natural 2}(i)}^\top\mathbf{M}\mathbf{X}_{\pi^\natural(i)}\mathbf{X}_{\pi^\natural(i)}^\top\mathbf{M}\mathbf{X}_{\pi^\natural(i)}\mathbf{X}_i^\top\mathbf{X}_j}_{\mathbb{1}_{i=\pi^{\natural 2}(i)}\left[\mathbb{1}_{i=j}\mathbb{E}\mathbf{X}_i^\top\mathbf{X}_i\mathbf{X}_{\pi^\natural(i)}^\top\mathbf{M}\mathbf{X}_{\pi^\natural(i)}\mathbf{X}_{\pi^\natural(i)}^\top\mathbf{M}\mathbf{X}_i\mathbf{X}_i^\top\mathbf{X}_{\pi^\natural(i)} + \mathbb{1}_{i\neq j}\mathbb{E}\left(\mathbf{X}_{\pi^\natural(i)}^\top\mathbf{M}\mathbf{X}_{\pi^\natural(i)}\right)^2\right]}$$

$$+ \underbrace{\mathbb{E}\mathbf{X}_j^\top\mathbf{X}_{\pi^\natural(i)}\mathbf{X}_{\pi^{\natural 2}(i)}^\top\mathbf{M}\mathbf{X}_{\pi^\natural(i)}\mathbf{X}_{\pi^\natural(i)}^\top\mathbf{M}\mathbf{X}_{\pi^{\natural 2}(i)}\mathbf{X}_{\pi^\natural(i)}^\top\mathbf{X}_j}_{\mathbb{1}_{j\neq\pi^{\natural 2}(i)}\mathbb{E}\left\|\mathbf{X}_{\pi^\natural(i)}\right\|_\mathrm{F}^2\mathbf{X}_{\pi^\natural(i)}^\top\mathbf{M}\mathbf{M}\mathbf{X}_{\pi^\natural(i)} + \mathbb{1}_{j=\pi^{\natural 2}(i)}\mathbb{E}\left(\mathbf{X}_j^\top\mathbf{X}_{\pi^\natural(i)}\right)^2\left(\mathbf{X}_{\pi^\natural(i)}^\top\mathbf{M}\mathbf{X}_j\right)^2}$$

$$+ \underbrace{\mathbb{E}\mathbf{X}_j^\top\mathbf{X}_{\pi^\natural(i)}\mathbf{X}_{\pi^{\natural 2}(i)}^\top\mathbf{M}\mathbf{X}_{\pi^\natural(i)}\mathbf{X}_{\pi^\natural(i)}^\top\mathbf{M}\mathbf{X}_{\pi^\natural(j)}\mathbf{X}_j^\top\mathbf{X}_j}_{\mathbb{1}_{i=j}\mathbb{1}_{i=\pi^{\natural 2}(i)}\mathbb{E}\mathbf{X}_i^\top\mathbf{X}_i\mathbf{X}_{\pi^\natural(i)}^\top\mathbf{M}\mathbf{X}_{\pi^\natural(i)}\mathbf{X}_{\pi^\natural(i)}^\top\mathbf{M}\mathbf{X}_i\mathbf{X}_i^\top\mathbf{X}_{\pi^\natural(i)}}$$

$$+ \underbrace{\mathbb{E}\mathbf{X}_j^\top\mathbf{X}_{\pi^\natural(i)}\mathbf{X}_{\pi^{\natural 2}(i)}^\top\mathbf{M}\mathbf{X}_{\pi^\natural(i)}\mathbf{X}_{\pi^\natural(i)}^\top\mathbf{M}\mathbf{X}_j\mathbf{X}_{\pi^{\natural-1}(j)}^\top\mathbf{X}_j}_{\mathbb{1}_{j=\pi^{\natural 2}(i)}\mathbb{E}\left(\mathbf{X}_j^\top\mathbf{X}_{\pi^\natural(i)}\right)^2\left(\mathbf{X}_{\pi^\natural(i)}^\top\mathbf{M}\mathbf{X}_j\right)^2}$$

$$+ \underbrace{\mathbb{E}\mathbf{X}_j^\top\mathbf{X}_j\mathbf{X}_{\pi^\natural(j)}^\top\mathbf{M}\mathbf{X}_{\pi^\natural(i)}\mathbf{X}_{\pi^\natural(i)}^\top\mathbf{M}\mathbf{X}_{\pi^\natural(i)}\mathbf{X}_i^\top\mathbf{X}_j}_{\mathbb{1}_{i=j}\mathbb{E}\|\mathbf{X}_i\|_\mathrm{F}^4\left(\mathbf{X}_{\pi^\natural(i)}^\top\mathbf{M}\mathbf{X}_{\pi^\natural(i)}\right)^2}$$

$$+ \underbrace{\mathbb{E}\mathbf{X}_j^\top\mathbf{X}_j\mathbf{X}_{\pi^\natural(j)}^\top\mathbf{M}\mathbf{X}_{\pi^\natural(i)}\mathbf{X}_{\pi^\natural(i)}^\top\mathbf{M}\mathbf{X}_{\pi^{\natural 2}(i)}\mathbf{X}_{\pi^\natural(i)}^\top\mathbf{X}_j}_{\mathbb{1}_{i=j}\mathbb{1}_{i=\pi^{\natural 2}(i)}\mathbb{E}\mathbf{X}_i^\top\mathbf{X}_i\mathbf{X}_{\pi^\natural(i)}^\top\mathbf{M}\mathbf{X}_{\pi^\natural(i)}\mathbf{X}_{\pi^\natural(i)}^\top\mathbf{M}\mathbf{X}_i\mathbf{X}_i^\top\mathbf{X}_{\pi^\natural(i)}}$$

$$+ \underbrace{\mathbb{E}\mathbf{X}_j^\top\mathbf{X}_j\mathbf{X}_{\pi^\natural(j)}^\top\mathbf{M}\mathbf{X}_{\pi^\natural(i)}\mathbf{X}_{\pi^\natural(i)}^\top\mathbf{M}\mathbf{X}_{\pi^\natural(j)}\mathbf{X}_j^\top\mathbf{X}_j}_{\mathbb{1}_{i=j}\mathbb{E}\|\mathbf{X}_i\|_\mathrm{F}^4\left(\mathbf{X}_{\pi^\natural(i)}^\top\mathbf{M}\mathbf{X}_{\pi^\natural(i)}\right)^2 + \mathbb{1}_{i\neq j}\mathbb{E}\|\mathbf{X}_j\|_\mathrm{F}^4\mathbf{X}_{\pi^\natural(i)}^\top\mathbf{M}\mathbf{M}\mathbf{X}_{\pi^\natural(i)}}$$

$$+ \underbrace{\mathbb{E}\mathbf{X}_j^\top\mathbf{X}_j\mathbf{X}_{\pi^\natural(j)}^\top\mathbf{M}\mathbf{X}_{\pi^\natural(i)}\mathbf{X}_{\pi^\natural(i)}^\top\mathbf{M}\mathbf{X}_j\mathbf{X}_{\pi^{\natural-1}(j)}^\top\mathbf{X}_j}_{\mathbb{1}_{j=\pi^{\natural 2}(j)}\left[\mathbb{1}_{i=j}\mathbb{E}\mathbf{X}_i^\top\mathbf{X}_i\mathbf{X}_{\pi^\natural(i)}^\top\mathbf{M}\mathbf{X}_{\pi^\natural(i)}\mathbf{X}_{\pi^\natural(i)}^\top\mathbf{M}\mathbf{X}_i\mathbf{X}_i^\top\mathbf{X}_{\pi^\natural(i)} + \mathbb{1}_{i\neq j}\mathbb{E}\|\mathbf{X}_j\|_\mathrm{F}^2\mathbf{X}_j^\top\mathbf{M}\mathbf{M}\mathbf{X}_j\right]}$$

$$+ \underbrace{\mathbb{E}\mathbf{X}_j^\top\mathbf{X}_{\pi^{\natural-1}(j)}\mathbf{X}_j^\top\mathbf{M}\mathbf{X}_{\pi^\natural(i)}\mathbf{X}_{\pi^\natural(i)}^\top\mathbf{M}\mathbf{X}_{\pi^\natural(i)}\mathbf{X}_i^\top\mathbf{X}_j}_{\mathbb{1}_{i=j}\mathbb{1}_{i=\pi^{\natural 2}(i)}\mathbb{E}\mathbf{X}_i^\top\mathbf{X}_i\mathbf{X}_{\pi^\natural(i)}^\top\mathbf{M}\mathbf{X}_{\pi^\natural(i)}\mathbf{X}_{\pi^\natural(i)}^\top\mathbf{M}\mathbf{X}_i\mathbf{X}_i^\top\mathbf{X}_{\pi^\natural(i)}}$$

$$+ \underbrace{\mathbb{E}\mathbf{X}_j^\top\mathbf{X}_{\pi^{\natural-1}(j)}\mathbf{X}_j^\top\mathbf{M}\mathbf{X}_{\pi^\natural(i)}\mathbf{X}_{\pi^\natural(i)}^\top\mathbf{M}\mathbf{X}_{\pi^{\natural 2}(i)}\mathbf{X}_{\pi^\natural(i)}^\top\mathbf{X}_j}_{\mathbb{1}_{j=\pi^{\natural 2}(i)}\mathbb{E}\left(\mathbf{X}_j^\top\mathbf{X}_{\pi^\natural(i)}\right)^2\left(\mathbf{X}_{\pi^\natural(i)}^\top\mathbf{M}\mathbf{X}_j\right)^2}$$

$$+ \underbrace{\mathbb{E}\mathbf{X}_j^\top\mathbf{X}_{\pi^{\natural-1}(j)}\mathbf{X}_j^\top\mathbf{M}\mathbf{X}_{\pi^\natural(i)}\mathbf{X}_{\pi^\natural(i)}^\top\mathbf{M}\mathbf{X}_{\pi^\natural(j)}\mathbf{X}_j^\top\mathbf{X}_j}_{\mathbb{1}_{j=\pi^{\natural 2}(j)}\left[\mathbb{1}_{i=j}\mathbb{E}\mathbf{X}_i^\top\mathbf{X}_i\mathbf{X}_{\pi^\natural(i)}^\top\mathbf{M}\mathbf{X}_{\pi^\natural(i)}\mathbf{X}_{\pi^\natural(i)}^\top\mathbf{M}\mathbf{X}_i\mathbf{X}_i^\top\mathbf{X}_{\pi^\natural(i)} + \mathbb{1}_{i\neq j}\mathbb{E}\|\mathbf{X}_j\|_\mathrm{F}^2\mathbf{X}_j^\top\mathbf{M}\mathbf{M}\mathbf{X}_j\right]}$$

$$+ \underbrace{\mathbb{E}\mathbf{X}_j^\top \left(\mathbf{X}_{\pi^{\natural-1}(j)}\mathbf{X}_j^\top\right)\mathbf{M}\mathbf{X}_{\pi^\natural(i)}\mathbf{X}_{\pi^\natural(i)}^\top\mathbf{M}\left(\mathbf{X}_j\mathbf{X}_{\pi^{\natural-1}(j)}^\top\right)\mathbf{X}_j}_{\mathbb{1}_{j=\pi^{\natural2}(i)}\mathbb{E}\left(\mathbf{X}_j^\top\mathbf{X}_{\pi^\natural(i)}\right)^2\left(\mathbf{X}_{\pi^\natural(i)}^\top\mathbf{M}\mathbf{X}_j\right)^2+\mathbb{1}_{j\neq\pi^{\natural2}(i)}\mathbb{E}\|\mathbf{X}_j\|_F^2\mathbf{X}_j^\top\mathbf{M}\mathbf{M}\mathbf{X}_j}$$

$$= 4\mathbb{1}_{i=j}p(p+2)\left[[\mathrm{Tr}(\mathbf{M})]^2 + 2\,\mathrm{Tr}(\mathbf{M}\mathbf{M})\right] + \mathbb{1}_{i\neq j}p(p+2)\,\mathrm{Tr}(\mathbf{M}\mathbf{M})$$

$$+ 8\mathbb{1}_{i=j}\mathbb{1}_{i=\pi^{\natural2}(i)}(p+2)\left[(\mathrm{Tr}(\mathbf{M}))^2 + 2\,\mathrm{Tr}(\mathbf{M}\mathbf{M})\right]$$

$$+ 4\mathbb{1}_{j=\pi^{\natural2}(i)}\left[2\left[\mathrm{Tr}(\mathbf{M})\right]^2 + (p+6)\,\mathrm{Tr}(\mathbf{M}\mathbf{M})\right]$$

$$+ 2\mathbb{1}_{j\neq\pi^{\natural2}(i)}(p+2)\,\mathrm{Tr}(\mathbf{M}\mathbf{M}) + 2\mathbb{1}_{i\neq j}\mathbb{1}_{j=\pi^{\natural2}(j)}(p+2)\,\mathrm{Tr}(\mathbf{M}\mathbf{M})$$

$$+ \mathbb{1}_{i\neq j}\left(p + 2\mathbb{1}_{i=\pi^{\natural2}(i)}\right)\left[(\mathrm{Tr}(\mathbf{M}))^2 + 2\,\mathrm{Tr}(\mathbf{M}\mathbf{M})\right]. \tag{40}$$

The term $\Lambda_{3,3}$ is computed as

$$\mathbb{E}\Lambda_{3,3} = \underbrace{\mathbb{E}\mathbf{X}_{\pi^\natural(i)}^\top\mathbf{X}_i\mathbf{X}_{\pi^\natural(i)}^\top\mathbf{M}\mathbf{X}_{\pi^\natural(i)}\mathbf{X}_{\pi^\natural(i)}^\top\mathbf{M}\mathbf{X}_{\pi^\natural(i)}\mathbf{X}_i^\top\mathbf{X}_j}_{0}$$

$$+ \underbrace{\mathbb{E}\mathbf{X}_{\pi^\natural(i)}^\top\mathbf{X}_i\mathbf{X}_{\pi^\natural(i)}^\top\mathbf{M}\mathbf{X}_{\pi^\natural(i)}\mathbf{X}_{\pi^\natural(i)}^\top\mathbf{M}\mathbf{X}_{\pi^{\natural2}(i)}\mathbf{X}_{\pi^\natural(i)}^\top\mathbf{X}_j}_{0}$$

$$+ \underbrace{\mathbb{E}\mathbf{X}_{\pi^\natural(i)}^\top\mathbf{X}_i\mathbf{X}_{\pi^\natural(i)}^\top\mathbf{M}\mathbf{X}_{\pi^\natural(i)}\mathbf{X}_{\pi^\natural(i)}^\top\mathbf{M}\mathbf{X}_{\pi^\natural(j)}\mathbf{X}_j^\top\mathbf{X}_j}_{\mathbb{1}_{i=\pi^\natural(j)}p\mathbb{E}\left[\mathbf{X}_{\pi^\natural(i)}^\top\mathbf{M}\mathbf{X}_{\pi^\natural(i)}\right]^2}$$

$$+ \underbrace{\mathbb{E}\mathbf{X}_{\pi^\natural(i)}^\top\mathbf{X}_i\mathbf{X}_{\pi^\natural(i)}^\top\mathbf{M}\mathbf{X}_{\pi^\natural(i)}\mathbf{X}_{\pi^\natural(i)}^\top\mathbf{M}\mathbf{X}_j\mathbf{X}_{\pi^{\natural-1}(j)}^\top\mathbf{X}_j}_{0}$$

$$+ \underbrace{\mathbb{E}\mathbf{X}_{\pi^\natural(i)}^\top\mathbf{X}_{\pi^\natural(i)}\mathbf{X}_{\pi^{\natural2}(i)}^\top\mathbf{M}\mathbf{X}_{\pi^\natural(i)}\mathbf{X}_{\pi^\natural(i)}^\top\mathbf{M}\mathbf{X}_{\pi^\natural(i)}\mathbf{X}_i^\top\mathbf{X}_j}_{0}$$

$$+ \underbrace{\mathbb{E}\mathbf{X}_{\pi^\natural(i)}^\top\mathbf{X}_{\pi^\natural(i)}\mathbf{X}_{\pi^{\natural2}(i)}^\top\mathbf{M}\mathbf{X}_{\pi^\natural(i)}\mathbf{X}_{\pi^\natural(i)}^\top\mathbf{M}\mathbf{X}_{\pi^{\natural2}(i)}\mathbf{X}_{\pi^\natural(i)}^\top\mathbf{X}_j}_{0}$$

$$+ \underbrace{\mathbb{E}\mathbf{X}_{\pi^\natural(i)}^\top\mathbf{X}_{\pi^\natural(i)}\mathbf{X}_{\pi^{\natural2}(i)}^\top\mathbf{M}\mathbf{X}_{\pi^\natural(i)}\mathbf{X}_{\pi^\natural(i)}^\top\mathbf{M}\mathbf{X}_{\pi^\natural(j)}\mathbf{X}_j^\top\mathbf{X}_j}_{0}$$

$$+ \underbrace{\mathbb{E}\mathbf{X}_{\pi^\natural(i)}^\top\mathbf{X}_{\pi^\natural(i)}\mathbf{X}_{\pi^{\natural2}(i)}^\top\mathbf{M}\mathbf{X}_{\pi^\natural(i)}\mathbf{X}_{\pi^\natural(i)}^\top\mathbf{M}\mathbf{X}_j\mathbf{X}_{\pi^{\natural-1}(j)}^\top\mathbf{X}_j}_{\mathbb{1}_{j=\pi^{\natural3}(i)}\mathbb{E}\left\|\mathbf{X}_{\pi^\natural(i)}\right\|_F^2\mathbf{X}_{\pi^\natural(i)}^\top\mathbf{M}\mathbf{M}\mathbf{X}_{\pi^\natural(i)}}$$

$$+ \underbrace{\mathbb{E}\mathbf{X}_{\pi^\natural(i)}^\top\mathbf{X}_j\mathbf{X}_{\pi^\natural(j)}^\top\mathbf{M}\mathbf{X}_{\pi^\natural(i)}\mathbf{X}_{\pi^\natural(i)}^\top\mathbf{M}\mathbf{X}_{\pi^\natural(i)}\mathbf{X}_i^\top\mathbf{X}_j}_{\mathbb{1}_{i=\pi^\natural(j)}\mathbb{E}\left[\mathbf{X}_{\pi^\natural(i)}^\top\mathbf{M}\mathbf{X}_{\pi^\natural(i)}\right]^2}$$

$$+ \underbrace{\mathbb{E}\mathbf{X}_{\pi^\natural(i)}^\top\mathbf{X}_j\mathbf{X}_{\pi^\natural(j)}^\top\mathbf{M}\mathbf{X}_{\pi^\natural(i)}\mathbf{X}_{\pi^\natural(i)}^\top\mathbf{M}\mathbf{X}_{\pi^{\natural2}(i)}\mathbf{X}_{\pi^\natural(i)}^\top\mathbf{X}_j}_{0}$$

$$+ \underbrace{\mathbb{E}\mathbf{X}_{\pi^\natural(i)}^\top\mathbf{X}_j\mathbf{X}_{\pi^\natural(j)}^\top\mathbf{M}\mathbf{X}_{\pi^\natural(i)}\mathbf{X}_{\pi^\natural(i)}^\top\mathbf{M}\mathbf{X}_{\pi^\natural(j)}\mathbf{X}_j^\top\mathbf{X}_j}_{0}$$

$$+ \underbrace{\mathbb{E}\mathbf{X}_{\pi^\natural(i)}^\top\mathbf{X}_j\mathbf{X}_{\pi^\natural(j)}^\top\mathbf{M}\mathbf{X}_{\pi^\natural(i)}\mathbf{X}_{\pi^\natural(i)}^\top\mathbf{M}\mathbf{X}_j\mathbf{X}_{\pi^{\natural-1}(j)}^\top\mathbf{X}_j}_{0}$$

$$+ \underbrace{\mathbb{E}\mathbf{X}_{\pi^\natural(i)}^\top\mathbf{X}_{\pi^{\natural-1}(j)}\mathbf{X}_j^\top\mathbf{M}\mathbf{X}_{\pi^\natural(i)}\mathbf{X}_{\pi^\natural(i)}^\top\mathbf{M}\mathbf{X}_{\pi^\natural(i)}\mathbf{X}_i^\top\mathbf{X}_j}_{0}$$

$$+ \underbrace{\mathbb{E}\mathbf{X}_{\pi^\natural(i)}^\top\mathbf{X}_{\pi^{\natural-1}(j)}\mathbf{X}_j^\top\mathbf{M}\mathbf{X}_{\pi^\natural(i)}\mathbf{X}_{\pi^\natural(i)}^\top\mathbf{M}\mathbf{X}_{\pi^{\natural2}(i)}\mathbf{X}_{\pi^\natural(i)}^\top\mathbf{X}_j}_{\mathbb{1}_{j=\pi^{\natural3}(i)}\mathbb{E}\left[\mathbf{X}_{\pi^\natural(i)}^\top\mathbf{M}\mathbf{X}_{\pi^\natural(i)}\right]^2}$$

$$+ \underbrace{\mathbb{E}\mathbf{X}_{\pi^\natural(i)}^\top \mathbf{X}_{\pi^{\natural-1}(j)}\mathbf{X}_j^\top \mathbf{M}\mathbf{X}_{\pi^\natural(i)}\mathbf{X}_{\pi^\natural(i)}^\top \mathbf{M}\mathbf{X}_{\pi^\natural(j)}\mathbf{X}_j^\top \mathbf{X}_j}_{0}$$

$$+ \underbrace{\mathbb{E}\mathbf{X}_{\pi^\natural(i)}^\top \mathbf{X}_{\pi^{\natural-1}(j)}\mathbf{X}_j^\top \mathbf{M}\mathbf{X}_{\pi^\natural(i)}\mathbf{X}_{\pi^\natural(i)}^\top \mathbf{M}\mathbf{X}_j\mathbf{X}_{\pi^{\natural-1}(j)}^\top \mathbf{X}_j}_{0}$$

$$= \left[(p+1)\mathbb{1}_{i=\pi^\natural(j)} + \mathbb{1}_{j=\pi^{\natural3}(i)}\right]\left[\mathrm{Tr}(\mathbf{M})\right]^2$$
$$+ \left[2(p+1)\mathbb{1}_{i=\pi^\natural(j)} + (p+4)\mathbb{1}_{j=\pi^{\natural3}(i)}\right]\mathrm{Tr}(\mathbf{M}\mathbf{M}). \tag{41}$$

Then, we consider the term $\mathbb{E}\Lambda_{3,4}$, which can be written as

$$\mathbb{E}\Lambda_{3,4} = \underbrace{\mathbb{E}\mathbf{X}_j^\top \mathbf{X}_i\mathbf{X}_{\pi^\natural(i)}^\top \mathbf{M}\mathbf{X}_{\pi^\natural(i)}\mathbf{X}_{\pi^\natural(i)}^\top \mathbf{M}\mathbf{X}_{\pi^\natural(i)}\mathbf{X}_i^\top \mathbf{X}_{\pi^\natural(i)}}_{0}$$

$$+ \underbrace{\mathbb{E}\mathbf{X}_j^\top \mathbf{X}_i\mathbf{X}_{\pi^\natural(i)}^\top \mathbf{M}\mathbf{X}_{\pi^\natural(i)}\mathbf{X}_{\pi^\natural(i)}^\top \mathbf{M}\mathbf{X}_{\pi^{\natural2}(i)}\mathbf{X}_{\pi^\natural(i)}^\top \mathbf{X}_{\pi^\natural(i)}}_{0}$$

$$+ \underbrace{\mathbb{E}\mathbf{X}_j^\top \mathbf{X}_i\mathbf{X}_{\pi^\natural(i)}^\top \mathbf{M}\mathbf{X}_{\pi^\natural(i)}\mathbf{X}_{\pi^\natural(i)}^\top \mathbf{M}\mathbf{X}_{\pi^\natural(j)}\mathbf{X}_j^\top \mathbf{X}_{\pi^\natural(i)}}_{\mathbb{1}_{i=\pi^\natural(j)}\mathbb{E}\left[\mathbf{X}_{\pi^\natural(i)}^\top \mathbf{M}\mathbf{X}_{\pi^\natural(i)}\right]^2}$$

$$+ \underbrace{\mathbb{E}\mathbf{X}_j^\top \mathbf{X}_i\mathbf{X}_{\pi^\natural(i)}^\top \mathbf{M}\mathbf{X}_{\pi^\natural(i)}\mathbf{X}_{\pi^\natural(i)}^\top \mathbf{M}\mathbf{X}_j\mathbf{X}_{\pi^{\natural-1}(j)}^\top \mathbf{X}_{\pi^\natural(i)}}_{0}$$

$$+ \underbrace{\mathbb{E}\mathbf{X}_j^\top \mathbf{X}_{\pi^\natural(i)}\mathbf{X}_{\pi^{\natural2}(i)}^\top \mathbf{M}\mathbf{X}_{\pi^\natural(i)}\mathbf{X}_{\pi^\natural(i)}^\top \mathbf{M}\mathbf{X}_{\pi^\natural(i)}\mathbf{X}_i^\top \mathbf{X}_{\pi^\natural(i)}}_{0}$$

$$+ \underbrace{\mathbb{E}\mathbf{X}_j^\top \mathbf{X}_{\pi^\natural(i)}\mathbf{X}_{\pi^{\natural2}(i)}^\top \mathbf{M}\mathbf{X}_{\pi^\natural(i)}\mathbf{X}_{\pi^\natural(i)}^\top \mathbf{M}\mathbf{X}_{\pi^{\natural2}(i)}\mathbf{X}_{\pi^\natural(i)}^\top \mathbf{X}_{\pi^\natural(i)}}_{0}$$

$$+ \underbrace{\mathbb{E}\mathbf{X}_j^\top \mathbf{X}_{\pi^\natural(i)}\mathbf{X}_{\pi^{\natural2}(i)}^\top \mathbf{M}\mathbf{X}_{\pi^\natural(i)}\mathbf{X}_{\pi^\natural(i)}^\top \mathbf{M}\mathbf{X}_{\pi^\natural(j)}\mathbf{X}_j^\top \mathbf{X}_{\pi^\natural(i)}}_{0}$$

$$+ \underbrace{\mathbb{E}\mathbf{X}_j^\top \mathbf{X}_{\pi^\natural(i)}\mathbf{X}_{\pi^{\natural2}(i)}^\top \mathbf{M}\mathbf{X}_{\pi^\natural(i)}\mathbf{X}_{\pi^\natural(i)}^\top \mathbf{M}\mathbf{X}_j\mathbf{X}_{\pi^{\natural-1}(j)}^\top \mathbf{X}_{\pi^\natural(i)}}_{\mathbb{1}_{j=\pi^{\natural3}(i)}\mathbb{E}\left[\mathbf{X}_{\pi^\natural(i)}^\top \mathbf{M}\mathbf{X}_{\pi^\natural(i)}\right]^2}$$

$$+ \underbrace{\mathbb{E}\mathbf{X}_j^\top \mathbf{X}_j\mathbf{X}_{\pi^\natural(j)}^\top \mathbf{M}\mathbf{X}_{\pi^\natural(i)}\mathbf{X}_{\pi^\natural(i)}^\top \mathbf{M}\mathbf{X}_{\pi^\natural(i)}\mathbf{X}_i^\top \mathbf{X}_{\pi^\natural(i)}}_{\mathbb{1}_{i=\pi^\natural(j)}p\mathbb{E}\left[\mathbf{X}_{\pi^\natural(i)}^\top \mathbf{M}\mathbf{X}_{\pi^\natural(i)}\right]^2}$$

$$+ \underbrace{\mathbb{E}\mathbf{X}_j^\top \mathbf{X}_j\mathbf{X}_{\pi^\natural(j)}^\top \mathbf{M}\mathbf{X}_{\pi^\natural(i)}\mathbf{X}_{\pi^\natural(i)}^\top \mathbf{M}\mathbf{X}_{\pi^{\natural2}(i)}\mathbf{X}_{\pi^\natural(i)}^\top \mathbf{X}_{\pi^\natural(i)}}_{0}$$

$$+ \underbrace{\mathbb{E}\mathbf{X}_j^\top \mathbf{X}_j\mathbf{X}_{\pi^\natural(j)}^\top \mathbf{M}\mathbf{X}_{\pi^\natural(i)}\mathbf{X}_{\pi^\natural(i)}^\top \mathbf{M}\mathbf{X}_{\pi^\natural(j)}\mathbf{X}_j^\top \mathbf{X}_{\pi^\natural(i)}}_{0}$$

$$+ \underbrace{\mathbb{E}\mathbf{X}_j^\top \mathbf{X}_j\mathbf{X}_{\pi^\natural(j)}^\top \mathbf{M}\mathbf{X}_{\pi^\natural(i)}\mathbf{X}_{\pi^\natural(i)}^\top \mathbf{M}\mathbf{X}_j\mathbf{X}_{\pi^{\natural-1}(j)}^\top \mathbf{X}_{\pi^\natural(i)}}_{0}$$

$$+ \underbrace{\mathbb{E}\mathbf{X}_j^\top \mathbf{X}_{\pi^{\natural-1}(j)}\mathbf{X}_j^\top \mathbf{M}\mathbf{X}_{\pi^\natural(i)}\mathbf{X}_{\pi^\natural(i)}^\top \mathbf{M}\mathbf{X}_{\pi^\natural(i)}\mathbf{X}_i^\top \mathbf{X}_{\pi^\natural(i)}}_{0}$$

$$+ \underbrace{\mathbb{E}\mathbf{X}_j^\top \mathbf{X}_{\pi^{\natural-1}(j)}\mathbf{X}_j^\top \mathbf{M}\mathbf{X}_{\pi^\natural(i)}\mathbf{X}_{\pi^\natural(i)}^\top \mathbf{M}\mathbf{X}_{\pi^{\natural2}(i)}\mathbf{X}_{\pi^\natural(i)}^\top \mathbf{X}_{\pi^\natural(i)}}_{\mathbb{1}_{j=\pi^{\natural3}(i)}\mathbb{E}\left\|\mathbf{X}_{\pi^\natural(i)}\right\|_{\mathrm{F}}^2\mathbf{X}_{\pi^\natural(i)}^\top \mathbf{M}\mathbf{M}\mathbf{X}_{\pi^\natural(i)}}$$

$$+ \underbrace{\mathbb{E}\mathbf{X}_j^\top \mathbf{X}_{\pi^{\natural-1}(j)}\mathbf{X}_j^\top \mathbf{M}\mathbf{X}_{\pi^\natural(i)}\mathbf{X}_{\pi^\natural(i)}^\top \mathbf{M}\mathbf{X}_{\pi^\natural(j)}\mathbf{X}_j^\top \mathbf{X}_{\pi^\natural(i)}}_{0}$$

$$+ \underbrace{\mathbb{E}\mathbf{X}_j^\top \mathbf{X}_{\pi^{\natural-1}(j)}\mathbf{X}_j^\top \mathbf{M}\mathbf{X}_{\pi^\natural(i)}\mathbf{X}_{\pi^\natural(i)}^\top \mathbf{M}\mathbf{X}_j\mathbf{X}_{\pi^{\natural-1}(j)}^\top \mathbf{X}_{\pi^\natural(i)}}_{0}$$

$$= \left[ (p+1)\mathbb{1}_{i=\pi^\natural(j)} + \mathbb{1}_{j=\pi^{\natural 3}(i)} \right] \left[ \mathrm{Tr}(\mathbf{M}) \right]^2$$
$$+ \left[ 2(p+1)\mathbb{1}_{i=\pi^\natural(j)} + (p+4)\mathbb{1}_{j=\pi^{\natural 3}(i)} \right] \mathrm{Tr}(\mathbf{MM}). \tag{42}$$

Combing (39), (40), (41), and (42) together then yields

$$\mathbb{E}\Lambda_3 = \frac{4hp(p+2)}{n^2} \left[ 1 + o(1) \right] \left[ \mathrm{Tr}(\mathbf{M}) \right]^2 + 2p^2 \left[ 1 + o(1) \right] \mathrm{Tr}(\mathbf{MM}). \tag{43}$$

The proof is thus completed by summarizing the computations thereof. $\qquad\square$

.

**Lemma 6.** *We have*

$$\mathbb{E}\Xi_2^2 = 2 \left[ (p+2)(p+3) + (n-2)(p+1) \right] \left\| \mathbf{B}^\natural \right\|_{\mathrm{F}}^2 = 2np \left( 1 + p/n + o(1) \right) \left\| \mathbf{B}^\natural \right\|_{\mathrm{F}}^2,$$

*where $\Xi_2$ is defined in* (21).

*Proof.* We have

$$\mathbb{E}\Xi_2^2 = \mathbb{E} \left[ \mathbf{X}_{\pi^\natural(i)}^\top \mathbf{B}^\natural \mathbf{W}^\top \mathbf{X} \left( \mathbf{X}_{\pi^\natural(i)} - \mathbf{X}_j \right) \left( \mathbf{X}_{\pi^\natural(i)} - \mathbf{X}_j \right)^\top \mathbf{X}^\top \mathbf{W} \mathbf{B}^{\natural\top} \mathbf{X}_{\pi^\natural(i)} \right]$$

$$= \mathbb{E} \left[ \mathbf{X}_{\pi^\natural(i)}^\top \mathbf{B}^\natural \mathrm{Tr} \left[ \mathbf{X} \left( \mathbf{X}_{\pi^\natural(i)} - \mathbf{X}_j \right) \left( \mathbf{X}_{\pi^\natural(i)} - \mathbf{X}_j \right)^\top \mathbf{X}^\top \right] \mathbf{B}^{\natural\top} \mathbf{X}_{\pi^\natural(i)} \right]$$

$$= \mathbb{E} \left[ \left\| \mathbf{X} \left( \mathbf{X}_{\pi^\natural(i)} - \mathbf{X}_j \right) \right\|_2^2 \mathbf{X}_{\pi^\natural(i)}^\top \mathbf{B}^\natural \mathbf{B}^{\natural\top} \mathbf{X}_{\pi^\natural(i)} \right]$$

$$= \mathbb{E} \left[ \left\| \mathbf{X} \left( \mathbf{X}_{\pi^\natural(i)} - \mathbf{X}_j \right) \right\|_2^2 \times \left\| \mathbf{B}^{\natural\top} \mathbf{X}_{\pi^\natural(i)} \right\|_2^2 \right].$$

For the conciseness of notation, we assume $\pi^\natural(i) = 1$ and $j = 2$ w.l.o.g. Decomposing the term $\left\| \mathbf{X} \left( \mathbf{X}_1 - \mathbf{X}_2 \right) \right\|_2^2$ as

$$\left\| \mathbf{X} \left( \mathbf{X}_1 - \mathbf{X}_2 \right) \right\|_{\mathrm{F}}^2 = \underbrace{\left[ \mathbf{X}_1^\top \left( \mathbf{X}_1 - \mathbf{X}_2 \right) \right]^2}_{\mathcal{T}_1} + \underbrace{\left[ \mathbf{X}_2^\top \left( \mathbf{X}_1 - \mathbf{X}_2 \right) \right]^2}_{\mathcal{T}_2} + \underbrace{\sum_{i=3}^n \left[ \mathbf{X}_i^\top \left( \mathbf{X}_1 - \mathbf{X}_2 \right) \right]^2}_{\mathcal{T}_3},$$

we then separately bound the above three terms. For the first term $\mathbb{E}\mathcal{T}_1 \left\| \mathbf{B}^{\natural\top} \mathbf{X}_1 \right\|_{\mathrm{F}}^2$, we have

$$\mathbb{E}\mathcal{T}_1 \left\| \mathbf{B}^{\natural\top} \mathbf{X}_1 \right\|_{\mathrm{F}}^2 = \mathbb{E} \left[ \left( \left\| \mathbf{X}_1 \right\|_2^4 + \left( \mathbf{X}_1^\top \mathbf{X}_2 \right)^2 \right) \left\| \mathbf{B}^{\natural\top} \mathbf{X}_1 \right\|_{\mathrm{F}}^2 \right]$$

$$= \underbrace{\mathbb{E} \left\| \mathbf{X}_1 \right\|_2^4 \left\| \mathbf{B}^{\natural\top} \mathbf{X}_1 \right\|_{\mathrm{F}}^2}_{(p+2)(p+4)\left\|\mathbf{B}^\natural\right\|_{\mathrm{F}}^2} + \underbrace{\mathbb{E} \left( \mathbf{X}_1^\top \mathbf{X}_2 \right)^2 \left\| \mathbf{B}^{\natural\top} \mathbf{X}_1 \right\|_{\mathrm{F}}^2}_{(p+2)\left\|\mathbf{B}^\natural\right\|_{\mathrm{F}}^2} \overset{\text{①}}{=} (p+2)(p+5) \left\| \mathbf{B}^\natural \right\|_{\mathrm{F}}^2, \tag{44}$$

where ① is due to (66) and (67).

Similarly, for term $\mathbb{E}\mathcal{T}_2 \left\| \mathbf{B}^{\natural\top} \mathbf{X}_1 \right\|_{\mathrm{F}}^2$, we invoke (66) and (67), which gives

$$\mathbb{E}\mathcal{T}_2 \left\| \mathbf{B}^{\natural\top} \mathbf{X}_1 \right\|_{\mathrm{F}}^2 = \underbrace{\mathbb{E} \left[ \left( \mathbf{X}_1^\top \mathbf{X}_2 \right)^2 \left\| \mathbf{B}^{\natural\top} \mathbf{X}_1 \right\|_{\mathrm{F}}^2 \right]}_{(p+2)\left\|\mathbf{B}^\natural\right\|_{\mathrm{F}}^2} + \underbrace{\mathbb{E} \left\| \mathbf{X}_2 \right\|_2^4 \times \mathbb{E} \left\| \mathbf{B}^{\natural\top} \mathbf{X}_1 \right\|_{\mathrm{F}}^2}_{p(p+2)\left\|\mathbf{B}^\natural\right\|_{\mathrm{F}}^2} \tag{45}$$

$$\overset{\text{②}}{=} (p+1)(p+2) \left\| \mathbf{B}^\natural \right\|_{\mathrm{F}}^2, \tag{46}$$

where ② is due to (66).

For the last term $\mathbb{E}\mathcal{T}_3 \left\| \mathbf{B}^{\natural\top} \mathbf{X}_1 \right\|_{\mathrm{F}}^2$, we exploit the independence among the rows of matrix $\mathbf{X}$ and have

$$\mathbb{E}T_3 \left\| \mathbf{B}^{\natural\top} \mathbf{X}_1 \right\|_{\mathrm{F}}^2 = \sum_{i\geq 3} \mathbb{E} \left[ \left( \mathbf{X}_i^\top \left( \mathbf{X}_1 - \mathbf{X}_2 \right) \right)^2 \left\| \mathbf{B}^{\natural\top} \mathbf{X}_1 \right\|_{\mathrm{F}}^2 \right]$$

$$= \sum_{i \geq 3} \mathbb{E} \left[ \|\mathbf{X}_1 - \mathbf{X}_2\|_2^2 \cdot \left\|\mathbf{B}^{\natural\top} \mathbf{X}_1\right\|_{\mathrm{F}}^2 \right]$$

$$= \sum_{i \geq 3} \mathbb{E} \left[ \left( \|\mathbf{X}_1\|_2^2 + \|\mathbf{X}_2\|_2^2 \right) \cdot \left\|\mathbf{B}^{\natural\top} \mathbf{X}_1\right\|_{\mathrm{F}}^2 \right]$$

$$= 2 \sum_{i \geq 3} (p+1) \left\|\mathbf{B}^\natural\right\|_{\mathrm{F}}^2 = 2(n-2)(p+1) \left\|\mathbf{B}^\natural\right\|_{\mathrm{F}}^2. \tag{47}$$

The proof is then completed by combining (44), (46), and (47).

$\square$

**Lemma 7.** *We have*

$$\mathbb{E}\Xi_3^2 = 2n^2 \left[ \frac{p}{n} + \left(1 - \frac{h}{n}\right)^2 + \frac{p^2}{n^2} + \frac{4p(n-h)^2}{n^3} + o(1) \right] \mathrm{Tr}(\mathbf{M}),$$

*where $\Xi_3$ is defined in (21).*

*Proof.* To begin with, we decompose the term $\mathbb{E}\Xi_3^2$ as

$$\mathbb{E}\Xi_3^2 = \mathbb{E} \underbrace{\left[ \left(\mathbf{X}_{\pi^\natural(i)} - \mathbf{X}_j\right)^\top \boldsymbol{\Sigma}\mathbf{M}\boldsymbol{\Sigma}^\top \left(\mathbf{X}_{\pi^\natural(i)} - \mathbf{X}_j\right) \right]}_{\triangleq \Lambda_1} + 2\mathbb{E} \underbrace{\left[ \left(\mathbf{X}_{\pi^\natural(i)} - \mathbf{X}_j\right)^\top \boldsymbol{\Sigma}\mathbf{M}\boldsymbol{\Delta}^\top \left(\mathbf{X}_{\pi^\natural(i)} - \mathbf{X}_j\right) \right]}_{\triangleq \Lambda_2}$$

$$+ \mathbb{E} \underbrace{\left[ \left(\mathbf{X}_{\pi^\natural(i)} - \mathbf{X}_j\right)^\top \boldsymbol{\Delta}\mathbf{M}\boldsymbol{\Delta}^\top \left(\mathbf{X}_{\pi^\natural(i)} - \mathbf{X}_j\right) \right]}_{\triangleq \Lambda_3}. \tag{48}$$

**Step I.** First we consider $\mathbb{E}\Lambda_1$, which can be written as

$$\mathbb{E}\Lambda_1 = 2\mathbb{E}\,\mathrm{Tr}\left(\boldsymbol{\Sigma}\mathbf{M}\boldsymbol{\Sigma}^\top\right) \overset{\text{\textcircled{1}}}{=} 2n^2 \left[ \frac{p}{n} + \left(1 - \frac{h}{n}\right)^2 + o(1) \right] \mathrm{Tr}(\mathbf{M}), \tag{49}$$

where \textcircled{1} is due to Lemma 13.

**Step II.** Then we turn to $\mathbb{E}\Lambda_2$, which can be written as

$$\mathbb{E}\Lambda_2 = (n-h)\mathbb{E} \underbrace{\left[ \mathbf{X}_{\pi^\natural(i)}^\top \mathbf{M}\boldsymbol{\Delta}^\top \mathbf{X}_{\pi^\natural(i)} \right]}_{\Lambda_{2,1}} + (n-h)\mathbb{E} \underbrace{\left[ \mathbf{X}_j^\top \mathbf{M}\boldsymbol{\Delta}^\top \mathbf{X}_j \right]}_{\Lambda_{2,2}}$$

$$- (n-h)\mathbb{E} \underbrace{\left[ \mathbf{X}_{\pi^\natural(i)}^\top \mathbf{M}\boldsymbol{\Delta}^\top \mathbf{X}_j \right]}_{\Lambda_{2,3}} - (n-h)\mathbb{E} \underbrace{\left[ \mathbf{X}_j^\top \mathbf{M}\boldsymbol{\Delta}^\top \mathbf{X}_{\pi^\natural(i)} \right]}_{\Lambda_{2,4}}.$$

**Case $(s,s)$: $i = \pi^\natural(i)$ and $j = \pi^\natural(j)$.** We have

$$\mathbb{E}\Lambda_{2,1} = \mathbb{E}\mathbf{X}_i^\top \mathbf{M} \left(\mathbf{X}_i\mathbf{X}_i^\top + \mathbf{X}_j\mathbf{X}_j^\top\right) \mathbf{X}_i = (p+3)\,\mathrm{Tr}(\mathbf{M}),$$

$$\mathbb{E}\Lambda_{2,2} = \mathbb{E}\mathbf{X}_j^\top \mathbf{M} \left(\mathbf{X}_i\mathbf{X}_i^\top + \mathbf{X}_j\mathbf{X}_j^\top\right) \mathbf{X}_j = (p+3)\,\mathrm{Tr}(\mathbf{M}).$$

In addition, we can verify that $\mathbb{E}\Lambda_{2,2}$ and $\Lambda_{2,3}$ are both zero, which suggests that

$$\mathbb{E}\Lambda_2 = 2(n-h)(p+3)\,\mathrm{Tr}(\mathbf{M}). \tag{50}$$

**Case $(s,d)$: $i = \pi^\natural(i)$ and $j \neq \pi^\natural(j)$.** We have

$$\mathbb{E}\Lambda_{2,1} = \mathbb{E}\mathbf{X}_i^\top \mathbf{M} \left(\mathbf{X}_i\mathbf{X}_i^\top + \mathbf{X}_j\mathbf{X}_{\pi^\natural(j)}^\top + \mathbf{X}_{\pi^{\natural-1}(j)}\mathbf{X}_j^\top\right) \mathbf{X}_i = (p+2)\,\mathrm{Tr}(\mathbf{M});$$

$$\mathbb{E}\Lambda_{2,2} = \mathbb{E}\mathbf{X}_j^\top \mathbf{M} \left(\mathbf{X}_i\mathbf{X}_i^\top + \mathbf{X}_j\mathbf{X}_{\pi^\natural(j)}^\top + \mathbf{X}_{\pi^{\natural-1}(j)}\mathbf{X}_j^\top\right) \mathbf{X}_j = \mathrm{Tr}(\mathbf{M}).$$

Moreover, we have both $\mathbb{E}\Lambda_{2,3}$ and $\mathbb{E}\Lambda_{2,4}$ be zero, which suggests that

$$\mathbb{E}\Lambda_2 = (n-h)(p+3)\operatorname{Tr}(\mathbf{M}). \tag{51}$$

**Case $(d,s)$: $i \neq \pi^\natural(i)$ and $j = \pi^\natural(j)$.** We have

$$\mathbb{E}\Lambda_{2,1} = \mathbb{E}\left(\mathbf{X}_{\pi^\natural(i)}^\top \mathbf{M}\mathbf{X}_j \mathbf{X}_j^\top \mathbf{X}_{\pi^\natural(i)}\right) = \operatorname{Tr}(\mathbf{M}),$$

$$\mathbb{E}\Lambda_{2,2} = \mathbb{E}\mathbf{X}_j^\top \mathbf{M}\mathbf{X}_j \mathbf{X}_j^\top \mathbf{X}_j = (p+2)\operatorname{Tr}(\mathbf{M}).$$

Similar as above, we can verify both $\mathbb{E}\Lambda_{2,3}$ and $\mathbb{E}\Lambda_{2,4}$ are zero, which suggests that

$$\mathbb{E}\Lambda_2 = (n-h)(p+3)\operatorname{Tr}(\mathbf{M}). \tag{52}$$

**Case $(d,d)$: $i \neq \pi^\natural(i)$ and $j \neq \pi^\natural(j)$.** Different from the above three cases, we have $\mathbb{E}\Lambda_{2,1}$ and $\mathbb{E}\Lambda_{2,2}$ be zero and focus on the calculation of $\mathbb{E}\Lambda_{2,3}$ and $\mathbb{E}\Lambda_{2,4}$, which proceeds as

$$\mathbb{E}\Lambda_{2,3} = \underbrace{\mathbb{E}\left[\mathbf{X}_{\pi^\natural(i)}^\top \mathbf{M}\mathbf{X}_{\pi^\natural(i)}\mathbf{X}_i^\top \mathbf{X}_j\right]}_{p\mathbb{1}_{i=j}\|\mathbf{B}^\natural\|_F^2} + \underbrace{\mathbb{E}\left[\mathbf{X}_{\pi^\natural(i)}^\top \mathbf{M}\mathbf{X}_{\pi^{\natural 2}(i)}\mathbf{X}_{\pi^\natural(i)}^\top \mathbf{X}_j\right]}_{\mathbb{1}_{j=\pi^{\natural 2}(i)}\|\mathbf{B}^\natural\|_F^2}$$

$$+ \underbrace{\mathbb{E}\left[\mathbf{X}_{\pi^\natural(i)}^\top \mathbf{M}\mathbf{X}_{\pi^\natural(j)}\mathbf{X}_j^\top \mathbf{X}_j\right]}_{p\mathbb{1}_{i=j}\|\mathbf{B}^\natural\|_F^2} + \underbrace{\mathbb{E}\left[\mathbf{X}_{\pi^\natural(i)}^\top \mathbf{M}\mathbf{X}_j\mathbf{X}_{\pi^{\natural-1}(j)}^\top \mathbf{X}_j\right]}_{\mathbb{1}_{\pi^\natural(i)=\pi^{\natural-1}(j)}\|\mathbf{B}^\natural\|_F^2}$$

$$= 2\left[p\mathbb{1}_{i=j} + \mathbb{1}_{j=\pi^{\natural 2}(i)}\right]\operatorname{Tr}(\mathbf{M});$$

$$\mathbb{E}\Lambda_{2,4} = \underbrace{\mathbb{E}\left[\mathbf{X}_j^\top \mathbf{M}\mathbf{X}_{\pi^\natural(i)}\mathbf{X}_i^\top \mathbf{X}_{\pi^\natural(i)}\right]}_{\mathbb{1}_{i=j}\|\mathbf{B}^\natural\|_F^2} + \underbrace{\mathbb{E}\left[\mathbf{X}_j^\top \mathbf{M}\mathbf{X}_{\pi^{\natural 2}(i)}\mathbf{X}_{\pi^\natural(i)}^\top \mathbf{X}_{\pi^\natural(i)}\right]}_{p\mathbb{1}_{j=\pi^{\natural 2}(i)}\|\mathbf{B}^\natural\|_F^2}$$

$$+ \underbrace{\mathbb{E}\left[\mathbf{X}_j^\top \mathbf{M}\mathbf{X}_{\pi^\natural(j)}\mathbf{X}_j^\top \mathbf{X}_{\pi^\natural(i)}\right]}_{\mathbb{1}_{i=j}\|\mathbf{B}^\natural\|_F^2} + \underbrace{\mathbb{E}\left[\mathbf{X}_j^\top \mathbf{M}\mathbf{X}_j\mathbf{X}_{\pi^{\natural-1}(j)}^\top \mathbf{X}_{\pi^\natural(i)}\right]}_{p\mathbb{1}_{j=\pi^{\natural 2}(i)}\|\mathbf{B}^\natural\|_F^2}$$

$$= 2\left[p\mathbb{1}_{j=\pi^{\natural 2}(i)} + \mathbb{1}_{i=j}\right]\operatorname{Tr}(\mathbf{M}),$$

which suggests that

$$\mathbb{E}\Lambda_2 = -2(n-h)(p+1)\left(\mathbb{1}_{j=\pi^{\natural 2}(i)} + \mathbb{1}_{i=j}\right)\operatorname{Tr}(\mathbf{M}). \tag{53}$$

Combing (50), (51), (52), and (53), we conclude

$$\mathbb{E}\Lambda_2 = \frac{2p(n-h)^2}{n}\operatorname{Tr}(\mathbf{M})\left[1+o(1)\right]. \tag{54}$$

**Step III.** Then we turn to the calculation of $\mathbb{E}\Lambda_3$. First we perform the following decomposition

$$\Lambda_3 = \underbrace{\mathbf{X}_{\pi^\natural(i)}^\top \boldsymbol{\Delta}\mathbf{M}\boldsymbol{\Delta}^\top \mathbf{X}_{\pi^\natural(i)}}_{\Lambda_{3,1}} + \underbrace{\mathbf{X}_j^\top \boldsymbol{\Delta}\mathbf{M}\boldsymbol{\Delta}^\top \mathbf{X}_j}_{\Lambda_{3,2}} - \underbrace{\mathbf{X}_{\pi^\natural(i)}^\top \boldsymbol{\Delta}\mathbf{M}\boldsymbol{\Delta}^\top \mathbf{X}_j}_{\Lambda_{3,3}} - \underbrace{\mathbf{X}_j^\top \boldsymbol{\Delta}\mathbf{M}\boldsymbol{\Delta}^\top \mathbf{X}_{\pi^\natural(i)}}_{\Lambda_{3,4}}.$$

**Case $(s,s)$: $i = \pi^\natural(i)$ and $j = \pi^\natural(j)$.** We have

$$\mathbb{E}\Lambda_{3,1} = \underbrace{\mathbb{E}\left(\mathbf{X}_i^\top \mathbf{X}_i \mathbf{X}_i^\top \mathbf{M}\mathbf{X}_i \mathbf{X}_i^\top \mathbf{X}_i\right)}_{\mathbb{E}\|\mathbf{X}_i\|_2^4 \mathbf{X}_i^\top \mathbf{M}\mathbf{X}_i} + \underbrace{\mathbb{E}\left(\mathbf{X}_i^\top \mathbf{X}_i \mathbf{X}_i^\top \mathbf{M}\mathbf{X}_j \mathbf{X}_j^\top \mathbf{X}_i\right)}_{(p+2)\|\mathbf{B}^\natural\|_F^2}$$

$$+ \underbrace{\mathbb{E}\left(\mathbf{X}_i^\top \mathbf{X}_j \mathbf{X}_j^\top \mathbf{M}\mathbf{X}_i \mathbf{X}_i^\top \mathbf{X}_i\right)}_{(p+2)\|\mathbf{B}^\natural\|_F^2} + \underbrace{\mathbb{E}\left(\mathbf{X}_i^\top \mathbf{X}_j \mathbf{X}_j^\top \mathbf{M}\mathbf{X}_j \mathbf{X}_j^\top \mathbf{X}_i\right)}_{(p+2)\|\mathbf{B}^\natural\|_F^2} = (p+2)(p+7)\operatorname{Tr}(\mathbf{M});$$

$$\mathbb{E}\Lambda_{3,2} = \mathbb{E}\mathbf{X}_j^\top \left(\mathbf{X}_i\mathbf{X}_i^\top + \mathbf{X}_j\mathbf{X}_j^\top\right)\mathbf{M}\left(\mathbf{X}_i\mathbf{X}_i^\top + \mathbf{X}_j\mathbf{X}_j^\top\right)\mathbf{X}_j = (p+2)(p+7)\operatorname{Tr}(\mathbf{M}).$$

As for $\mathbb{E}\Lambda_{3,3}$ and $\mathbb{E}\Lambda_{3,4}$, easily we can verify that they are both zero and hence have

$$\mathbb{E}\Lambda_3 = 2(p+2)(p+7)\operatorname{Tr}(\mathbf{M}) = 2p^2\operatorname{Tr}(\mathbf{M})\left[1+o(1)\right]. \tag{55}$$

**Case $(s, d)$: $i = \pi^{\natural}(i)$ and $j \neq \pi^{\natural}(j)$.** We can write $\Lambda_{3,1}$ as

$$
\mathbb{E}\Lambda_{3,1} = \underbrace{\mathbb{E}\left(\mathbf{X}_i^{\top}\mathbf{X}_i\mathbf{X}_i^{\top}\mathbf{M}\mathbf{X}_i\mathbf{X}_i^{\top}\mathbf{X}_i\right)}_{\mathbb{E}\|\mathbf{X}_i\|_2^4\mathbf{X}_i^{\top}\mathbf{M}\mathbf{X}_i} + \underbrace{\mathbb{E}\left(\mathbf{X}_i^{\top}\mathbf{X}_i\mathbf{X}_i^{\top}\mathbf{M}\mathbf{X}_{\pi^{\natural}(j)}\mathbf{X}_j^{\top}\mathbf{X}_i\right)}_{0}
$$

$$
+ \underbrace{\mathbb{E}\left(\mathbf{X}_i^{\top}\mathbf{X}_i\mathbf{X}_i^{\top}\mathbf{M}\mathbf{X}_j\mathbf{X}_{\pi^{\natural-1}(j)}^{\top}\mathbf{X}_i\right)}_{0}
$$

$$
+ \underbrace{\mathbb{E}\left(\mathbf{X}_i^{\top}\mathbf{X}_j\mathbf{X}_{\pi^{\natural}(j)}^{\top}\mathbf{M}\mathbf{X}_i\mathbf{X}_i^{\top}\mathbf{X}_i\right)}_{0} + \underbrace{\mathbb{E}\left(\mathbf{X}_i^{\top}\mathbf{X}_j\mathbf{X}_{\pi^{\natural}(j)}^{\top}\mathbf{M}\mathbf{X}_{\pi^{\natural}(j)}\mathbf{X}_j^{\top}\mathbf{X}_i\right)}_{p\|\mathbf{B}^{\natural}\|_{\mathrm{F}}^2}
$$

$$
+ \underbrace{\mathbb{E}\left(\mathbf{X}_i^{\top}\mathbf{X}_j\mathbf{X}_{\pi^{\natural}(j)}^{\top}\mathbf{M}\mathbf{X}_j\mathbf{X}_{\pi^{\natural-1}(j)}^{\top}\mathbf{X}_i\right)}_{\mathbb{1}_{j=\pi^{\natural 2}(j)}\operatorname{Tr}(\mathbf{M})}
$$

$$
+ \underbrace{\mathbb{E}\left(\mathbf{X}_i^{\top}\mathbf{X}_{\pi^{\natural-1}(j)}\mathbf{X}_j^{\top}\mathbf{M}\mathbf{X}_i\mathbf{X}_i^{\top}\mathbf{X}_i\right)}_{0} + \underbrace{\mathbb{E}\left(\mathbf{X}_i^{\top}\mathbf{X}_{\pi^{\natural-1}(j)}\mathbf{X}_j^{\top}\mathbf{M}\mathbf{X}_{\pi^{\natural}(j)}\mathbf{X}_j^{\top}\mathbf{X}_i\right)}_{\mathbb{1}_{j=\pi^{\natural 2}(j)}\operatorname{Tr}(\mathbf{M})}
$$

$$
+ \underbrace{\mathbb{E}\left(\mathbf{X}_i^{\top}\mathbf{X}_{\pi^{\natural-1}(j)}\mathbf{X}_j^{\top}\mathbf{M}\mathbf{X}_j\mathbf{X}_{\pi^{\natural-1}(j)}^{\top}\mathbf{X}_i\right)}_{p\|\mathbf{B}^{\natural}\|_{\mathrm{F}}^2}
$$

$$
= \left(p^2 + 8p + 8 + 2\mathbb{1}_{j=\pi^{\natural 2}(j)}\right)\operatorname{Tr}(\mathbf{M}).
$$

Mean $\Lambda_{3,2}$ can be written as

$$
\mathbb{E}\Lambda_{3,2} = \underbrace{\mathbb{E}\left(\mathbf{X}_j^{\top}\mathbf{X}_i\mathbf{X}_i^{\top}\mathbf{M}\mathbf{X}_i\mathbf{X}_i^{\top}\mathbf{X}_j\right)}_{(p+2)\|\mathbf{B}^{\natural}\|_{\mathrm{F}}^2} + \underbrace{\mathbb{E}\left(\mathbf{X}_j^{\top}\mathbf{X}_i\mathbf{X}_i^{\top}\mathbf{M}\mathbf{X}_{\pi^{\natural}(j)}\mathbf{X}_j^{\top}\mathbf{X}_j\right)}_{0}
$$

$$
+ \underbrace{\mathbb{E}\left(\mathbf{X}_j^{\top}\mathbf{X}_i\mathbf{X}_i^{\top}\mathbf{M}\mathbf{X}_j\mathbf{X}_{\pi^{\natural-1}(j)}^{\top}\mathbf{X}_j\right)}_{0}
$$

$$
+ \underbrace{\mathbb{E}\left(\mathbf{X}_j^{\top}\mathbf{X}_j\mathbf{X}_{\pi^{\natural}(j)}^{\top}\mathbf{M}\mathbf{X}_i\mathbf{X}_i^{\top}\mathbf{X}_j\right)}_{0} + \underbrace{\mathbb{E}\left(\mathbf{X}_j^{\top}\mathbf{X}_j\mathbf{X}_{\pi^{\natural}(j)}^{\top}\mathbf{M}\mathbf{X}_{\pi^{\natural}(j)}\mathbf{X}_j^{\top}\mathbf{X}_j\right)}_{\mathbb{E}\|\mathbf{X}_j\|_{\mathrm{F}}^4\operatorname{Tr}(\mathbf{M})}
$$

$$
+ \underbrace{\mathbb{E}\left(\mathbf{X}_j^{\top}\mathbf{X}_j\mathbf{X}_{\pi^{\natural}(j)}^{\top}\mathbf{M}\mathbf{X}_j\mathbf{X}_{\pi^{\natural-1}(j)}^{\top}\mathbf{X}_j\right)}_{\mathbb{1}_{\pi^{\natural-1}(j)=\pi^{\natural}(j)}(p+2)\|\mathbf{B}^{\natural}\|_{\mathrm{F}}^2}
$$

$$
+ \underbrace{\mathbb{E}\left(\mathbf{X}_j^{\top}\mathbf{X}_{\pi^{\natural-1}(j)}\mathbf{X}_j^{\top}\mathbf{M}\mathbf{X}_i\mathbf{X}_i^{\top}\mathbf{X}_j\right)}_{0} + \underbrace{\mathbb{E}\left(\mathbf{X}_j^{\top}\mathbf{X}_{\pi^{\natural-1}(j)}\mathbf{X}_j^{\top}\mathbf{M}\mathbf{X}_{\pi^{\natural}(j)}\mathbf{X}_j^{\top}\mathbf{X}_j\right)}_{\mathbb{1}_{\pi^{\natural-1}(j)=\pi^{\natural}(j)}(p+2)\|\mathbf{B}^{\natural}\|_{\mathrm{F}}^2}
$$

$$
+ \underbrace{\mathbb{E}\left(\mathbf{X}_j^{\top}\mathbf{X}_{\pi^{\natural-1}(j)}\mathbf{X}_j^{\top}\mathbf{M}\mathbf{X}_j\mathbf{X}_{\pi^{\natural-1}(j)}^{\top}\mathbf{X}_j\right)}_{(p+2)\|\mathbf{B}^{\natural}\|_{\mathrm{F}}^2}
$$

$$
= (p + 2)\left(p + 2 + 2\mathbb{1}_{j=\pi^{\natural 2}(j)}\right)\operatorname{Tr}(\mathbf{M}).
$$

And for $\mathbb{E}\Lambda_{3,3}$ and $\mathbb{E}\Lambda_{3,4}$, easily we can verify that they are both zero. Then we conclude

$$
\mathbb{E}\Lambda_3 = 2\left(p^2 + 6p + 6 + (p+3)\mathbb{1}_{j=\pi^{\natural 2}(j)}\right)\operatorname{Tr}(\mathbf{M}) = 2p^2\operatorname{Tr}(\mathbf{M})\left[1 + o(1)\right]. \tag{56}
$$

**Case $(d, s)$: $i \neq \pi^{\natural}(i)$ and $j = \pi^{\natural}(j)$.** In this case, we can write $\Lambda_{3,1}$ as

$$
\mathbb{E}\Lambda_{3,1} = \underbrace{\mathbb{E}\left(\mathbf{X}_{\pi^{\natural}(i)}^{\top}\mathbf{X}_i\mathbf{X}_{\pi^{\natural}(i)}^{\top}\mathbf{M}\mathbf{X}_{\pi^{\natural}(i)}\mathbf{X}_i^{\top}\mathbf{X}_{\pi^{\natural}(i)}\right)}_{\mathbb{E}\|\mathbf{X}_i\|_{\mathrm{F}}^2\mathbf{X}_i^{\top}\mathbf{M}\mathbf{X}_i}
$$

$$
+ \underbrace{\mathbb{E}\left(\mathbf{X}_{\pi^{\natural}(i)}^{\top}\mathbf{X}_i\mathbf{X}_{\pi^{\natural}(i)}^{\top}\mathbf{M}\mathbf{X}_{\pi^{\natural 2}(i)}\mathbf{X}_{\pi^{\natural}(i)}^{\top}\mathbf{X}_{\pi^{\natural}(i)}\right)}_{\mathbb{1}_{i=\pi^{\natural 2}(i)}\mathbb{E}\|\mathbf{X}_i\|_{\mathrm{F}}^2\mathbf{X}_i^{\top}\mathbf{M}\mathbf{X}_i}
$$

$$+ \underbrace{\mathbb{E} \left( \mathbf{X}_{\pi^\natural(i)}^\top \mathbf{X}_i \mathbf{X}_{\pi^\natural(i)}^\top \mathbf{M} \mathbf{X}_j \mathbf{X}_j^\top \mathbf{X}_{\pi^\natural(i)} \right)}_{0}$$

$$+ \underbrace{\mathbb{E} \left( \mathbf{X}_{\pi^\natural(i)}^\top \mathbf{X}_{\pi^\natural(i)} \mathbf{X}_{\pi^{\natural 2}(i)}^\top \mathbf{M} \mathbf{X}_{\pi^\natural(i)} \mathbf{X}_i^\top \mathbf{X}_{\pi^\natural(i)} \right)}_{\mathbb{1}_{i = \pi^{\natural 2}(i)} \mathbb{E} \|\mathbf{X}_i\|_\mathrm{F}^2 \mathbf{X}_i^\top \mathbf{M} \mathbf{X}_i}$$

$$+ \underbrace{\mathbb{E} \left( \mathbf{X}_{\pi^\natural(i)}^\top \mathbf{X}_{\pi^\natural(i)} \mathbf{X}_{\pi^{\natural 2}(i)}^\top \mathbf{M} \mathbf{X}_{\pi^{\natural 2}(i)} \mathbf{X}_{\pi^\natural(i)}^\top \mathbf{X}_{\pi^\natural(i)} \right)}_{\mathbb{E} \|\mathbf{X}_i\|_\mathrm{F}^4 \operatorname{Tr}(\mathbf{M})}$$

$$+ \underbrace{\mathbb{E} \left( \mathbf{X}_{\pi^\natural(i)}^\top \mathbf{X}_{\pi^\natural(i)} \mathbf{X}_{\pi^{\natural 2}(i)}^\top \mathbf{M} \mathbf{X}_j \mathbf{X}_j^\top \mathbf{X}_{\pi^\natural(i)} \right)}_{0}$$

$$+ \underbrace{\mathbb{E} \left( \mathbf{X}_{\pi^\natural(i)}^\top \mathbf{X}_j \mathbf{X}_j^\top \mathbf{M} \mathbf{X}_{\pi^\natural(i)} \mathbf{X}_i^\top \mathbf{X}_{\pi^\natural(i)} \right)}_{0}$$

$$+ \underbrace{\mathbb{E} \left( \mathbf{X}_{\pi^\natural(i)}^\top \mathbf{X}_j \mathbf{X}_j^\top \mathbf{M} \mathbf{X}_{\pi^{\natural 2}(i)} \mathbf{X}_{\pi^\natural(i)}^\top \mathbf{X}_{\pi^\natural(i)} \right)}_{0}$$

$$+ \underbrace{\mathbb{E} \left( \mathbf{X}_{\pi^\natural(i)}^\top \mathbf{X}_j \mathbf{X}_j^\top \mathbf{M} \mathbf{X}_j \mathbf{X}_j^\top \mathbf{X}_{\pi^\natural(i)} \right)}_{\mathbb{E} \|\mathbf{X}_i\|_\mathrm{F}^2 \mathbf{X}_i^\top \mathbf{M} \mathbf{X}_i}$$

$$= (p + 2) \left( p + 2 + 2\mathbb{1}_{i = \pi^{\natural 2}(i)} \right) \operatorname{Tr}(\mathbf{M}).$$

We consider $\Lambda_{3,2}$ as

$$\mathbb{E}\Lambda_{3,2} = \underbrace{\mathbb{E} \mathbf{X}_j^\top \mathbf{X}_i \mathbf{X}_{\pi^\natural(i)}^\top \mathbf{M} \mathbf{X}_{\pi^\natural(i)} \mathbf{X}_i^\top \mathbf{X}_j}_{p \operatorname{Tr}(\mathbf{M})} + \underbrace{\mathbb{E} \mathbf{X}_j^\top \mathbf{X}_i \mathbf{X}_{\pi^\natural(i)}^\top \mathbf{M} \mathbf{X}_{\pi^{\natural 2}(i)} \mathbf{X}_{\pi^\natural(i)}^\top \mathbf{X}_j}_{\mathbb{1}_{i = \pi^{\natural 2}(i)} \operatorname{Tr}(\mathbf{M})}$$

$$+ \underbrace{\mathbb{E} \mathbf{X}_j^\top \mathbf{X}_i \mathbf{X}_{\pi^\natural(i)}^\top \mathbf{M} \mathbf{X}_j \mathbf{X}_j^\top \mathbf{X}_j}_{0}$$

$$+ \underbrace{\mathbb{E} \mathbf{X}_j^\top \mathbf{X}_{\pi^\natural(i)} \mathbf{X}_{\pi^{\natural 2}(i)}^\top \mathbf{M} \mathbf{X}_{\pi^\natural(i)} \mathbf{X}_i^\top \mathbf{X}_j}_{\mathbb{1}_{i = \pi^{\natural 2}(i)} \operatorname{Tr}(\mathbf{M})} + \underbrace{\mathbb{E} \mathbf{X}_j^\top \mathbf{X}_{\pi^\natural(i)} \mathbf{X}_{\pi^{\natural 2}(i)}^\top \mathbf{M} \mathbf{X}_{\pi^{\natural 2}(i)} \mathbf{X}_{\pi^\natural(i)}^\top \mathbf{X}_j}_{p \operatorname{Tr}(\mathbf{M})}$$

$$+ \underbrace{\mathbb{E} \mathbf{X}_j^\top \mathbf{X}_{\pi^\natural(i)} \mathbf{X}_{\pi^{\natural 2}(i)}^\top \mathbf{M} \mathbf{X}_j \mathbf{X}_j^\top \mathbf{X}_j}_{0}$$

$$+ \underbrace{\mathbb{E} \mathbf{X}_j^\top \mathbf{X}_j \mathbf{X}_j^\top \mathbf{M} \mathbf{X}_{\pi^\natural(i)} \mathbf{X}_i^\top \mathbf{X}_j}_{0} + \underbrace{\mathbb{E} \mathbf{X}_j^\top \mathbf{X}_j \mathbf{X}_j^\top \mathbf{M} \mathbf{X}_{\pi^{\natural 2}(i)} \mathbf{X}_{\pi^\natural(i)}^\top \mathbf{X}_j}_{0} + \underbrace{\mathbb{E} \mathbf{X}_j^\top \mathbf{X}_j \mathbf{X}_j^\top \mathbf{M} \mathbf{X}_j \mathbf{X}_j^\top \mathbf{X}_j}_{\mathbb{E} \|\mathbf{X}_i\|_\mathrm{F}^4 \mathbf{X}_i^\top \mathbf{M} \mathbf{X}_i}$$

$$= \left( p^2 + 8p + 8 + 2\mathbb{1}_{i = \pi^{\natural 2}(i)} \right) \operatorname{Tr}(\mathbf{M}).$$

Similarly, as above, we can verify that $\mathbb{E}\Lambda_{3,3} = 0$ and $\mathbb{E}\Lambda_{3,4} = 0$. Hence, we can conclude

$$\mathbb{E}\Lambda_3 = 2 \left( p^2 + 6p + 6 + (p + 3)\mathbb{1}_{i = \pi^{\natural 2}(i)} \right) \operatorname{Tr}(\mathbf{M}) = 2p^2 \operatorname{Tr}(\mathbf{M}) \left[ 1 + o(1) \right]. \tag{57}$$

**Case $(d, d)$: $i \neq \pi^\natural(i)$ and $j \neq \pi^\natural(j)$.** We write $\Lambda_{3,1}$ as

$$\mathbb{E}\Lambda_{3,1} = \underbrace{\mathbb{E} \, \mathbf{X}_{\pi^\natural(i)}^\top \mathbf{X}_i \mathbf{X}_{\pi^\natural(i)}^\top \mathbf{M} \mathbf{X}_{\pi^\natural(i)} \mathbf{X}_i^\top \mathbf{X}_{\pi^\natural(i)}}_{\mathbb{E} \|\mathbf{X}_i\|_\mathrm{F}^2 \mathbf{X}_i^\top \mathbf{M} \mathbf{X}_i} + \underbrace{\mathbb{E} \, \mathbf{X}_{\pi^\natural(i)}^\top \mathbf{X}_i \mathbf{X}_{\pi^\natural(i)}^\top \mathbf{M} \mathbf{X}_{\pi^{\natural 2}(i)} \mathbf{X}_{\pi^\natural(i)}^\top \mathbf{X}_{\pi^\natural(i)}}_{\mathbb{1}_{i = \pi^{\natural 2}(i)} \mathbb{E} \|\mathbf{X}_i\|_\mathrm{F}^2 \mathbf{X}_i^\top \mathbf{M} \mathbf{X}_i}$$

$$+ \underbrace{\mathbb{E} \, \mathbf{X}_{\pi^\natural(i)}^\top \mathbf{X}_i \mathbf{X}_{\pi^\natural(i)}^\top \mathbf{M} \mathbf{X}_{\pi^\natural(j)} \mathbf{X}_j^\top \mathbf{X}_{\pi^\natural(i)}}_{\mathbb{1}_{i = j} \mathbb{E} \|\mathbf{X}_i\|_\mathrm{F}^2 \mathbf{X}_i^\top \mathbf{M} \mathbf{X}_i} + \underbrace{\mathbb{E} \, \mathbf{X}_{\pi^\natural(i)}^\top \mathbf{X}_i \mathbf{X}_{\pi^\natural(i)}^\top \mathbf{M} \mathbf{X}_j \mathbf{X}_{\pi^{\natural-1}(j)}^\top \mathbf{X}_{\pi^\natural(i)}}_{\mathbb{1}_{i = j} \mathbb{1}_{i = \pi^{\natural 2}(i)} \mathbb{E} \|\mathbf{X}_i\|_\mathrm{F}^2 \mathbf{X}_i^\top \mathbf{M} \mathbf{X}_i}$$

$$+ \underbrace{\mathbb{E} \, \mathbf{X}_{\pi^\natural(i)}^\top \mathbf{X}_{\pi^\natural(i)} \mathbf{X}_{\pi^{\natural 2}(i)}^\top \mathbf{M} \mathbf{X}_{\pi^\natural(i)} \mathbf{X}_i^\top \mathbf{X}_{\pi^\natural(i)}}_{\mathbb{1}_{i = \pi^{\natural 2}(i)} \mathbb{E} \|\mathbf{X}_i\|_\mathrm{F}^2 \mathbf{X}_i^\top \mathbf{M} \mathbf{X}_i} + \underbrace{\mathbb{E} \, \mathbf{X}_{\pi^\natural(i)}^\top \mathbf{X}_{\pi^\natural(i)} \mathbf{X}_{\pi^{\natural 2}(i)}^\top \mathbf{M} \mathbf{X}_{\pi^{\natural 2}(i)} \mathbf{X}_{\pi^\natural(i)}^\top \mathbf{X}_{\pi^\natural(i)}}_{\mathbb{E} \|\mathbf{X}_i\|_\mathrm{F}^4 \operatorname{Tr}(\mathbf{M})}$$

$$
+ \underbrace{\mathbb{E}\, \mathbf{X}_{\pi^{\natural}(i)}^{\top}\mathbf{X}_{\pi^{\natural}(i)}\mathbf{X}_{\pi^{\natural 2}(i)}^{\top}\mathbf{M}\mathbf{X}_{\pi^{\natural}(j)}\mathbf{X}_{j}^{\top}\mathbf{X}_{\pi^{\natural}(i)}}_{\mathbb{1}_{i=j}\mathbb{1}_{i=\pi^{\natural 2}(i)}\mathbb{E}\|\mathbf{X}_i\|_{\mathrm{F}}^2\mathbf{X}_i^{\top}\mathbf{M}\mathbf{X}_i} + \underbrace{\mathbb{E}\, \mathbf{X}_{\pi^{\natural}(i)}^{\top}\mathbf{X}_{\pi^{\natural}(i)}\mathbf{X}_{\pi^{\natural 2}(i)}^{\top}\mathbf{M}\mathbf{X}_{j}\mathbf{X}_{\pi^{\natural -1}(j)}^{\top}\mathbf{X}_{\pi^{\natural}(i)}}_{\mathbb{1}_{j=\pi^{\natural 2}(i)}\mathbb{E}\|\mathbf{X}_i\|_{\mathrm{F}}^4\operatorname{Tr}(\mathbf{M})}
$$

$$
+ \underbrace{\mathbb{E}\, \mathbf{X}_{\pi^{\natural}(i)}^{\top}\mathbf{X}_{j}\mathbf{X}_{\pi^{\natural}(j)}^{\top}\mathbf{M}\mathbf{X}_{\pi^{\natural}(i)}\mathbf{X}_{i}^{\top}\mathbf{X}_{\pi^{\natural}(i)}}_{\mathbb{1}_{i=j}\mathbb{E}\|\mathbf{X}_i\|_{\mathrm{F}}^2\mathbf{X}_i^{\top}\mathbf{M}\mathbf{X}_i} + \underbrace{\mathbb{E}\, \mathbf{X}_{\pi^{\natural}(i)}^{\top}\mathbf{X}_{j}\mathbf{X}_{\pi^{\natural}(j)}^{\top}\mathbf{M}\mathbf{X}_{\pi^{\natural 2}(i)}\mathbf{X}_{\pi^{\natural}(i)}^{\top}\mathbf{X}_{\pi^{\natural}(i)}}_{\mathbb{1}_{i=j}\mathbb{1}_{i=\pi^{\natural 2}(i)}\mathbb{E}\|\mathbf{X}_i\|_{\mathrm{F}}^2\mathbf{X}_i^{\top}\mathbf{M}\mathbf{X}_i}
$$

$$
+ \underbrace{\mathbb{E}\, \mathbf{X}_{\pi^{\natural}(i)}^{\top}\mathbf{X}_{j}\mathbf{X}_{\pi^{\natural}(j)}^{\top}\mathbf{M}\mathbf{X}_{\pi^{\natural}(j)}\mathbf{X}_{j}^{\top}\mathbf{X}_{\pi^{\natural}(i)}}_{\mathbb{1}_{i=j}\mathbb{E}\|\mathbf{X}_i\|_{\mathrm{F}}^2\mathbf{X}_i^{\top}\mathbf{M}\mathbf{X}_i+\mathbb{1}_{i\neq j}p\operatorname{Tr}(\mathbf{M})} + \underbrace{\mathbb{E}\, \mathbf{X}_{\pi^{\natural}(i)}^{\top}\mathbf{X}_{j}\mathbf{X}_{\pi^{\natural}(j)}^{\top}\mathbf{M}\mathbf{X}_{j}\mathbf{X}_{\pi^{\natural -1}(j)}^{\top}\mathbf{X}_{\pi^{\natural}(i)}}_{\mathbb{1}_{j=\pi^{\natural 2}(j)}\left(\mathbb{1}_{i=j}\mathbb{E}\|\mathbf{X}_i\|_{\mathrm{F}}^2\mathbf{X}_i^{\top}\mathbf{M}\mathbf{X}_i+\ \mathbb{1}_{i\neq j}\operatorname{Tr}(\mathbf{M})\right)}
$$

$$
+ \underbrace{\mathbb{E}\, \mathbf{X}_{\pi^{\natural}(i)}^{\top}\mathbf{X}_{\pi^{\natural -1}(j)}\mathbf{X}_{j}^{\top}\mathbf{M}\mathbf{X}_{\pi^{\natural}(i)}\mathbf{X}_{i}^{\top}\mathbf{X}_{\pi^{\natural}(i)}}_{\mathbb{1}_{i=j}\mathbb{1}_{i=\pi^{\natural 2}(i)}\mathbb{E}\|\mathbf{X}_i\|_{\mathrm{F}}^2\mathbf{X}_i^{\top}\mathbf{M}\mathbf{X}_i} + \underbrace{\mathbb{E}\, \mathbf{X}_{\pi^{\natural}(i)}^{\top}\mathbf{X}_{\pi^{\natural -1}(j)}\mathbf{X}_{j}^{\top}\mathbf{M}\mathbf{X}_{\pi^{\natural 2}(i)}\mathbf{X}_{\pi^{\natural}(i)}^{\top}\mathbf{X}_{\pi^{\natural}(i)}}_{\mathbb{1}_{j=\pi^{\natural 2}(i)}\mathbb{E}\|\mathbf{X}_i\|_{\mathrm{F}}^4\operatorname{Tr}(\mathbf{M})}
$$

$$
+ \underbrace{\mathbb{E}\, \mathbf{X}_{\pi^{\natural}(i)}^{\top}\mathbf{X}_{\pi^{\natural -1}(j)}\mathbf{X}_{j}^{\top}\mathbf{M}\mathbf{X}_{\pi^{\natural}(j)}\mathbf{X}_{j}^{\top}\mathbf{X}_{\pi^{\natural}(i)}}_{\mathbb{1}_{j=\pi^{\natural 2}(j)}\left[\mathbb{1}_{i=j}\mathbb{E}\|\mathbf{X}_i\|_{\mathrm{F}}^2\mathbf{X}_i^{\top}\mathbf{M}\mathbf{X}_i+\mathbb{1}_{i\neq j}\operatorname{Tr}(\mathbf{M})\right]} + \underbrace{\mathbb{E}\, \mathbf{X}_{\pi^{\natural}(i)}^{\top}\mathbf{X}_{\pi^{\natural -1}(j)}\mathbf{X}_{j}^{\top}\mathbf{M}\mathbf{X}_{j}\mathbf{X}_{\pi^{\natural -1}(j)}^{\top}\mathbf{X}_{\pi^{\natural}(i)}}_{\mathbb{1}_{j=\pi^{\natural 2}(i)}\mathbb{E}\|\mathbf{X}_i\|_{\mathrm{F}}^4\operatorname{Tr}(\mathbf{M})+\mathbb{1}_{j\neq\pi^{\natural 2}(i)}p\operatorname{Tr}(\mathbf{M})}
$$

$$
= \left(p^2 + 5p + 2\right)\operatorname{Tr}(\mathbf{M}) + \mathbb{1}_{i=\pi^{\natural 2}(i)}2(p+2)\operatorname{Tr}(\mathbf{M}) + \mathbb{1}_{j=\pi^{\natural 2}(i)}(3p^2 + 5p)\operatorname{Tr}(\mathbf{M})
$$

$$
+ 2\mathbb{1}_{j=\pi^{\natural 2}(j)}\operatorname{Tr}(\mathbf{M}) + \mathbb{1}_{i=j}2(p+3)\operatorname{Tr}(\mathbf{M}) + \mathbb{1}_{i=j}\mathbb{1}_{i=\pi^{\natural 2}(i)}2(3p+5)\operatorname{Tr}(\mathbf{M}).
$$

We consider $\Lambda_{3,2}$ as

$$
\mathbb{E}\Lambda_{3,2} = \underbrace{\mathbb{E}\mathbf{X}_{j}^{\top}\mathbf{X}_{i}\mathbf{X}_{\pi^{\natural}(i)}^{\top}\mathbf{M}\mathbf{X}_{\pi^{\natural}(i)}\mathbf{X}_{i}^{\top}\mathbf{X}_{j}}_{\mathbb{1}_{i=j}\mathbb{E}\|\mathbf{X}_i\|_{\mathrm{F}}^4\operatorname{Tr}(\mathbf{M})+\mathbb{1}_{i\neq j}p\operatorname{Tr}(\mathbf{M})} + \underbrace{\mathbb{E}\mathbf{X}_{j}^{\top}\mathbf{X}_{i}\mathbf{X}_{\pi^{\natural}(i)}^{\top}\mathbf{M}\mathbf{X}_{\pi^{\natural 2}(i)}\mathbf{X}_{\pi^{\natural}(i)}^{\top}\mathbf{X}_{j}}_{\mathbb{1}_{i=\pi^{\natural 2}(i)}\left[\mathbb{1}_{i=j}\mathbb{E}\|\mathbf{X}_i\|_{\mathrm{F}}^2\mathbf{X}_i^{\top}\mathbf{M}\mathbf{X}_i+\mathbb{1}_{i\neq j}\operatorname{Tr}(\mathbf{M})\right]}
$$

$$
+ \underbrace{\mathbb{E}\mathbf{X}_{j}^{\top}\mathbf{X}_{i}\mathbf{X}_{\pi^{\natural}(i)}^{\top}\mathbf{M}\mathbf{X}_{\pi^{\natural}(j)}\mathbf{X}_{j}^{\top}\mathbf{X}_{j}}_{\mathbb{1}_{i=j}\mathbb{E}\|\mathbf{X}_i\|_{\mathrm{F}}^4\operatorname{Tr}(\mathbf{M})} + \underbrace{\mathbb{E}\mathbf{X}_{j}^{\top}\mathbf{X}_{i}\mathbf{X}_{\pi^{\natural}(i)}^{\top}\mathbf{M}\mathbf{X}_{j}\mathbf{X}_{\pi^{\natural -1}(j)}^{\top}\mathbf{X}_{j}}_{\mathbb{1}_{i=j}\mathbb{1}_{i=\pi^{\natural 2}(i)}\mathbb{E}\|\mathbf{X}_i\|_{\mathrm{F}}^2\mathbf{X}_i^{\top}\mathbf{M}\mathbf{X}_i}
$$

$$
+ \underbrace{\mathbb{E}\mathbf{X}_{j}^{\top}\mathbf{X}_{\pi^{\natural}(i)}\mathbf{X}_{\pi^{\natural 2}(i)}^{\top}\mathbf{M}\mathbf{X}_{\pi^{\natural}(i)}\mathbf{X}_{i}^{\top}\mathbf{X}_{j}}_{\mathbb{1}_{i=\pi^{\natural 2}(i)}\left[\mathbb{1}_{i=j}\mathbb{E}\|\mathbf{X}_i\|_{\mathrm{F}}^2\mathbf{X}_i^{\top}\mathbf{M}\mathbf{X}_i+\mathbb{1}_{i\neq j}\operatorname{Tr}(\mathbf{M})\right]} + \underbrace{\mathbb{E}\mathbf{X}_{j}^{\top}\mathbf{X}_{\pi^{\natural}(i)}\mathbf{X}_{\pi^{\natural 2}(i)}^{\top}\mathbf{M}\mathbf{X}_{\pi^{\natural 2}(i)}\mathbf{X}_{\pi^{\natural}(i)}^{\top}\mathbf{X}_{j}}_{\mathbb{1}_{j=\pi^{\natural 2}(i)}\mathbb{E}\|\mathbf{X}_i\|_{\mathrm{F}}^2\mathbf{X}_i^{\top}\mathbf{M}\mathbf{X}_i+\mathbb{1}_{j\neq\pi^{\natural 2}(i)}p\operatorname{Tr}(\mathbf{M})}
$$

$$
+ \underbrace{\mathbb{E}\mathbf{X}_{j}^{\top}\mathbf{X}_{\pi^{\natural}(i)}\mathbf{X}_{\pi^{\natural 2}(i)}^{\top}\mathbf{M}\mathbf{X}_{\pi^{\natural}(j)}\mathbf{X}_{j}^{\top}\mathbf{X}_{j}}_{\mathbb{1}_{i=j}\mathbb{1}_{i=\pi^{\natural 2}(i)}\mathbb{E}\|\mathbf{X}_i\|_{\mathrm{F}}^2\mathbf{X}_i^{\top}\mathbf{M}\mathbf{X}_i} + \underbrace{\mathbb{E}\mathbf{X}_{j}^{\top}\mathbf{X}_{\pi^{\natural}(i)}\mathbf{X}_{\pi^{\natural 2}(i)}^{\top}\mathbf{M}\mathbf{X}_{j}\mathbf{X}_{\pi^{\natural -1}(j)}^{\top}\mathbf{X}_{j}}_{\mathbb{1}_{j=\pi^{\natural 2}(i)}\mathbb{E}\|\mathbf{X}_i\|_{\mathrm{F}}^2\mathbf{X}_i^{\top}\mathbf{M}\mathbf{X}_i}
$$

$$
+ \underbrace{\mathbb{E}\mathbf{X}_{j}^{\top}\mathbf{X}_{j}\mathbf{X}_{\pi^{\natural}(j)}^{\top}\mathbf{M}\mathbf{X}_{\pi^{\natural}(i)}\mathbf{X}_{i}^{\top}\mathbf{X}_{j}}_{\mathbb{1}_{i=j}\mathbb{E}\|\mathbf{X}_i\|_{\mathrm{F}}^4\operatorname{Tr}(\mathbf{M})} + \underbrace{\mathbb{E}\mathbf{X}_{j}^{\top}\mathbf{X}_{j}\mathbf{X}_{\pi^{\natural}(j)}^{\top}\mathbf{M}\mathbf{X}_{\pi^{\natural 2}(i)}\mathbf{X}_{\pi^{\natural}(i)}^{\top}\mathbf{X}_{j}}_{\mathbb{1}_{i=j}\mathbb{1}_{i=\pi^{\natural 2}(i)}\mathbb{E}\|\mathbf{X}_i\|_{\mathrm{F}}^2\mathbf{X}_i^{\top}\mathbf{M}\mathbf{X}_i}
$$

$$
+ \underbrace{\mathbb{E}\mathbf{X}_{j}^{\top}\mathbf{X}_{j}\mathbf{X}_{\pi^{\natural}(j)}^{\top}\mathbf{M}\mathbf{X}_{\pi^{\natural}(j)}\mathbf{X}_{j}^{\top}\mathbf{X}_{j}}_{\mathbb{E}\|\mathbf{X}_i\|_{\mathrm{F}}^4\operatorname{Tr}(\mathbf{M})} + \underbrace{\mathbb{E}\mathbf{X}_{j}^{\top}\mathbf{X}_{j}\mathbf{X}_{\pi^{\natural}(j)}^{\top}\mathbf{M}\mathbf{X}_{j}\mathbf{X}_{\pi^{\natural -1}(j)}^{\top}\mathbf{X}_{j}}_{\mathbb{1}_{j=\pi^{\natural 2}(j)}\mathbb{E}\|\mathbf{X}_i\|_{\mathrm{F}}^2\mathbf{X}_i^{\top}\mathbf{M}\mathbf{X}_i}
$$

$$
+ \underbrace{\mathbb{E}\mathbf{X}_{j}^{\top}\mathbf{X}_{\pi^{\natural -1}(j)}\mathbf{X}_{j}^{\top}\mathbf{M}\mathbf{X}_{\pi^{\natural}(i)}\mathbf{X}_{i}^{\top}\mathbf{X}_{j}}_{\mathbb{1}_{i=j}\mathbb{1}_{i=\pi^{\natural 2}(i)}\mathbb{E}\|\mathbf{X}_i\|_{\mathrm{F}}^2\mathbf{X}_i^{\top}\mathbf{M}\mathbf{X}_i} + \underbrace{\mathbb{E}\mathbf{X}_{j}^{\top}\mathbf{X}_{\pi^{\natural -1}(j)}\mathbf{X}_{j}^{\top}\mathbf{M}\mathbf{X}_{\pi^{\natural 2}(i)}\mathbf{X}_{\pi^{\natural}(i)}^{\top}\mathbf{X}_{j}}_{\mathbb{1}_{j=\pi^{\natural 2}(i)}\mathbb{E}\|\mathbf{X}_i\|_{\mathrm{F}}^2\mathbf{X}_i^{\top}\mathbf{M}\mathbf{X}_i}
$$

$$
+ \underbrace{\mathbb{E}\mathbf{X}_{j}^{\top}\mathbf{X}_{\pi^{\natural -1}(j)}\mathbf{X}_{j}^{\top}\mathbf{M}\mathbf{X}_{\pi^{\natural}(j)}\mathbf{X}_{j}^{\top}\mathbf{X}_{j}}_{\mathbb{1}_{j=\pi^{\natural 2}(j)}\mathbb{E}\|\mathbf{X}_i\|_{\mathrm{F}}^2\mathbf{X}_i^{\top}\mathbf{M}\mathbf{X}_i} + \underbrace{\mathbb{E}\mathbf{X}_{j}^{\top}\mathbf{X}_{\pi^{\natural -1}(j)}\mathbf{X}_{j}^{\top}\mathbf{M}\mathbf{X}_{j}\mathbf{X}_{\pi^{\natural -1}(j)}^{\top}\mathbf{X}_{j}}_{\mathbb{E}\|\mathbf{X}_i\|_{\mathrm{F}}^2\mathbf{X}_i^{\top}\mathbf{M}\mathbf{X}_i}
$$

$$
= \left(p^2 + 5p + 2\right)\operatorname{Tr}(\mathbf{M}) + \mathbb{1}_{j=\pi^{\natural 2}(j)}2(p+2)\operatorname{Tr}(\mathbf{M}) + \mathbb{1}_{i=j}\left(3p^2 + 5p\right)\operatorname{Tr}(\mathbf{M})
$$

$$
+ 2\mathbb{1}_{i=\pi^{\natural 2}(i)}\operatorname{Tr}(\mathbf{M}) + \mathbb{1}_{j=\pi^{\natural 2}(i)}2(p+3)\operatorname{Tr}(\mathbf{M}) + \mathbb{1}_{i=j}\mathbb{1}_{i=\pi^{\natural 2}(i)}2\left(3p+5\right)\operatorname{Tr}(\mathbf{M}).
$$

We consider $\Lambda_{3,3}$ as

$$
\mathbb{E}\Lambda_{3,3} = \underbrace{\mathbb{E}\, \mathbf{X}_{\pi^{\natural}(i)}^{\top}\mathbf{X}_{i}\mathbf{X}_{\pi^{\natural}(i)}^{\top}\mathbf{M}\mathbf{X}_{\pi^{\natural}(i)}\mathbf{X}_{i}^{\top}\mathbf{X}_{j}}_{0} + \underbrace{\mathbb{E}\mathbf{X}_{\pi^{\natural}(i)}^{\top}\mathbf{X}_{i}\mathbf{X}_{\pi^{\natural}(i)}^{\top}\mathbf{M}\mathbf{X}_{\pi^{\natural 2}(i)}\mathbf{X}_{\pi^{\natural}(i)}^{\top}\mathbf{X}_{j}}_{0}
$$

$$
+ \underbrace{\mathbb{E}\, \mathbf{X}_{\pi^{\natural}(i)}^{\top}\mathbf{X}_{i}\mathbf{X}_{\pi^{\natural}(i)}^{\top}\mathbf{M}\mathbf{X}_{\pi^{\natural}(j)}\mathbf{X}_{j}^{\top}\mathbf{X}_{j}}_{\mathbb{1}_{i=\pi^{\natural}(j)}p\operatorname{Tr}(\mathbf{M})} + \underbrace{\mathbb{E}\, \mathbf{X}_{\pi^{\natural}(i)}^{\top}\mathbf{X}_{i}\mathbf{X}_{\pi^{\natural}(i)}^{\top}\mathbf{M}\mathbf{X}_{j}\mathbf{X}_{\pi^{\natural -1}(j)}^{\top}\mathbf{X}_{j}}_{0}
$$

$$+ \underbrace{\mathbb{E}\, \mathbf{X}_{\pi^\natural(i)}^\top \mathbf{X}_{\pi^\natural(i)} \mathbf{X}_{\pi^{\natural 2}(i)}^\top \mathbf{M} \mathbf{X}_{\pi^\natural(i)} \mathbf{X}_i^\top \mathbf{X}_j}_{0} + \underbrace{\mathbb{E}\, \mathbf{X}_{\pi^\natural(i)}^\top \mathbf{X}_{\pi^\natural(i)} \mathbf{X}_{\pi^{\natural 2}(i)}^\top \mathbf{M} \mathbf{X}_{\pi^{\natural 2}(i)} \mathbf{X}_{\pi^\natural(i)}^\top \mathbf{X}_j}_{0}$$

$$+ \underbrace{\mathbb{E}\, \mathbf{X}_{\pi^\natural(i)}^\top \mathbf{X}_{\pi^\natural(i)} \mathbf{X}_{\pi^{\natural 2}(i)}^\top \mathbf{M} \mathbf{X}_{\pi^\natural(j)} \mathbf{X}_j^\top \mathbf{X}_j}_{0} + \underbrace{\mathbb{E}\, \mathbf{X}_{\pi^\natural(i)}^\top \mathbf{X}_{\pi^\natural(i)} \mathbf{X}_{\pi^{\natural 2}(i)}^\top \mathbf{M} \mathbf{X}_j \mathbf{X}_{\pi^{\natural -1}(j)}^\top \mathbf{X}_j}_{\mathbb{1}_{j=\pi^{\natural 3}(i)}\, p\, \mathrm{Tr}(\mathbf{M})}$$

$$+ \underbrace{\mathbb{E}\, \mathbf{X}_{\pi^\natural(i)}^\top \mathbf{X}_j \mathbf{X}_{\pi^\natural(j)}^\top \mathbf{M} \mathbf{X}_{\pi^\natural(i)} \mathbf{X}_i^\top \mathbf{X}_j}_{\mathbb{1}_{i=\pi^\natural(j)}\, \mathrm{Tr}(\mathbf{M})} + \underbrace{\mathbb{E}\, \mathbf{X}_{\pi^\natural(i)}^\top \mathbf{X}_j \mathbf{X}_{\pi^\natural(j)}^\top \mathbf{M} \mathbf{X}_{\pi^{\natural 2}(i)} \mathbf{X}_{\pi^\natural(i)}^\top \mathbf{X}_j}_{0}$$

$$+ \underbrace{\mathbb{E}\, \mathbf{X}_{\pi^\natural(i)}^\top \mathbf{X}_j \mathbf{X}_{\pi^\natural(j)}^\top \mathbf{M} \mathbf{X}_{\pi^\natural(j)} \mathbf{X}_j^\top \mathbf{X}_j}_{0} + \underbrace{\mathbb{E}\, \mathbf{X}_{\pi^\natural(i)}^\top \mathbf{X}_j \mathbf{X}_{\pi^\natural(j)}^\top \mathbf{M} \mathbf{X}_j \mathbf{X}_{\pi^{\natural -1}(j)}^\top \mathbf{X}_j}_{0}$$

$$+ \underbrace{\mathbb{E}\, \mathbf{X}_{\pi^\natural(i)}^\top \mathbf{X}_{\pi^{\natural -1}(j)} \mathbf{X}_j^\top \mathbf{M} \mathbf{X}_{\pi^\natural(i)} \mathbf{X}_i^\top \mathbf{X}_j}_{0} + \underbrace{\mathbb{E}\, \mathbf{X}_{\pi^\natural(i)}^\top \mathbf{X}_{\pi^{\natural -1}(j)} \mathbf{X}_j^\top \mathbf{M} \mathbf{X}_{\pi^{\natural 2}(i)} \mathbf{X}_{\pi^\natural(i)}^\top \mathbf{X}_j}_{\mathbb{1}_{j=\pi^{\natural 3}(i)}\, \mathrm{Tr}(\mathbf{M})}$$

$$+ \underbrace{\mathbb{E}\, \mathbf{X}_{\pi^\natural(i)}^\top \mathbf{X}_{\pi^{\natural -1}(j)} \mathbf{X}_j^\top \mathbf{M} \mathbf{X}_{\pi^\natural(j)} \mathbf{X}_j^\top \mathbf{X}_j}_{0} + \underbrace{\mathbb{E}\, \mathbf{X}_{\pi^\natural(i)}^\top \mathbf{X}_{\pi^{\natural -1}(j)} \mathbf{X}_j^\top \mathbf{M} \mathbf{X}_j \mathbf{X}_{\pi^{\natural -1}(j)}^\top \mathbf{X}_j}_{0}$$

$$= (p+1)\left[\mathbb{1}_{i=\pi^\natural(j)} + \mathbb{1}_{j=\pi^{\natural 3}(i)}\right] \mathrm{Tr}(\mathbf{M}).$$

Then we consider $\Lambda_{3,4}$ as

$$\mathbb{E}\Lambda_{3,4} = \underbrace{\mathbb{E}\mathbf{X}_j^\top \mathbf{X}_i \mathbf{X}_{\pi^\natural(i)}^\top \mathbf{M} \mathbf{X}_{\pi^\natural(i)} \mathbf{X}_i^\top \mathbf{X}_{\pi^\natural(i)}}_{0} + \underbrace{\mathbb{E}\mathbf{X}_j^\top \mathbf{X}_i \mathbf{X}_{\pi^\natural(i)}^\top \mathbf{M} \mathbf{X}_{\pi^{\natural 2}(i)} \mathbf{X}_{\pi^\natural(i)}^\top \mathbf{X}_{\pi^\natural(i)}}_{0}$$

$$+ \underbrace{\mathbb{E}\mathbf{X}_j^\top \mathbf{X}_i \mathbf{X}_{\pi^\natural(i)}^\top \mathbf{M} \mathbf{X}_{\pi^\natural(j)} \mathbf{X}_j^\top \mathbf{X}_{\pi^\natural(i)}}_{\mathbb{1}_{i=\pi^\natural(j)}\, \mathrm{Tr}(\mathbf{M})} + \underbrace{\mathbb{E}\mathbf{X}_j^\top \mathbf{X}_i \mathbf{X}_{\pi^\natural(i)}^\top \mathbf{M} \mathbf{X}_j \mathbf{X}_{\pi^{\natural -1}(j)}^\top \mathbf{X}_{\pi^\natural(i)}}_{0}$$

$$+ \underbrace{\mathbb{E}\mathbf{X}_j^\top \mathbf{X}_{\pi^\natural(i)} \mathbf{X}_{\pi^{\natural 2}(i)}^\top \mathbf{M} \mathbf{X}_{\pi^\natural(i)} \mathbf{X}_i^\top \mathbf{X}_{\pi^\natural(i)}}_{0} + \underbrace{\mathbb{E}\mathbf{X}_j^\top \mathbf{X}_{\pi^\natural(i)} \mathbf{X}_{\pi^{\natural 2}(i)}^\top \mathbf{M} \mathbf{X}_{\pi^{\natural 2}(i)} \mathbf{X}_{\pi^\natural(i)}^\top \mathbf{X}_{\pi^\natural(i)}}_{0}$$

$$+ \underbrace{\mathbb{E}\mathbf{X}_j^\top \mathbf{X}_{\pi^\natural(i)} \mathbf{X}_{\pi^{\natural 2}(i)}^\top \mathbf{M} \mathbf{X}_{\pi^\natural(j)} \mathbf{X}_j^\top \mathbf{X}_{\pi^\natural(i)}}_{0} + \underbrace{\mathbb{E}\mathbf{X}_j^\top \mathbf{X}_{\pi^\natural(i)} \mathbf{X}_{\pi^{\natural 2}(i)}^\top \mathbf{M} \mathbf{X}_j \mathbf{X}_{\pi^{\natural -1}(j)}^\top \mathbf{X}_{\pi^\natural(i)}}_{\mathbb{1}_{j=\pi^{\natural 3}(i)}\, \mathrm{Tr}(\mathbf{M})}$$

$$+ \underbrace{\mathbb{E}\mathbf{X}_j^\top \mathbf{X}_j \mathbf{X}_{\pi^\natural(j)}^\top \mathbf{M} \mathbf{X}_{\pi^\natural(i)} \mathbf{X}_i^\top \mathbf{X}_{\pi^\natural(i)}}_{p\mathbb{1}_{i=\pi^\natural(j)}\, \mathrm{Tr}(\mathbf{M})} + \underbrace{\mathbb{E}\mathbf{X}_j^\top \mathbf{X}_j \mathbf{X}_{\pi^\natural(j)}^\top \mathbf{M} \mathbf{X}_{\pi^{\natural 2}(i)} \mathbf{X}_{\pi^\natural(i)}^\top \mathbf{X}_{\pi^\natural(i)}}_{0}$$

$$+ \underbrace{\mathbb{E}\mathbf{X}_j^\top \mathbf{X}_j \mathbf{X}_{\pi^\natural(j)}^\top \mathbf{M} \mathbf{X}_{\pi^\natural(j)} \mathbf{X}_j^\top \mathbf{X}_{\pi^\natural(i)}}_{0} + \underbrace{\mathbb{E}\mathbf{X}_j^\top \mathbf{X}_j \mathbf{X}_{\pi^\natural(j)}^\top \mathbf{M} \mathbf{X}_j \mathbf{X}_{\pi^{\natural -1}(j)}^\top \mathbf{X}_{\pi^\natural(i)}}_{0}$$

$$+ \underbrace{\mathbb{E}\mathbf{X}_j^\top \mathbf{X}_{\pi^{\natural -1}(j)} \mathbf{X}_j^\top \mathbf{M} \mathbf{X}_{\pi^\natural(i)} \mathbf{X}_i^\top \mathbf{X}_{\pi^\natural(i)}}_{0} + \underbrace{\mathbb{E}\mathbf{X}_j^\top \mathbf{X}_{\pi^{\natural -1}(j)} \mathbf{X}_j^\top \mathbf{M} \mathbf{X}_{\pi^{\natural 2}(i)} \mathbf{X}_{\pi^\natural(i)}^\top \mathbf{X}_{\pi^\natural(i)}}_{p\mathbb{1}_{j=\pi^{\natural 3}(i)}\, \mathrm{Tr}(\mathbf{M})}$$

$$+ \underbrace{\mathbb{E}\mathbf{X}_j^\top \mathbf{X}_{\pi^{\natural -1}(j)} \mathbf{X}_j^\top \mathbf{M} \mathbf{X}_{\pi^\natural(j)} \mathbf{X}_j^\top \mathbf{X}_{\pi^\natural(i)}}_{0} + \underbrace{\mathbb{E}\mathbf{X}_j^\top \mathbf{X}_{\pi^{\natural -1}(j)} \mathbf{X}_j^\top \mathbf{M} \mathbf{X}_j \mathbf{X}_{\pi^{\natural -1}(j)}^\top \mathbf{X}_{\pi^\natural(i)}}_{0}$$

$$= (p+1)\left(\mathbb{1}_{i=\pi^\natural(j)} + \mathbb{1}_{j=\pi^{\natural 3}(i)}\right) \mathrm{Tr}(\mathbf{M}).$$

In summary, we have

$$\mathbb{E}\Lambda_3 = 2\left(p^2 + 5p + 2\right) \mathrm{Tr}(\mathbf{M}) + 2(p+3)\left[\mathbb{1}_{i=\pi^{\natural 2}(i)} + \mathbb{1}_{j=\pi^{\natural 2}(j)}\right] \mathrm{Tr}(\mathbf{M})$$
$$+ \left(3p^2 + 7p + 6\right)\left(\mathbb{1}_{i=j} + \mathbb{1}_{j=\pi^{\natural 2}(i)}\right) \mathrm{Tr}(\mathbf{M}) + \mathbb{1}_{i=j}\mathbb{1}_{i=\pi^{\natural 2}(i)} 4\left(3p+5\right) \mathrm{Tr}(\mathbf{M})$$
$$- 2(p+1)\left[\mathbb{1}_{i=\pi^\natural(j)} + \mathbb{1}_{j=\pi^{\natural 3}(i)}\right] \mathrm{Tr}(\mathbf{M}) = 2p^2 \mathrm{Tr}(\mathbf{M})\left[1 + o(1)\right]. \qquad (58)$$

Combining (55), (56), (57), and (58) then yields

$$\mathbb{E}\Lambda_3 = 2p^2 \mathrm{Tr}(\mathbf{M})\left[1 + o(1)\right]. \qquad (59)$$

The proof is then completed by (48), (49), (54), and (59).

$$\qquad\square$$

**Lemma 8.** *We have*

$$\mathbb{E}\Xi_4^2 = m(m+1)\left[p\left(p+2\right)\left(\mathbb{1}_{i=\pi^\natural(i)} + \mathbb{1}_{i=j}\right) + p\left(\mathbb{1}_{i\neq\pi^\natural(i)} + \mathbb{1}_{i\neq j}\right)\right] + 2mp(n+p+1)$$

$$= \frac{(n-h)m^2p^2}{n}\left[1 + o(1)\right],$$

*where $\Xi_4$ is defined in* (21).

*Proof.* For the conciseness of notation, we define $\boldsymbol{\Gamma}$ as $\mathbf{X}\left(\mathbf{X}_{\pi^\natural(i)} - \mathbf{X}_j\right)\left(\mathbf{X}_{\pi^\natural(i)} - \mathbf{X}_j\right)^\top\mathbf{X}^\top$ and hence have

$$\mathbb{E}\Xi_4^2 = \mathbb{E}\mathbf{W}_i^\top\mathbf{W}^\top\boldsymbol{\Gamma}\mathbf{W}\mathbf{W}_i.$$

We begin the discussion by expanding $\mathbf{W}\mathbf{W}_i$ as

$$\begin{bmatrix}\mathbf{W}_1^\top \\ \mathbf{W}_2^\top \\ \cdots \\ \mathbf{W}_n^\top\end{bmatrix}\mathbf{W}_i = \begin{bmatrix}\mathbf{W}_1^\top\mathbf{W}_i \\ \mathbf{W}_2^\top\mathbf{W}_i \\ \cdots \\ \mathbf{W}_n^\top\mathbf{W}_i\end{bmatrix}.$$

Then we obtain

$$\mathbb{E}\Xi_4^2 = \sum_{s=1}^n\sum_{t=1}^n \Gamma_{ij}\mathbb{E}\left[\left(\mathbf{W}_s^\top\mathbf{W}_i\right)\left(\mathbf{W}_t^\top\mathbf{W}_i\right)\right] = \Gamma_{ii}\mathbb{E}\left(\mathbf{W}_i^\top\mathbf{W}_i\right)^2 + \underbrace{\sum_{s\neq i}\sum_{t\neq i}\Gamma_{st}\mathbb{E}\left(\mathbf{W}_s^\top\mathbf{W}_i\mathbf{W}_t^\top\mathbf{W}_i\right)}_{\sum_{s\neq i}\Gamma_{ss}\mathbb{E}(\mathbf{W}_s^\top\mathbf{W}_i)^2}$$

$$= \Gamma_{ii}\mathbb{E}\left(\sum_{j=1}^m W_{ij}^2\right)^2 + \sum_{s\neq i}\Gamma_{ss}\cdot m = m(m+1)\mathbb{E}\Gamma_{ii} + m\mathbb{E}\operatorname{Tr}(\boldsymbol{\Gamma}). \tag{60}$$

We can thus complete the proof by separately computing $\mathbb{E}\operatorname{Tr}(\mathbf{M})$ and $\mathbb{E}\Gamma_{ii}$. First we compute $\mathbb{E}\Gamma_{ii}$, which proceeds as

$$\mathbb{E}\Gamma_{ii} = \mathbb{E}\left(\mathbf{X}_i^\top\mathbf{X}_{\pi^\natural(i)}\right)^2 + \mathbb{E}\left(\mathbf{X}_i^\top\mathbf{X}_j\right)^2$$

$$= \mathbb{1}_{i=\pi^\natural(i)}p(p+2) + \mathbb{1}_{i\neq\pi^\natural(i)}p + \mathbb{1}_{i=j}p(p+2) + \mathbb{1}_{i\neq j}p$$

$$= p\left(p+2\right)\left[\mathbb{1}_{i=\pi^\natural(i)} + \mathbb{1}_{i=j}\right] + p\left[\mathbb{1}_{i\neq\pi^\natural(i)} + \mathbb{1}_{i\neq j}\right]. \tag{61}$$

Then we turn to the computation of $\mathbb{E}\operatorname{Tr}(\mathbf{M})$, which proceeds as

$$\mathbb{E}\operatorname{Tr}(\boldsymbol{\Gamma}) = \left\|\mathbf{X}\left(\mathbf{X}_{\pi^\natural(i)} - \mathbf{X}_j\right)\right\|_{\mathrm{F}}^2$$

$$= \mathbb{E}\left\|\mathbf{X}_{\pi^\natural(i)}^\top\left(\mathbf{X}_{\pi^\natural(i)} - \mathbf{X}_j\right)\right\|_2^2 + \mathbb{E}\left\|\mathbf{X}_j^\top\left(\mathbf{X}_{\pi^\natural(i)} - \mathbf{X}_j\right)\right\|_2^2 + \sum_{s\neq\pi^\natural(i),j}\mathbb{E}\left\|\mathbf{X}_s^\top\left(\mathbf{X}_{\pi^\natural(i)} - \mathbf{X}_j\right)\right\|_2^2$$

$$= 2\mathbb{E}\left\|\mathbf{X}_{\pi^\natural(i)}\right\|_2^4 + 2\mathbb{E}\left(\mathbf{X}_{\pi^\natural(i)}^\top\mathbf{X}_j\right)^2 + 2\sum_{s\neq\pi^\natural(i),j}\mathbb{E}\|\mathbf{X}_s\|_2^2$$

$$= 2p(p+3) + 2(n-2)p = 2p(n+p+1). \tag{62}$$

The proof is thus completed by combing (60), (62), and (61). $\qquad\square$

**Lemma 9.** *We have*

$$\mathbb{E}\Xi_1\Xi_4 = \frac{mp(n-h)(n+p-h)}{n}\left[1 + o(1)\right]\operatorname{Tr}(\mathbf{M}),$$

*where $\Xi_1$ and $\Xi_4$ are defined in* (21).

*Proof.* We have

$$\mathbb{E}\Xi_1\Xi_4 = \mathbb{E}\underbrace{\mathbf{X}_{\pi^\natural(i)}^\top\mathbf{M}\mathbf{X}^\top\boldsymbol{\Pi}^{\natural\top}\mathbf{X}\left(\mathbf{X}_{\pi^\natural(i)} - \mathbf{X}_j\right)\left(\mathbf{X}_{\pi^\natural(i)} - \mathbf{X}_j\right)^\top\mathbf{X}^\top}_{\triangleq\boldsymbol{v}^\top}\mathbf{W}\mathbf{W}_i.$$

First we conditional on $\mathbf{X}$. Expanding the product $\mathbf{W}\mathbf{W}_i$ as

$$
\mathbb{E}
\begin{bmatrix}
\mathbf{W}_1^\top \mathbf{W}_i \\
\mathbf{W}_2^\top \mathbf{W}_i \\
\cdots \\
\mathbf{W}_i^\top \mathbf{W}_i \\
\cdots \\
\mathbf{W}_n^\top \mathbf{W}_i
\end{bmatrix}
=
\begin{bmatrix}
0 \\
0 \\
\cdots \\
m \\
\cdots \\
0
\end{bmatrix},
$$

we can compute $\mathbb{E}\Xi_1\Xi_2$ w.r.t. $\mathbf{W}$ as

$$
\mathbb{E}\left(\Xi_1 \Xi_2\right) = \mathbb{E}\boldsymbol{v}^\top \mathbf{W}\mathbf{W}_i = m\mathbb{E}\boldsymbol{v}_i,
$$

where $\boldsymbol{v}_i$ denotes the $i$-th entry of $\boldsymbol{v}$ and can be written as

$$
\boldsymbol{v}_i = \underbrace{\mathbf{X}_i^\top \left(\mathbf{X}_{\pi^\natural(i)} - \mathbf{X}_j\right)\left(\mathbf{X}_{\pi^\natural(i)} - \mathbf{X}_j\right)^\top \boldsymbol{\Sigma}\mathbf{M}\mathbf{X}_{\pi^\natural(i)}}_{\Lambda_1} + \underbrace{\mathbf{X}_i^\top \left(\mathbf{X}_{\pi^\natural(i)} - \mathbf{X}_j\right)\left(\mathbf{X}_{\pi^\natural(i)} - \mathbf{X}_j\right)^\top \boldsymbol{\Delta}\mathbf{M}\mathbf{X}_{\pi^\natural(i)}}_{\Lambda_2}.
$$

For $\Lambda_1$, we conclude

$$
\begin{aligned}
\mathbb{E}\Lambda_1 &= \mathbb{E}\mathbf{X}_i^\top \mathbf{X}_{\pi^\natural(i)}\mathbf{X}_{\pi^\natural(i)}^\top \boldsymbol{\Sigma}\mathbf{M}\mathbf{X}_{\pi^\natural(i)} + \mathbb{E}\mathbf{X}_i^\top \mathbf{X}_j \mathbf{X}_j^\top \boldsymbol{\Sigma}\mathbf{M}\mathbf{X}_{\pi^\natural(i)} \\
&\quad - \mathbb{E}\mathbf{X}_i^\top \mathbf{X}_{\pi^\natural(i)}\mathbf{X}_j^\top \boldsymbol{\Sigma}\mathbf{M}\mathbf{X}_{\pi^\natural(i)} - \mathbb{E}\mathbf{X}_i^\top \mathbf{X}_j \mathbf{X}_{\pi^\natural(i)}^\top \boldsymbol{\Sigma}\mathbf{M}\mathbf{X}_{\pi^\natural(i)} \\
&= \mathbb{1}_{i=\pi^\natural(i)}(p+3)\mathbb{E}\operatorname{Tr}(\boldsymbol{\Sigma}\mathbf{M}) - \mathbb{1}_{i=j}(p+1)\mathbb{E}\operatorname{Tr}(\boldsymbol{\Sigma}\mathbf{M}) \\
&= (n-h)\left(\mathbb{1}_{i=\pi^\natural(i)}(p+3) - \mathbb{1}_{i=j}(p+1)\right)\operatorname{Tr}(\mathbf{M}) = \frac{p(n-h)^2}{n}\left[1+o(1)\right]\operatorname{Tr}(\mathbf{M}).
\end{aligned}
$$

Then we turn to $\mathbb{E}\Lambda_2$ and obtain

$$
\begin{aligned}
\mathbb{E}\Lambda_2 &= \mathbb{E}\underbrace{\mathbf{X}_i^\top \mathbf{X}_{\pi^\natural(i)}\mathbf{X}_{\pi^\natural(i)}^\top \boldsymbol{\Delta}\mathbf{M}\mathbf{X}_{\pi^\natural(i)}}_{\Lambda_{2,1}} + \mathbb{E}\underbrace{\mathbf{X}_i^\top \mathbf{X}_j \mathbf{X}_j^\top \boldsymbol{\Delta}\mathbf{M}\mathbf{X}_{\pi^\natural(i)}}_{\Lambda_{2,2}} \\
&\quad - \mathbb{E}\underbrace{\mathbf{X}_i^\top \mathbf{X}_{\pi^\natural(i)}\mathbf{X}_j^\top \boldsymbol{\Delta}\mathbf{M}\mathbf{X}_{\pi^\natural(i)}}_{\Lambda_{2,3}} - \mathbb{E}\underbrace{\mathbf{X}_i^\top \mathbf{X}_j \mathbf{X}_{\pi^\natural(i)}^\top \boldsymbol{\Delta}\mathbf{M}\mathbf{X}_{\pi^\natural(i)}}_{\Lambda_{2,4}}.
\end{aligned}
$$

We compute the value of $\mathbb{E}\Lambda_2$ under the four different cases.

**Case $(s,s)$: $i = \pi^\natural(i)$ and $j = \pi^\natural(j)$.** In this case, we have $\boldsymbol{\Delta}$ be

$$
\boldsymbol{\Delta} = \boldsymbol{\Delta}^{(s,s)} = \mathbf{X}_i\mathbf{X}_i^\top + \mathbf{X}_j\mathbf{X}_j^\top.
$$

We have

$$
\begin{aligned}
\mathbb{E}\Lambda_{2,1} &= \mathbb{E}\mathbf{X}_i^\top \mathbf{X}_i\mathbf{X}_i^\top \left(\mathbf{X}_i\mathbf{X}_i^\top + \mathbf{X}_j\mathbf{X}_j^\top\right)\mathbf{M}\mathbf{X}_i = (p+2)(p+5)\operatorname{Tr}(\mathbf{M}), \\
\mathbb{E}\Lambda_{2,2} &= \mathbb{E}\mathbf{X}_i^\top \mathbf{X}_j\mathbf{X}_j^\top \left(\mathbf{X}_i\mathbf{X}_i^\top + \mathbf{X}_j\mathbf{X}_j^\top\right)\mathbf{M}\mathbf{X}_i = 2(p+2)\operatorname{Tr}(\mathbf{M}), \\
\mathbb{E}\Lambda_{2,3} &= \mathbb{E}\mathbf{X}_i^\top \mathbf{X}_i\mathbf{X}_j^\top \left(\mathbf{X}_i\mathbf{X}_i^\top + \mathbf{X}_j\mathbf{X}_j^\top\right)\mathbf{M}\mathbf{X}_i = 0, \\
\mathbb{E}\Lambda_{2,4} &= \mathbb{E}\mathbf{X}_i^\top \mathbf{X}_j\mathbf{X}_i^\top \left(\mathbf{X}_i\mathbf{X}_i^\top + \mathbf{X}_j\mathbf{X}_j^\top\right)\mathbf{M}\mathbf{X}_i = 0,
\end{aligned}
$$

which implies

$$
\mathbb{E}\Lambda_2 = (p+2)(p+7)\operatorname{Tr}(\mathbf{M}).
$$

**Case $(s,d)$: $i = \pi^\natural(i)$ and $j \neq \pi^\natural(j)$.** First we write $\boldsymbol{\Delta}$ as

$$
\boldsymbol{\Delta}^{(s,d)} = \mathbf{X}_i\mathbf{X}_i^\top + \mathbf{X}_j\mathbf{X}_{\pi^\natural(j)}^\top + \mathbf{X}_{\pi^{\natural-1}(j)}\mathbf{X}_j^\top.
$$

Then we conclude

$$
\mathbb{E}\Lambda_{2,1} = \mathbb{E}\mathbf{X}_i^\top \mathbf{X}_i\mathbf{X}_i^\top \left(\mathbf{X}_i\mathbf{X}_i^\top + \mathbf{X}_j\mathbf{X}_{\pi^\natural(j)}^\top + \mathbf{X}_{\pi^{\natural-1}(j)}\mathbf{X}_j^\top\right)\mathbf{M}\mathbf{X}_i = (p+2)(p+4)\operatorname{Tr}(\mathbf{M}),
$$

$$\mathbb{E}\Lambda_{2,2} = \mathbb{E}\mathbf{X}_i^\top \mathbf{X}_j \mathbf{X}_j^\top \left( \mathbf{X}_i \mathbf{X}_i^\top + \mathbf{X}_j \mathbf{X}_{\pi^\natural(j)}^\top + \mathbf{X}_{\pi^{\natural-1}(j)} \mathbf{X}_j^\top \right) \mathbf{M} \mathbf{X}_i = (p+2)\operatorname{Tr}(\mathbf{M}),$$

$$\mathbb{E}\Lambda_{2,3} = \mathbb{E}\mathbf{X}_i^\top \mathbf{X}_i \mathbf{X}_j^\top \left( \mathbf{X}_i \mathbf{X}_i^\top + \mathbf{X}_j \mathbf{X}_{\pi^\natural(j)}^\top + \mathbf{X}_{\pi^{\natural-1}(j)} \mathbf{X}_j^\top \right) \mathbf{M} \mathbf{X}_i = 0,$$

$$\mathbb{E}\Lambda_{2,4} = \mathbb{E}\mathbf{X}_i^\top \mathbf{X}_j \mathbf{X}_i^\top \left( \mathbf{X}_i \mathbf{X}_i^\top + \mathbf{X}_j \mathbf{X}_{\pi^\natural(j)}^\top + \mathbf{X}_{\pi^{\natural-1}(j)} \mathbf{X}_j^\top \right) \mathbf{M} \mathbf{X}_i = 0,$$

which suggests that

$$\mathbb{E}\Lambda_2 = (p+2)(p+5)\operatorname{Tr}(\mathbf{M}).$$

**Case $(d,s)$:** $i \neq \pi^\natural(i)$ **and** $j = \pi^\natural(j)$**.** In this case, $\mathbf{\Delta}$ reduces to

$$\mathbf{\Delta}^{(d,s)} = \mathbf{X}_i \mathbf{X}_{\pi^\natural(i)}^\top + \mathbf{X}_{\pi^\natural(i)} \mathbf{X}_{\pi^{\natural 2}(i)}^\top + \mathbf{X}_j \mathbf{X}_j^\top.$$

We have

$$\mathbb{E}\Lambda_{2,1} = \underbrace{\mathbb{E}\mathbf{X}_i^\top \mathbf{X}_{\pi^\natural(i)} \mathbf{X}_{\pi^\natural(i)}^\top \mathbf{X}_i \mathbf{X}_{\pi^\natural(i)}^\top \mathbf{M} \mathbf{X}_{\pi^\natural(i)}}_{\mathbb{E}\|\mathbf{X}_i\|_2^2 \mathbf{X}_i^\top \mathbf{M} \mathbf{X}_i} + \underbrace{\mathbb{E}\mathbf{X}_i^\top \mathbf{X}_{\pi^\natural(i)} \mathbf{X}_{\pi^\natural(i)}^\top \mathbf{X}_{\pi^\natural(i)} \mathbf{X}_{\pi^{\natural 2}(i)}^\top \mathbf{M} \mathbf{X}_{\pi^\natural(i)}}_{\mathbb{1}_{i=\pi^{\natural 2}(i)} \mathbb{E}\|\mathbf{X}_i\|_2^2 \mathbf{X}_i^\top \mathbf{M} \mathbf{X}_i}$$

$$+ \underbrace{\mathbb{E}\mathbf{X}_i^\top \mathbf{X}_{\pi^\natural(i)} \mathbf{X}_{\pi^\natural(i)}^\top \mathbf{X}_j \mathbf{X}_j^\top \mathbf{M} \mathbf{X}_{\pi^\natural(i)}}_{0} = (p+2)\left[1 + \mathbb{1}_{i=\pi^{\natural 2}(i)}\right]\operatorname{Tr}(\mathbf{M}),$$

$$\mathbb{E}\Lambda_{2,2} = \underbrace{\mathbb{E}\mathbf{X}_i^\top \mathbf{X}_j \mathbf{X}_j^\top \mathbf{X}_i \mathbf{X}_{\pi^\natural(i)}^\top \mathbf{M} \mathbf{X}_{\pi^\natural(i)}}_{p \operatorname{Tr}(\mathbf{M})} + \underbrace{\mathbb{E}\mathbf{X}_i^\top \mathbf{X}_j \mathbf{X}_j^\top \mathbf{X}_{\pi^\natural(i)} \mathbf{X}_{\pi^{\natural 2}(i)}^\top \mathbf{M} \mathbf{X}_{\pi^\natural(i)}}_{\mathbb{1}_{i=\pi^{\natural 2}(i)} \operatorname{Tr}(\mathbf{M})}$$

$$+ \underbrace{\mathbb{E}\mathbf{X}_i^\top \mathbf{X}_j \mathbf{X}_j^\top \mathbf{X}_j \mathbf{X}_j^\top \mathbf{M} \mathbf{X}_{\pi^\natural(i)}}_{0} = \left(p + \mathbb{1}_{i=\pi^{\natural 2}(i)}\right)\operatorname{Tr}(\mathbf{M}),$$

$$\mathbb{E}\Lambda_{2,3} = 0,$$
$$\mathbb{E}\Lambda_{2,4} = 0,$$

which suggests

$$\mathbb{E}\Lambda_2 = 2(p+1)\operatorname{Tr}(\mathbf{M}) + \mathbb{1}_{i=\pi^{\natural 2}(i)}(p+3)\operatorname{Tr}(\mathbf{M}).$$

**Case $(d,d)$:** $i \neq \pi^\natural(i)$ **and** $j \neq \pi^\natural(j)$**.** In this case, $\mathbf{\Delta}$ is written as

$$\mathbf{\Delta}^{(d,d)} = \mathbf{X}_i \mathbf{X}_{\pi^\natural(i)}^\top + \mathbf{X}_{\pi^\natural(i)} \mathbf{X}_{\pi^{\natural 2}(i)}^\top + \mathbf{X}_j \mathbf{X}_{\pi^\natural(j)}^\top + \mathbf{X}_{\pi^{\natural-1}(j)} \mathbf{X}_j^\top.$$

We have

$$\mathbb{E}\Lambda_{2,1} = \underbrace{\mathbb{E}\mathbf{X}_i^\top \mathbf{X}_{\pi^\natural(i)} \mathbf{X}_{\pi^\natural(i)}^\top \mathbf{X}_i \mathbf{X}_{\pi^\natural(i)}^\top \mathbf{M} \mathbf{X}_{\pi^\natural(i)}}_{\mathbb{E}\|\mathbf{X}_i\|_2^2 \mathbf{X}_i^\top \mathbf{M} \mathbf{X}_i} + \underbrace{\mathbb{E}\mathbf{X}_i^\top \mathbf{X}_{\pi^\natural(i)} \mathbf{X}_{\pi^\natural(i)}^\top \mathbf{X}_{\pi^\natural(i)} \mathbf{X}_{\pi^{\natural 2}(i)}^\top \mathbf{M} \mathbf{X}_{\pi^\natural(i)}}_{\mathbb{1}_{i=\pi^{\natural 2}(i)} \mathbb{E}\|\mathbf{X}_i\|_2^2 \mathbf{X}_i^\top \mathbf{M} \mathbf{X}_i}$$

$$+ \underbrace{\mathbb{E}\mathbf{X}_i^\top \mathbf{X}_{\pi^\natural(i)} \mathbf{X}_{\pi^\natural(i)}^\top \mathbf{X}_j \mathbf{X}_{\pi^\natural(j)}^\top \mathbf{M} \mathbf{X}_{\pi^\natural(i)}}_{\mathbb{1}_{i=j} \mathbb{E}\|\mathbf{X}_i\|_2^2 \mathbf{X}_i^\top \mathbf{M} \mathbf{X}_i} + \underbrace{\mathbb{E}\mathbf{X}_i^\top \mathbf{X}_{\pi^\natural(i)} \mathbf{X}_{\pi^\natural(i)}^\top \mathbf{X}_{\pi^{\natural-1}(j)} \mathbf{X}_j^\top \mathbf{M} \mathbf{X}_{\pi^\natural(i)}}_{\mathbb{1}_{i=j}\mathbb{1}_{i=\pi^{\natural 2}(i)} \mathbb{E}\|\mathbf{X}_i\|_2^2 \mathbf{X}_i^\top \mathbf{M} \mathbf{X}_i};$$

$$\mathbb{E}\Lambda_{2,2} = \underbrace{\mathbb{E}\mathbf{X}_i^\top \mathbf{X}_j \mathbf{X}_j^\top \mathbf{X}_i \mathbf{X}_{\pi^\natural(i)}^\top \mathbf{M} \mathbf{X}_{\pi^\natural(i)}}_{\mathbb{1}_{i=j}p(p+2)\operatorname{Tr}(\mathbf{M}) + \mathbb{1}_{i\neq j}p\operatorname{Tr}(\mathbf{M})} + \underbrace{\mathbb{E}\mathbf{X}_i^\top \mathbf{X}_j \mathbf{X}_j^\top \mathbf{X}_{\pi^\natural(i)} \mathbf{X}_{\pi^{\natural 2}(i)}^\top \mathbf{M} \mathbf{X}_{\pi^\natural(i)}}_{\mathbb{1}_{i=\pi^{\natural 2}(i)}\left[\mathbb{1}_{i=j}\mathbb{E}\|\mathbf{X}_i\|_2^2 \mathbf{X}_i^\top \mathbf{M} \mathbf{X}_i + \mathbb{1}_{i\neq j}\operatorname{Tr}(\mathbf{M})\right]}$$

$$+ \underbrace{\mathbb{E}\mathbf{X}_i^\top \mathbf{X}_j \mathbf{X}_j^\top \mathbf{X}_j \mathbf{X}_{\pi^\natural(j)}^\top \mathbf{M} \mathbf{X}_{\pi^\natural(i)}}_{\mathbb{1}_{i=j}p(p+2)\operatorname{Tr}(\mathbf{M})} + \underbrace{\mathbb{E}\mathbf{X}_i^\top \mathbf{X}_j \mathbf{X}_j^\top \mathbf{X}_{\pi^{\natural-1}(j)} \mathbf{X}_j^\top \mathbf{M} \mathbf{X}_{\pi^\natural(i)}}_{\mathbb{1}_{i=j}\mathbb{1}_{j=\pi^{\natural 2}(i)} \mathbb{E}\|\mathbf{X}\|_2^2 \mathbf{X}^\top \mathbf{M} \mathbf{X}};$$

$$\mathbb{E}\Lambda_{2,3} = \underbrace{\mathbb{E}\mathbf{X}_i^\top \mathbf{X}_{\pi^\natural(i)} \mathbf{X}_j^\top \mathbf{X}_i \mathbf{X}_{\pi^\natural(i)}^\top \mathbf{M} \mathbf{X}_{\pi^\natural(i)}}_{0} + \underbrace{\mathbb{E}\mathbf{X}_i^\top \mathbf{X}_{\pi^\natural(i)} \mathbf{X}_j^\top \mathbf{X}_{\pi^\natural(i)} \mathbf{X}_{\pi^{\natural 2}(i)}^\top \mathbf{M} \mathbf{X}_{\pi^\natural(i)}}_{0}$$

$$+ \underbrace{\mathbb{E}\mathbf{X}_i^\top \mathbf{X}_{\pi^\natural(i)} \mathbf{X}_j^\top \mathbf{X}_j \mathbf{X}_{\pi^\natural(j)}^\top \mathbf{M} \mathbf{X}_{\pi^\natural(i)}}_{\mathbb{1}_{i=\pi^\natural(j)}p\operatorname{Tr}(\mathbf{M})} + \underbrace{\mathbb{E}\mathbf{X}_i^\top \mathbf{X}_{\pi^\natural(i)} \mathbf{X}_j^\top \mathbf{X}_{\pi^{\natural-1}(j)} \mathbf{X}_j^\top \mathbf{M} \mathbf{X}_{\pi^\natural(i)}}_{0};$$

$$\mathbb{E}\Lambda_{2,4} = \underbrace{\mathbb{E}\mathbf{X}_i^\top \mathbf{X}_j \mathbf{X}_{\pi^\natural(i)}^\top \mathbf{X}_i \mathbf{X}_{\pi^\natural(i)}^\top \mathbf{M}\mathbf{X}_{\pi^\natural(i)}}_{0} + \underbrace{\mathbb{E}\mathbf{X}_i^\top \mathbf{X}_j \mathbf{X}_{\pi^\natural(i)}^\top \mathbf{X}_{\pi^\natural(i)} \mathbf{X}_{\pi^{\natural 2}(i)}^\top \mathbf{M}\mathbf{X}_{\pi^\natural(i)}}_{0}$$

$$+ \underbrace{\mathbb{E}\mathbf{X}_i^\top \mathbf{X}_j \mathbf{X}_{\pi^\natural(i)}^\top \mathbf{X}_j \mathbf{X}_{\pi^\natural(j)}^\top \mathbf{M}\mathbf{X}_{\pi^\natural(i)}}_{\mathbb{1}_{i=\pi^\natural(j)}\operatorname{Tr}(\mathbf{M})} + \underbrace{\mathbb{E}\mathbf{X}_i^\top \mathbf{X}_j \mathbf{X}_{\pi^\natural(i)}^\top \mathbf{X}_{\pi^{\natural-1}(j)} \mathbf{X}_j^\top \mathbf{M}\mathbf{X}_{\pi^\natural(i)}}_{0}.$$

Hence we conclude

$$\mathbb{E}\Lambda_2 = 2(p+1)\operatorname{Tr}(\mathbf{M}) + \mathbb{1}_{i=j}2(p+1)^2\operatorname{Tr}(\mathbf{M}) + \mathbb{1}_{i=\pi^\natural(j)}(p+1)\operatorname{Tr}(\mathbf{M})$$
$$+ \mathbb{1}_{i=\pi^{\natural 2}(i)}(p+3)\operatorname{Tr}(\mathbf{M}) + \mathbb{1}_{i=j}\mathbb{1}_{i=\pi^{\natural 2}(i)}(3p+5)\operatorname{Tr}(\mathbf{M}).$$

$\square$

**Lemma 10.** *We have*

$$\mathbb{E}\Xi_2\Xi_3 = \frac{p(n-h)(n+p-h)}{n}\operatorname{Tr}(\mathbf{M})\left[1+o(1)\right],$$

*where $\Xi_2$ and $\Xi_3$ are defined in* (21).

*Proof.* To start with, we write the expectation as $\mathbb{E}\Xi_2\Xi_3$

$$\mathbb{E}\Xi_2\Xi_3 = \mathbb{E}\underbrace{\mathbf{X}_{\pi^\natural(i)}^\top \mathbf{B}^\natural}_{\boldsymbol{u}^\top}\mathbf{W}^\top \underbrace{\mathbf{X}\left(\mathbf{X}_{\pi^\natural(i)} - \mathbf{X}_j\right)}_{\boldsymbol{p}\in\mathbb{R}^{n\times 1}}\mathbf{W}_i^\top \underbrace{\mathbf{B}^{\natural\top}\mathbf{X}^\top\mathbf{\Pi}^{\natural\top}\mathbf{X}\left(\mathbf{X}_{\pi^\natural(i)} - \mathbf{X}_j\right)}_{\boldsymbol{v}}$$
$$= \mathbb{E}\boldsymbol{u}^\top \mathbf{W}^\top \boldsymbol{p}\mathbf{W}_i^\top \boldsymbol{v} = \mathbb{E}\left\langle\mathbf{W}_i, \boldsymbol{u}^\top \mathbf{W}^\top \boldsymbol{p}\boldsymbol{v}\right\rangle.$$

Exploiting the independence among $\mathbf{X}$ and $\mathbf{W}$, we condition on $\mathbf{X}$ and have

$$\mathbb{E}_{\mathbf{W}}\left\langle\mathbf{W}_i, \boldsymbol{u}^\top \mathbf{W}^\top \boldsymbol{p}\boldsymbol{v}\right\rangle = \mathbb{E}_{\mathbf{W}}\operatorname{Tr}\left(\nabla_{\mathbf{W}_i}\boldsymbol{u}^\top \mathbf{W}^\top \boldsymbol{p}\boldsymbol{v}\right).$$

Note that only the diagonal entries of the Hessian matrix $\nabla_{\mathbf{W}_i}\boldsymbol{u}^\top \mathbf{W}^\top \boldsymbol{p}\boldsymbol{v}$ matters. For an arbitrary index $s$, we can compute the gradient of the $s$-th entry of $\boldsymbol{u}^\top \mathbf{W}^\top \boldsymbol{p}\boldsymbol{v}$ w.r.t. $\mathbf{W}_{i,s}$ as

$$\frac{d}{dW_{i,s}}\left(\boldsymbol{u}^\top \mathbf{W}^\top \boldsymbol{p}\boldsymbol{v}_s\right) = \boldsymbol{v}_s\frac{d}{dW_{i,s}}\left(\boldsymbol{u}^\top \mathbf{W}^\top \boldsymbol{p}\right) = \boldsymbol{v}_s\sum_{t=1}^n\frac{d}{dW_{i,s}}\left(\boldsymbol{p}_t\mathbf{W}_t^\top \boldsymbol{u}\right) = \boldsymbol{v}_s\frac{d}{dW_{i,s}}\boldsymbol{p}_i\mathbf{W}_i^\top \boldsymbol{u} = \boldsymbol{p}_i\boldsymbol{v}_s\boldsymbol{u}_s.$$

Invoking the definitions of $\boldsymbol{p}, \boldsymbol{v}$ and $\boldsymbol{u}$, we have

$$\mathbb{E}_{\mathbf{W},\mathbf{X}}\left\langle\mathbf{W}_i, \boldsymbol{u}^\top \mathbf{W}^\top \boldsymbol{p}\boldsymbol{v}\right\rangle = \mathbb{E}_{\mathbf{X}}\left(\mathbf{X}_{\pi^\natural(i)} - \mathbf{X}_j\right)^\top \mathbf{X}_i\sum_{s=1}^m\left[\mathbf{X}_{\pi^\natural(i)}^\top \left(\mathbf{B}^{\natural\top}\right)_s\left(\mathbf{B}^{\natural\top}\right)_s^\top \mathbf{X}^\top \mathbf{\Pi}^{\natural\top}\mathbf{X}\left(\mathbf{X}_{\pi^\natural(i)} - \mathbf{X}_j\right)\right]$$

$$\overset{\textcircled{1}}{=} \underbrace{\mathbb{E}\left[\left(\mathbf{X}_{\pi^\natural(i)} - \mathbf{X}_j\right)^\top \mathbf{X}_i\mathbf{X}_{\pi^\natural(i)}^\top \mathbf{M}\mathbf{\Sigma}^\top \left(\mathbf{X}_{\pi^\natural(i)} - \mathbf{X}_j\right)\right]}_{\Lambda_1} + \underbrace{\mathbb{E}\left[\left(\mathbf{X}_{\pi^\natural(i)} - \mathbf{X}_j\right)^\top \mathbf{X}_i\mathbf{X}_{\pi^\natural(i)}^\top \mathbf{M}\mathbf{\Delta}^\top \left(\mathbf{X}_{\pi^\natural(i)} - \mathbf{X}_j\right)\right]}_{\Lambda_2},$$

where in $\textcircled{1}$ we use the relation $\sum_{s=1}^m\left(\mathbf{B}^{\natural\top}\right)_s\left(\mathbf{B}^{\natural\top}\right)_s^\top = \mathbf{B}^\natural\mathbf{B}^{\natural\top} = \mathbf{M}$.

For the first term $\Lambda_1$, we obtain

$$\mathbb{E}\Lambda_1 = \underbrace{\mathbb{E}\left(\mathbf{X}_{\pi^\natural(i)}^\top \mathbf{X}_i\mathbf{X}_{\pi^\natural(i)}^\top \mathbf{M}\mathbf{\Sigma}^\top \mathbf{X}_{\pi^\natural(i)}\right)}_{\mathbb{1}_{i=\pi^\natural(i)}\mathbb{E}\|\mathbf{X}_i\|_{\mathrm{F}}^2\mathbf{X}_i^\top \mathbf{M}\mathbf{\Sigma}^\top \mathbf{X}_i} + \underbrace{\mathbb{E}\left(\mathbf{X}_j^\top \mathbf{X}_i\mathbf{X}_{\pi^\natural(i)}^\top \mathbf{M}\mathbf{\Sigma}^\top \mathbf{X}_j\right)}_{\mathbb{1}_{i=\pi^\natural(i)}\mathbb{E}\mathbf{X}_i^\top \mathbf{M}\mathbf{\Sigma}^\top \mathbf{X}_i}$$

$$- \underbrace{\mathbb{E}\left(\mathbf{X}_{\pi^\natural(i)}^\top \mathbf{X}_i\mathbf{X}_{\pi^\natural(i)}^\top \mathbf{M}\mathbf{\Sigma}^\top \mathbf{X}_j\right)}_{\mathbb{1}_{i=j}\mathbb{1}_{i\neq\pi^\natural(i)}\mathbb{E}\mathbf{X}_{\pi^\natural(i)}^\top \mathbf{M}\mathbf{\Sigma}^\top \mathbf{X}_{\pi^\natural(i)}} - \underbrace{\mathbb{E}\left(\mathbf{X}_j^\top \mathbf{X}_i\mathbf{X}_{\pi^\natural(i)}^\top \mathbf{M}\mathbf{\Sigma}^\top \mathbf{X}_{\pi^\natural(i)}\right)}_{p\mathbb{1}_{i=j}\mathbb{1}_{i\neq\pi^\natural(i)}\mathbb{E}\mathbf{X}_{\pi^\natural(i)}^\top \mathbf{M}\mathbf{\Sigma}^\top \mathbf{X}_{\pi^\natural(i)}}$$

$$= (n-h)\left[\mathbb{1}_{i=\pi^\natural(i)}(p+3) - (p+1)\mathbb{1}_{i=j}\mathbb{1}_{i\neq\pi^\natural(i)}\right]\operatorname{Tr}(\mathbf{M}).$$

Then we consider the second term $\Lambda_2$, which can be decomposed further into four sub-terms reading as

$$\mathbb{E}\Lambda_2 = \mathbb{E}\underbrace{\left(\mathbf{X}_{\pi^\natural(i)}^\top \mathbf{X}_i \mathbf{X}_{\pi^\natural(i)}^\top \mathbf{M}\boldsymbol{\Delta}^\top \mathbf{X}_{\pi^\natural(i)}\right)}_{\Lambda_{2,1}} + \mathbb{E}\underbrace{\left(\mathbf{X}_j^\top \mathbf{X}_i \mathbf{X}_{\pi^\natural(i)}^\top \mathbf{M}\boldsymbol{\Delta}^\top \mathbf{X}_j\right)}_{\Lambda_{2,2}}$$

$$- \mathbb{E}\underbrace{\left(\mathbf{X}_{\pi^\natural(i)}^\top \mathbf{X}_i \mathbf{X}_{\pi^\natural(i)}^\top \mathbf{M}\boldsymbol{\Delta}^\top \mathbf{X}_j\right)}_{\Lambda_{2,3}} - \mathbb{E}\underbrace{\left(\mathbf{X}_j^\top \mathbf{X}_i \mathbf{X}_{\pi^\natural(i)}^\top \mathbf{M}\boldsymbol{\Delta}^\top \mathbf{X}_{\pi^\natural(i)}\right)}_{\Lambda_{2,4}}.$$

**Case $(s,s)$:** $i = \pi^\natural(i)$ **and** $j = \pi^\natural(j)$**.** In this case, we have $\boldsymbol{\Delta}$ be

$$\boldsymbol{\Delta}^{(s,s)} = \mathbf{X}_i \mathbf{X}_i^\top + \mathbf{X}_j \mathbf{X}_j^\top.$$

Hence we conclude

$$\mathbb{E}\Lambda_{2,1} = \mathbb{E}\left[\mathbf{X}_i^\top \mathbf{X}_i \mathbf{X}_i^\top \mathbf{M}\left(\mathbf{X}_i \mathbf{X}_i^\top + \mathbf{X}_j \mathbf{X}_j^\top\right)\mathbf{X}_i\right] = \mathbb{E}\|\mathbf{X}_i\|_\mathrm{F}^4 \mathbf{X}_i^\top \mathbf{M}\mathbf{X}_i + \mathbb{E}\|\mathbf{X}_i\|_\mathrm{F}^2 \mathbf{X}_i^\top \mathbf{M}\mathbf{X}_i$$
$$= (p+2)(p+4)\operatorname{Tr}(\mathbf{M}) + (p+2)\operatorname{Tr}(\mathbf{M}),$$

$$\mathbb{E}\Lambda_{2,2} = \mathbb{E}\left[\mathbf{X}_j^\top \mathbf{X}_i \mathbf{X}_i^\top \mathbf{M}\left(\mathbf{X}_i \mathbf{X}_i^\top + \mathbf{X}_j \mathbf{X}_j^\top\right)\mathbf{X}_j\right] = \mathbb{E}\|\mathbf{X}_i\|_\mathrm{F}^2 \mathbf{X}_i^\top \mathbf{M}\mathbf{X}_i + \mathbb{E}\|\mathbf{X}_j\|_\mathrm{F}^2 \mathbf{X}_j^\top \mathbf{M}\mathbf{X}_j$$
$$= 2\mathbb{E}\|\mathbf{X}_i\|_\mathrm{F}^2 \mathbf{X}_i^\top \mathbf{M}\mathbf{X}_i = 2(p+2)\operatorname{Tr}(\mathbf{M}),$$

$$\mathbb{E}\Lambda_{2,3} = \mathbb{E}\left[\mathbf{X}_i^\top \mathbf{X}_i \mathbf{X}_i^\top \mathbf{M}\left(\mathbf{X}_i \mathbf{X}_i^\top + \mathbf{X}_j \mathbf{X}_j^\top\right)\mathbf{X}_j\right] = 0,$$

$$\mathbb{E}\Lambda_{2,4} = \mathbb{E}\left[\mathbf{X}_j^\top \mathbf{X}_i \mathbf{X}_i^\top \mathbf{M}\left(\mathbf{X}_i \mathbf{X}_i^\top + \mathbf{X}_j \mathbf{X}_j^\top\right)\mathbf{X}_i\right] = 0,$$

which suggests $\mathbb{E}\Lambda_2 = (p+2)(p+7)\operatorname{Tr}(\mathbf{M})$.

**Case $(s,d)$:** $i = \pi^\natural(i)$ **and** $j \neq \pi^\natural(j)$**.** First we write $\boldsymbol{\Delta}$ as

$$\boldsymbol{\Delta}^{(s,d)\top} = \mathbf{X}_i \mathbf{X}_i^\top + \mathbf{X}_{\pi^\natural(j)}\mathbf{X}_j^\top + \mathbf{X}_j \mathbf{X}_{\pi^{\natural-1}(j)}^\top.$$

Then we conclude

$$\mathbb{E}\Lambda_{2,1} = \underbrace{\mathbb{E}\left(\mathbf{X}_i^\top \mathbf{X}_i \mathbf{X}_i^\top \mathbf{M}\mathbf{X}_i \mathbf{X}_i^\top \mathbf{X}_i\right)}_{\mathbb{E}\|\mathbf{X}_i\|_\mathrm{F}^4 \mathbf{X}_i^\top \mathbf{M}\mathbf{X}_i} + \underbrace{\mathbb{E}\left(\mathbf{X}_i^\top \mathbf{X}_i \mathbf{X}_i^\top \mathbf{M}\mathbf{X}_{\pi^\natural(j)}\mathbf{X}_j^\top \mathbf{X}_i\right)}_{0}$$
$$+ \underbrace{\mathbb{E}\left(\mathbf{X}_i^\top \mathbf{X}_i \mathbf{X}_i^\top \mathbf{M}\mathbf{X}_j \mathbf{X}_{\pi^{\natural-1}(j)}^\top \mathbf{X}_i\right)}_{0} = \mathbb{E}\|\mathbf{X}_i\|_\mathrm{F}^4 \mathbf{X}_i^\top \mathbf{M}\mathbf{X}_i = (p+2)(p+4)\operatorname{Tr}(\mathbf{M});$$

$$\mathbb{E}\Lambda_{2,2} = \underbrace{\mathbb{E}\left(\mathbf{X}_j^\top \mathbf{X}_i \mathbf{X}_i^\top \mathbf{M}\mathbf{X}_i \mathbf{X}_i^\top \mathbf{X}_j\right)}_{\mathbb{E}\|\mathbf{X}_i\|_\mathrm{F}^2 \mathbf{X}_i^\top \mathbf{M}\mathbf{X}_i} + \underbrace{\mathbb{E}\left(\mathbf{X}_j^\top \mathbf{X}_i \mathbf{X}_i^\top \mathbf{M}\mathbf{X}_{\pi^\natural(j)}\mathbf{X}_j^\top \mathbf{X}_j\right)}_{0}$$
$$+ \underbrace{\mathbb{E}\left(\mathbf{X}_j^\top \mathbf{X}_i \mathbf{X}_i^\top \mathbf{M}\mathbf{X}_j \mathbf{X}_{\pi^{\natural-1}(j)}^\top \mathbf{X}_j\right)}_{0} = \mathbb{E}\|\mathbf{X}_i\|_\mathrm{F}^2 \mathbf{X}_i^\top \mathbf{M}\mathbf{X}_i = (p+2)\operatorname{Tr}(\mathbf{M}),$$

$$\mathbb{E}\Lambda_{2,3} = \mathbb{E}\left[\mathbf{X}_i^\top \mathbf{X}_i \mathbf{X}_i^\top \mathbf{M}\left(\mathbf{X}_i \mathbf{X}_i^\top + \mathbf{X}_{\pi^\natural(j)}\mathbf{X}_j^\top + \mathbf{X}_j \mathbf{X}_{\pi^{\natural-1}(j)}^\top\right)\mathbf{X}_j\right] = 0,$$

$$\mathbb{E}\Lambda_{2,4} = \mathbb{E}\left[\mathbf{X}_j^\top \mathbf{X}_i \mathbf{X}_i^\top \mathbf{M}\left(\mathbf{X}_i \mathbf{X}_i^\top + \mathbf{X}_{\pi^\natural(j)}\mathbf{X}_j^\top + \mathbf{X}_j \mathbf{X}_{\pi^{\natural-1}(j)}^\top\right)\mathbf{X}_i\right] = 0,$$

which suggests $\mathbb{E}\Lambda_2 = (p+2)(p+5)\operatorname{Tr}(\mathbf{M})$.

**Case $(d,s)$:** $i \neq \pi^\natural(i)$ **and** $j = \pi^\natural(j)$**.** In this case, $\boldsymbol{\Delta}$ reduces to

$$\boldsymbol{\Delta}^{(d,s)\top} = \mathbf{X}_{\pi^\natural(i)}\mathbf{X}_i^\top + \mathbf{X}_{\pi^{\natural2}(i)}\mathbf{X}_{\pi^\natural(i)}^\top + \mathbf{X}_j \mathbf{X}_j^\top.$$

Then we obtain

$$\mathbb{E}\Lambda_{2,1} = \underbrace{\mathbb{E}\left(\mathbf{X}_{\pi^\natural(i)}^\top \mathbf{X}_i \mathbf{X}_{\pi^\natural(i)}^\top \mathbf{M}\mathbf{X}_{\pi^\natural(i)}\mathbf{X}_i^\top \mathbf{X}_{\pi^\natural(i)}\right)}_{\mathbb{E}\|\mathbf{X}_i\|_\mathrm{F}^2 \mathbf{X}_i^\top \mathbf{M}\mathbf{X}_i} + \underbrace{\mathbb{E}\left(\mathbf{X}_{\pi^\natural(i)}^\top \mathbf{X}_i \mathbf{X}_{\pi^\natural(i)}^\top \mathbf{M}\mathbf{X}_{\pi^{\natural2}(i)}\mathbf{X}_{\pi^\natural(i)}^\top \mathbf{X}_{\pi^\natural(i)}\right)}_{\mathbb{1}_{i=\pi^{\natural2}(i)}\mathbb{E}\|\mathbf{X}_i\|_\mathrm{F}^2 \mathbf{X}_i^\top \mathbf{M}\mathbf{X}_i}$$
$$+ \underbrace{\mathbb{E}\left(\mathbf{X}_{\pi^\natural(i)}^\top \mathbf{X}_i \mathbf{X}_{\pi^\natural(i)}^\top \mathbf{M}\mathbf{X}_j \mathbf{X}_j^\top \mathbf{X}_{\pi^\natural(i)}\right)}_{0} = \left(1 + \mathbb{1}_{i=\pi^{\natural2}(i)}\right)(p+2)\operatorname{Tr}(\mathbf{M}),$$

$$\mathbb{E}\Lambda_{2,2} = \underbrace{\mathbb{E}\left(\mathbf{X}_j^\top \mathbf{X}_i \mathbf{X}_{\pi^\natural(i)}^\top \mathbf{M}\mathbf{X}_{\pi^\natural(i)}\mathbf{X}_i^\top \mathbf{X}_j\right)}_{\mathbb{1}_{i=j}\mathbb{E}\|\mathbf{X}_i\|_{\mathrm{F}}^4 \operatorname{Tr}(\mathbf{M}) + \mathbb{1}_{i\neq j} p \operatorname{Tr}(\mathbf{M})} + \underbrace{\mathbb{E}\left(\mathbf{X}_j^\top \mathbf{X}_i \mathbf{X}_{\pi^\natural(i)}^\top \mathbf{M}\mathbf{X}_{\pi^{\natural 2}(i)}\mathbf{X}_{\pi^\natural(i)}^\top \mathbf{X}_j\right)}_{\mathbb{1}_{i=\pi^{\natural 2}(i)}\left[\mathbb{1}_{i=j}\mathbb{E}\|\mathbf{X}_i\|_{\mathrm{F}}^2 \mathbf{X}_i^\top \mathbf{M}\mathbf{X}_i + \mathbb{1}_{i\neq j}\operatorname{Tr}(\mathbf{M})\right]}$$

$$+ \underbrace{\mathbb{E}\left(\mathbf{X}_j^\top \mathbf{X}_i \mathbf{X}_{\pi^\natural(i)}^\top \mathbf{M}\mathbf{X}_j \mathbf{X}_j^\top \mathbf{X}_j\right)}_{0} = \left(p + \mathbb{1}_{i=\pi^{\natural 2}(i)}\right)\operatorname{Tr}(\mathbf{M}),$$

$$\mathbb{E}\Lambda_{2,3} = \mathbb{E}\left[\mathbf{X}_{\pi^\natural(i)}^\top \mathbf{X}_i \mathbf{X}_{\pi^\natural(i)}^\top \mathbf{M}\left(\mathbf{X}_{\pi^\natural(i)}\mathbf{X}_i^\top + \mathbf{X}_{\pi^{\natural 2}(i)}\mathbf{X}_{\pi^\natural(i)}^\top + \mathbf{X}_j\mathbf{X}_j^\top\right)\mathbf{X}_j\right] = 0,$$

$$\mathbb{E}\Lambda_{2,4} = \mathbb{E}\left[\mathbf{X}_j^\top \mathbf{X}_i \mathbf{X}_{\pi^\natural(i)}^\top \mathbf{M}\left(\mathbf{X}_{\pi^\natural(i)}\mathbf{X}_i^\top + \mathbf{X}_{\pi^{\natural 2}(i)}\mathbf{X}_{\pi^\natural(i)}^\top + \mathbf{X}_j\mathbf{X}_j^\top\right)\mathbf{X}_{\pi^\natural(i)}\right] = 0,$$

which suggests

$$\mathbb{E}\Lambda_2 = 2(p+1)\operatorname{Tr}(\mathbf{M}) + (p+3)\mathbb{1}_{i=\pi^{\natural 2}(i)}\operatorname{Tr}(\mathbf{M}).$$

**Case $(d,d)$: $i \neq \pi^\natural(i)$ and $j \neq \pi^\natural(j)$.** In this case, $\boldsymbol{\Delta}$ is written as

$$\boldsymbol{\Delta}^{(d,d)\top} = \mathbf{X}_{\pi^\natural(i)}\mathbf{X}_i^\top + \mathbf{X}_{\pi^{\natural 2}(i)}\mathbf{X}_{\pi^\natural(i)}^\top + \mathbf{X}_{\pi^\natural(j)}\mathbf{X}_j^\top + \mathbf{X}_j\mathbf{X}_{\pi^{\natural -1}(j)}^\top.$$

Then we have

$$\mathbb{E}\Lambda_{2,1} = \underbrace{\mathbb{E}\left(\mathbf{X}_{\pi^\natural(i)}^\top \mathbf{X}_i \mathbf{X}_{\pi^\natural(i)}^\top \mathbf{M}\mathbf{X}_{\pi^\natural(i)}\mathbf{X}_i^\top \mathbf{X}_{\pi^\natural(i)}\right)}_{\mathbb{E}\|\mathbf{X}_i\|_{\mathrm{F}}^2 \mathbf{X}_i^\top \mathbf{M}\mathbf{X}_i} + \underbrace{\mathbb{E}\left(\mathbf{X}_{\pi^\natural(i)}^\top \mathbf{X}_i \mathbf{X}_{\pi^\natural(i)}^\top \mathbf{M}\mathbf{X}_{\pi^{\natural 2}(i)}\mathbf{X}_{\pi^\natural(i)}^\top \mathbf{X}_{\pi^\natural(i)}\right)}_{\mathbb{1}_{i=\pi^{\natural 2}(i)}\mathbb{E}\|\mathbf{X}_i\|_{\mathrm{F}}^2 \mathbf{X}_i^\top \mathbf{M}\mathbf{X}_i}$$

$$+ \underbrace{\mathbb{E}\left(\mathbf{X}_{\pi^\natural(i)}^\top \mathbf{X}_i \mathbf{X}_{\pi^\natural(i)}^\top \mathbf{M}\mathbf{X}_{\pi^\natural(j)}\mathbf{X}_j^\top \mathbf{X}_{\pi^\natural(i)}\right)}_{\mathbb{1}_{i=j}\mathbb{E}\|\mathbf{X}_i\|_{\mathrm{F}}^2 \mathbf{X}_i^\top \mathbf{M}\mathbf{X}_i} + \underbrace{\mathbb{E}\left(\mathbf{X}_{\pi^\natural(i)}^\top \mathbf{X}_i \mathbf{X}_{\pi^\natural(i)}^\top \mathbf{M}\mathbf{X}_j \mathbf{X}_{\pi^{\natural -1}(j)}^\top \mathbf{X}_{\pi^\natural(i)}\right)}_{\mathbb{1}_{i=j}\mathbb{1}_{i=\pi^{\natural 2}(i)}\mathbb{E}\|\mathbf{X}_i\|_{\mathrm{F}}^2 \mathbf{X}_i^\top \mathbf{M}\mathbf{X}_i}$$

$$= (p+2)\left[1 + \mathbb{1}_{i=\pi^{\natural 2}(i)} + \mathbb{1}_{i=j} + \mathbb{1}_{i=\pi^{\natural 2}(i)}\mathbb{1}_{i=j}\right]\operatorname{Tr}(\mathbf{M}),$$

$$\mathbb{E}\Lambda_{2,2} = \underbrace{\mathbb{E}\left(\mathbf{X}_j^\top \mathbf{X}_i \mathbf{X}_{\pi^\natural(i)}^\top \mathbf{M}\mathbf{X}_{\pi^\natural(i)}\mathbf{X}_i^\top \mathbf{X}_j\right)}_{\mathbb{1}_{i=j}\mathbb{E}\|\mathbf{X}_i\|_{\mathrm{F}}^4 \operatorname{Tr}(\mathbf{M}) + p\mathbb{1}_{i\neq j}\operatorname{Tr}(\mathbf{M})} + \underbrace{\mathbb{E}\left(\mathbf{X}_j^\top \mathbf{X}_i \mathbf{X}_{\pi^\natural(i)}^\top \mathbf{M}\mathbf{X}_{\pi^{\natural 2}(i)}\mathbf{X}_{\pi^\natural(i)}^\top \mathbf{X}_j\right)}_{\mathbb{1}_{i=\pi^{\natural 2}(i)}\left[\mathbb{1}_{i=j}(p+2)\operatorname{Tr}(\mathbf{M}) + \mathbb{1}_{i\neq j}\operatorname{Tr}(\mathbf{M})\right]}$$

$$+ \underbrace{\mathbb{E}\left(\mathbf{X}_j^\top \mathbf{X}_i \mathbf{X}_{\pi^\natural(i)}^\top \mathbf{M}\mathbf{X}_{\pi^\natural(j)}\mathbf{X}_j^\top \mathbf{X}_j\right)}_{\mathbb{1}_{i=j}\mathbb{E}\|\mathbf{X}_i\|_{\mathrm{F}}^4 \operatorname{Tr}(\mathbf{M})} + \underbrace{\mathbb{E}\left(\mathbf{X}_j^\top \mathbf{X}_i \mathbf{X}_{\pi^\natural(i)}^\top \mathbf{M}\mathbf{X}_j \mathbf{X}_{\pi^{\natural -1}(j)}^\top \mathbf{X}_j\right)}_{\mathbb{1}_{i=j}\mathbb{1}_{i=\pi^{\natural 2}(i)}\mathbb{E}\|\mathbf{X}_i\|_{\mathrm{F}}^2 \mathbf{X}_i^\top \mathbf{M}\mathbf{X}_i}$$

$$= 2\mathbb{1}_{i=j}p(p+2)\operatorname{Tr}(\mathbf{M}) + p\mathbb{1}_{i\neq j}\operatorname{Tr}(\mathbf{M}) + \mathbb{1}_{i=\pi^{\natural 2}(i)}\left[\mathbb{1}_{i=j}2(p+2)\operatorname{Tr}(\mathbf{M}) + \mathbb{1}_{i\neq j}\operatorname{Tr}(\mathbf{M})\right],$$

$$\mathbb{E}\Lambda_{2,3} = \underbrace{\mathbb{E}\left(\mathbf{X}_{\pi^\natural(i)}^\top \mathbf{X}_i \mathbf{X}_{\pi^\natural(i)}^\top \mathbf{M}\mathbf{X}_{\pi^\natural(i)}\mathbf{X}_i^\top \mathbf{X}_j\right)}_{0} + \underbrace{\mathbb{E}\left(\mathbf{X}_{\pi^\natural(i)}^\top \mathbf{X}_i \mathbf{X}_{\pi^\natural(i)}^\top \mathbf{M}\mathbf{X}_{\pi^{\natural 2}(i)}\mathbf{X}_{\pi^\natural(i)}^\top \mathbf{X}_j\right)}_{0}$$

$$+ \underbrace{\mathbb{E}\left(\mathbf{X}_{\pi^\natural(i)}^\top \mathbf{X}_i \mathbf{X}_{\pi^\natural(i)}^\top \mathbf{M}\mathbf{X}_{\pi^\natural(j)}\mathbf{X}_j^\top \mathbf{X}_j\right)}_{p\mathbb{1}_{i=\pi^\natural(j)}\operatorname{Tr}(\mathbf{M})} + \underbrace{\mathbb{E}\left(\mathbf{X}_{\pi^\natural(i)}^\top \mathbf{X}_i \mathbf{X}_{\pi^\natural(i)}^\top \mathbf{M}\mathbf{X}_j \mathbf{X}_{\pi^{\natural -1}(j)}^\top \mathbf{X}_j\right)}_{0},$$

$$\mathbb{E}\Lambda_{2,4} = \underbrace{\mathbb{E}\left(\mathbf{X}_j^\top \mathbf{X}_i \mathbf{X}_{\pi^\natural(i)}^\top \mathbf{M}\mathbf{X}_{\pi^\natural(i)}\mathbf{X}_i^\top \mathbf{X}_{\pi^\natural(i)}\right)}_{0} + \underbrace{\mathbb{E}\left(\mathbf{X}_j^\top \mathbf{X}_i \mathbf{X}_{\pi^\natural(i)}^\top \mathbf{M}\mathbf{X}_{\pi^{\natural 2}(i)}\mathbf{X}_{\pi^\natural(i)}^\top \mathbf{X}_{\pi^\natural(i)}\right)}_{0}$$

$$+ \underbrace{\mathbb{E}\left(\mathbf{X}_j^\top \mathbf{X}_i \mathbf{X}_{\pi^\natural(i)}^\top \mathbf{M}\mathbf{X}_{\pi^\natural(j)}\mathbf{X}_j^\top \mathbf{X}_{\pi^\natural(i)}\right)}_{\mathbb{1}_{i=\pi^\natural(j)}\operatorname{Tr}(\mathbf{M})} + \underbrace{\mathbb{E}\left(\mathbf{X}_j^\top \mathbf{X}_i \mathbf{X}_{\pi^\natural(i)}^\top \mathbf{M}\mathbf{X}_j \mathbf{X}_{\pi^{\natural -1}(j)}^\top \mathbf{X}_{\pi^\natural(i)}\right)}_{0},$$

which gives

$$\mathbb{E}\Lambda_2 = (p+1)\left[2 + 2(p+1)\mathbb{1}_{i=j} + \mathbb{1}_{i=\pi^\natural(j)}\right]\operatorname{Tr}(\mathbf{M}) + \mathbb{1}_{i=\pi^{\natural 2}(i)}\left[p + 3 + (3p+5)\mathbb{1}_{i=j}\right]\operatorname{Tr}(\mathbf{M}).$$

$\square$

### C.2.3 SUPPORTING LEMMAS

First, we study the higher order expectations of Gaussian random vectors' inner product, which hopefully will serve independent interests.

**Lemma 11.** *Assume $\boldsymbol{x} \in \mathbb{R}^p$ and $\boldsymbol{y} \in \mathbb{R}^p$ are Gaussian distributed random vectors whose entries follow the i.i.d. standard normal distribution, then we have*

$$\mathbb{E} \operatorname{Tr} \left( \boldsymbol{y}\boldsymbol{y}^\top \boldsymbol{x}\boldsymbol{x}^\top \mathbf{M} \right) = \operatorname{Tr}(\mathbf{M}), \tag{63}$$

$$\mathbb{E} \operatorname{Tr} \left( \boldsymbol{y}\boldsymbol{y}^\top \boldsymbol{x}^\top \mathbf{M}\boldsymbol{x} \right) = \mathbb{E}\|\boldsymbol{y}\|_2^2 \operatorname{Tr} \left( \boldsymbol{x}^\top \mathbf{M}\boldsymbol{x} \right) = p \operatorname{Tr}(\mathbf{M}), \tag{64}$$

$$\mathbb{E} \left( \boldsymbol{x}^\top \mathbf{M}\boldsymbol{x} \right)^2 = [\operatorname{Tr}(\mathbf{M})]^2 + \operatorname{Tr}(\mathbf{M}\mathbf{M}) + \operatorname{Tr} \left( \mathbf{M}^\top \mathbf{M} \right), \tag{65}$$

$$\mathbb{E}\|\boldsymbol{x}\|_2^2 (\boldsymbol{x}^\top \mathbf{M}\boldsymbol{x}) = (p+2) \operatorname{Tr}(\mathbf{M}), \tag{66}$$

$$\mathbb{E}\|\boldsymbol{x}\|_2^4 (\boldsymbol{x}^\top \mathbf{M}\boldsymbol{x}) = (p+2)(p+4) \operatorname{Tr}(\mathbf{M}), \tag{67}$$

$$\mathbb{E}\|\boldsymbol{x}\|_2^2 \left( \boldsymbol{x}^\top \mathbf{M}\boldsymbol{x} \right)^2 = (p+4) \left[ (\operatorname{Tr}(\mathbf{M}))^2 + \operatorname{Tr}(\mathbf{M}\mathbf{M}) + \operatorname{Tr} \left( \mathbf{M}^\top \mathbf{M} \right) \right], \tag{68}$$

$$\mathbb{E}\|\boldsymbol{x}\|_2^4 \left( \boldsymbol{x}^\top \mathbf{M}\boldsymbol{x} \right)^2 = (p+4)(p+6) \left[ (\operatorname{Tr}(\mathbf{M}))^2 + \operatorname{Tr}(\mathbf{M}\mathbf{M}) + \operatorname{Tr} \left( \mathbf{M}^\top \mathbf{M} \right) \right], \tag{69}$$

$$\mathbb{E}(\boldsymbol{x}^\top \boldsymbol{y})^2 \boldsymbol{y}^\top \mathbf{M}_1 \boldsymbol{x}\boldsymbol{x}^\top \mathbf{M}_2 \boldsymbol{y} = 2 \operatorname{Tr}(\mathbf{M}_1) \operatorname{Tr}(\mathbf{M}_2) + (p+4) \operatorname{Tr}(\mathbf{M}_1 \mathbf{M}_2) + 2 \operatorname{Tr}(\mathbf{M}_1 \mathbf{M}_2^\top), \tag{70}$$

*where $\mathbf{M} \in \mathbb{R}^{p \times p}$ is a fixed matrix.*

**Remark 5.** *If we assume $\mathbf{M} = \mathbf{I}_{p \times p}$, we can get $\mathbb{E}\|\boldsymbol{x}\|_2^4 = p(p+2)$, $\mathbb{E}\|\boldsymbol{x}\|_2^6 = p(p+2)(p+4)$, and $\mathbb{E}\|\boldsymbol{x}\|_2^8 = p(p+2)(p+4)(p+6)$.*

*Proof.* This lemma is proved by iteratively applying the Wick's theorem in Theorem 4, Stein's lemma in Lemma 20, and Lemma 19.

- **Proof of** (63) **and** (64). The proof can be conducted easily with the property such that $\operatorname{Tr}(\boldsymbol{u}\boldsymbol{v}^\top) = \boldsymbol{u}^\top \boldsymbol{v} = \operatorname{Tr}(\boldsymbol{v}\boldsymbol{u}^\top)$ holds for arbitrary vectors $\boldsymbol{u}$ and $\boldsymbol{v}$.

- **Proof of** (65). This property is a direct consequence of Neudecker & Wansbeek (1987) (Equation (3.2)), which is attached in Lemma 19 for the sake of self-containing.

- **Proof of** (66). Invoking the Stein's lemma, we have

$$\mathbb{E}\|\boldsymbol{x}\|_2^2 (\boldsymbol{x}^\top \mathbf{M}\boldsymbol{x}) = \mathbb{E} \left[ \nabla_{\boldsymbol{x}} (\boldsymbol{x}^\top \mathbf{M}\boldsymbol{x}) \boldsymbol{x} \right].$$

Then our goal transforms to computing the trace of the Hessian matrix $\operatorname{Tr} \left[ \nabla_{\boldsymbol{x}} \operatorname{Tr}(\boldsymbol{x}^\top \mathbf{M}\boldsymbol{x}) \boldsymbol{x} \right]$. For the $i$-th entry of the gradient, we have

$$\frac{d}{dx_i} \boldsymbol{x}^\top \mathbf{M}\boldsymbol{x} = \langle \mathbf{M}_i, \boldsymbol{x} \rangle + \langle (\mathbf{M}^\top)_i, \boldsymbol{x} \rangle,$$

where $\mathbf{M}_i$ is the $i$-th row (or column) of $\mathbf{M}$. Then we obtain

$$\frac{d}{dx_i} \left[ x_i \operatorname{Tr}(\boldsymbol{x}^\top \mathbf{M}\boldsymbol{x}) \right] = \boldsymbol{x}^\top \mathbf{M}\boldsymbol{x} + x_i \left[ \langle \mathbf{M}_i, \boldsymbol{x} \rangle + \langle (\mathbf{M}^\top)_i, \boldsymbol{x} \rangle \right],$$

and hence

$$\mathbb{E}\|\boldsymbol{x}\|_2^2 (\boldsymbol{x}^\top \mathbf{M}\boldsymbol{x}) = \sum_{i=1}^p \mathbb{E}(\boldsymbol{x}^\top \mathbf{M}\boldsymbol{x}) + \sum_{i=1}^p \mathbb{E} \left[ x_i \left( \langle \mathbf{M}_i, \boldsymbol{x} \rangle + \langle (\mathbf{M}^\top)_i, \boldsymbol{x} \rangle \right) \right]$$

$$= p \operatorname{Tr}(\mathbf{M}) + 2 \sum_i \mathbf{M}_{ii} = (p+2) \operatorname{Tr}(\mathbf{M}).$$

- **Proof of** (67). Following the same strategy as in proving (66), we have

$$\mathbb{E}\|\boldsymbol{x}\|_F^4 (\boldsymbol{x}^\top \mathbf{M}\boldsymbol{x}) = \mathbb{E} \left[ \nabla_{\boldsymbol{x}} \|\boldsymbol{x}\|_2^2 (\boldsymbol{x}^\top \mathbf{M}\boldsymbol{x}) \boldsymbol{x} \right].$$

Then our goal transforms to computing the trace of the Hessian matrix $\mathrm{Tr}\left[\nabla_{\boldsymbol{x}}\|\boldsymbol{x}\|_2^2(\boldsymbol{x}^\top\mathbf{M}\boldsymbol{x})\boldsymbol{x}\right]$. For the $i$-th entry of the gradient, we obtain

$$
\frac{d}{dx_i}\left[x_i\|\boldsymbol{x}\|_2^2(\boldsymbol{x}^\top\mathbf{M}\boldsymbol{x})\right] = \|\boldsymbol{x}\|_2^2\cdot(\boldsymbol{x}^\top\mathbf{M}\boldsymbol{x}) + x_i\frac{d}{dx_i}\left[\|\boldsymbol{x}\|_2^2(\boldsymbol{x}^\top\mathbf{M}\boldsymbol{x})\right]
$$
$$
= \|\boldsymbol{x}\|_2^2\cdot(\boldsymbol{x}^\top\mathbf{M}\boldsymbol{x}) + 2x_i^2(\boldsymbol{x}^\top\mathbf{M}\boldsymbol{x}) + x_i\|\boldsymbol{x}\|_2^2\left[\langle\mathbf{M}_i,\boldsymbol{x}\rangle + \langle(\mathbf{M}^\top)_i,\boldsymbol{x}\rangle\right],
$$

whose expectation reads as

$$
\mathbb{E}\|\boldsymbol{x}\|_2^2\cdot(\boldsymbol{x}^\top\mathbf{M}\boldsymbol{x}) + 2\mathbb{E}\left[x_i^2(\boldsymbol{x}^\top\mathbf{M}\boldsymbol{x})\right] + \mathbb{E}x_i\|\boldsymbol{x}\|_2^2\left[\langle\mathbf{M}_i,\boldsymbol{x}\rangle + \langle(\mathbf{M}^\top)_i,\boldsymbol{x}\rangle\right]
$$
$$
= (p+2)\mathrm{Tr}(\mathbf{M}) + 2\mathbb{E}\left[x_i^2 M_{ii}x_i^2 + x_i^2\left(\sum_{j\neq i}M_{jj}x_j^2\right)\right] + 2M_{ii}\mathbb{E}\left[x_i^2\|\boldsymbol{x}\|_2^2\right]
$$
$$
= (p+2)\mathrm{Tr}(\mathbf{M}) + 2M_{ii}\left(\mathbb{E}x_i^4\right) + 2\sum_{j\neq i}M_{jj}(\mathbb{E}x_i^2)(\mathbb{E}x_j^2) + 2M_{ii}\left[\mathbb{E}(x_i^4) + \sum_{j\neq i}(\mathbb{E}x_i^2)(\mathbb{E}x_j^2)\right]
$$
$$
= (p+2)\mathrm{Tr}(\mathbf{M}) + 6M_{ii} + 2\sum_{j\neq i}M_{jj} + 2M_{ii}(3+p-1)
$$
$$
= (p+4)\mathrm{Tr}(\mathbf{M}) + 2(p+4)M_{ii}.
$$

Then we conclude

$$
\mathbb{E}\,\mathrm{Tr}\left[\nabla_{\boldsymbol{x}}\|\boldsymbol{x}\|_2^2(\boldsymbol{x}^\top\mathbf{M}\boldsymbol{x})\boldsymbol{x}\right] = p(p+2)\mathrm{Tr}(\mathbf{M}) + 2(p+4)\sum_i M_{ii} + 2p\,\mathrm{Tr}(\mathbf{M})
$$
$$
= (p+2)(p+4)\mathrm{Tr}(\mathbf{M}).
$$

- **Proof of** (68). Invoking the Stein's lemma, we have

$$
\mathbb{E}\|\boldsymbol{x}\|_2^2\left(\boldsymbol{x}^\top\mathbf{M}\boldsymbol{x}\right)^2 = \sum_i\frac{d}{dx_i}\left[x_i\left(\boldsymbol{x}^\top\mathbf{M}\boldsymbol{x}\right)^2\right] = p\left(\boldsymbol{x}^\top\mathbf{M}\boldsymbol{x}\right)^2 + 4\sum_i x_i\left(\boldsymbol{x}^\top\mathbf{M}\boldsymbol{x}\right)\left\langle\mathbf{M}_i^{(\text{sym})},\boldsymbol{x}\right\rangle.
$$

The proof is then completed by invoking Lemma 12.

- **Proof of** (69). Following the same strategy as in proving (68), we consider the $i$-th gradient w.r.t $x_i$, which can be written as

$$
\mathbb{E}\|\boldsymbol{x}\|_2^4\left(\boldsymbol{x}^\top\mathbf{M}\boldsymbol{x}\right)^2 = \sum_i\frac{d}{dx_i}\left[x_i\|\boldsymbol{x}\|_2^2\left(\boldsymbol{x}^\top\mathbf{M}\boldsymbol{x}\right)^2\right]
$$
$$
= \sum_i\|\boldsymbol{x}\|_2^2\left(\boldsymbol{x}^\top\mathbf{M}\boldsymbol{x}\right)^2 + 2\sum_i x_i^2\left(\boldsymbol{x}^\top\mathbf{M}\boldsymbol{x}\right)^2 + 4\sum_i x_i\|\boldsymbol{x}\|_2^2(\boldsymbol{x}^\top\mathbf{M}\boldsymbol{x})\left\langle\mathbf{M}^{(\text{sym})},\boldsymbol{x}\right\rangle
$$
$$
= \sum_i\|\boldsymbol{x}\|_2^2\left(\boldsymbol{x}^\top\mathbf{M}\boldsymbol{x}\right)^2 + 2\sum_i\mathbb{E}\frac{d}{dx_i}\left[x_i\left(\boldsymbol{x}^\top\mathbf{M}\boldsymbol{x}\right)^2\right] + 4\sum_i\mathbb{E}\frac{d}{dx_i}\left[\|\boldsymbol{x}\|_2^2(\boldsymbol{x}^\top\mathbf{M}\boldsymbol{x})\left\langle\mathbf{M}_i^{(\text{sym})},\boldsymbol{x}\right\rangle\right].
$$
$$\tag{71}$$

Noticing the following relations

$$
\frac{d}{dx_i}\left[x_i\left(\boldsymbol{x}^\top\mathbf{M}\boldsymbol{x}\right)^2\right] = \left(\boldsymbol{x}^\top\mathbf{M}\boldsymbol{x}\right)^2 + 4x_i\left(\boldsymbol{x}^\top\mathbf{M}\boldsymbol{x}\right)\left\langle\mathbf{M}_i^{(\text{sym})},\boldsymbol{x}\right\rangle,\tag{72}
$$
$$
\frac{d}{dx_i}\left[\|\boldsymbol{x}\|_2^2\mathrm{Tr}(\boldsymbol{x}^\top\mathbf{M}\boldsymbol{x})\left\langle\mathbf{M}_i^{(\text{sym})},\boldsymbol{x}\right\rangle\right] = 2x_i\left(\boldsymbol{x}^\top\mathbf{M}\boldsymbol{x}\right)\left\langle\mathbf{M}_i^{(\text{sym})},\boldsymbol{x}\right\rangle
$$
$$
+ 2\|\boldsymbol{x}\|_2^2\left\langle\mathbf{M}_i^{(\text{sym})},\boldsymbol{x}\right\rangle^2 + M_{ii}\|\boldsymbol{x}\|_2^2\left(\boldsymbol{x}^\top\mathbf{M}\boldsymbol{x}\right),\tag{73}
$$

we can conclude the proof by combining (68), (71), (72), (73), and Lemma 12.

- **Proof of** (70). Due to the independence between $\boldsymbol{x}$ and $\boldsymbol{y}$, we first condition on $\boldsymbol{x}$ and have

$$\mathbb{E}(\boldsymbol{x}^\top \boldsymbol{y})^2 \boldsymbol{y}^\top \mathbf{M}_1 \boldsymbol{x} \boldsymbol{x}^\top \mathbf{M}_2 \boldsymbol{y} = \mathbb{E}_{\boldsymbol{x}} \mathbb{E}_{\boldsymbol{y}} \boldsymbol{y}^\top \boldsymbol{x} \boldsymbol{x}^\top \boldsymbol{y} \boldsymbol{y}^\top \mathbf{M}_1 \boldsymbol{x} \boldsymbol{x}^\top \mathbf{M}_2 \boldsymbol{y}$$

$$\stackrel{\textcircled{1}}{=} \mathbb{E}_{\boldsymbol{x}} \operatorname{Tr}\left(\boldsymbol{x}\boldsymbol{x}^\top\right) \operatorname{Tr}\left(\mathbf{M}_1 \boldsymbol{x}\boldsymbol{x}^\top \mathbf{M}_2\right) + \mathbb{E}_{\boldsymbol{x}} \operatorname{Tr}\left(\boldsymbol{x}\boldsymbol{x}^\top \mathbf{M}_1 \boldsymbol{x}\boldsymbol{x}^\top \mathbf{M}_2\right) + \mathbb{E}_{\boldsymbol{x}} \operatorname{Tr}\left(\boldsymbol{x}\boldsymbol{x}^\top \mathbf{M}_2^\top \boldsymbol{x}\boldsymbol{x} \mathbf{M}_1^\top\right)$$

$$= \mathbb{E}_{\boldsymbol{x}} \|\boldsymbol{x}\|_2^2 \boldsymbol{x}^\top \mathbf{M}_2 \mathbf{M}_1 \boldsymbol{x} + \mathbb{E}_{\boldsymbol{x}} \boldsymbol{x}^\top \mathbf{M}_1 \boldsymbol{x} \boldsymbol{x}^\top \mathbf{M}_2 \boldsymbol{x} + \mathbb{E}_{\boldsymbol{x}} \boldsymbol{x}^\top \mathbf{M}_1^\top \boldsymbol{x} \boldsymbol{x}^\top \mathbf{M}_2^\top \boldsymbol{x}$$

$$\stackrel{\textcircled{2}}{=} 2 \operatorname{Tr}(\mathbf{M}_1) \operatorname{Tr}(\mathbf{M}_2) + (p+4) \operatorname{Tr}(\mathbf{M}_1 \mathbf{M}_2) + 2 \operatorname{Tr}(\mathbf{M}_1 \mathbf{M}_2^\top),$$

where in ① and ② we both use Lemma 19.

$\square$

**Lemma 12.** *For a fixed matrix $\mathbf{M} \in \mathbb{R}^{p \times p}$, we associate it with a symmetric matrix $\mathbf{M}^{(\mathsf{sym})}$ defined as ${}^{(\mathbf{M}+\mathbf{M}^\top)}/_2$. Consider the Gaussian distributed random vector $\boldsymbol{x} \sim \mathsf{N}(\boldsymbol{0}, \mathbf{I})$, we have*

$$\mathbb{E} \sum_i x_i (\boldsymbol{x}^\top \mathbf{M} \boldsymbol{x}) \left\langle \mathbf{M}_i^{(\mathsf{sym})}, \boldsymbol{x} \right\rangle = (\operatorname{Tr}(\mathbf{M}))^2 + \|\mathbf{M}\|_{\mathrm{F}}^2 + \operatorname{Tr}(\mathbf{M}\mathbf{M}).$$

*Proof.* This lemma is a direct application of Wick's theorem, which is completed by showing

$$\mathbb{E} x_i \left(\boldsymbol{x}^\top \mathbf{M} \boldsymbol{x}\right) \left\langle \mathbf{M}_i^{(\mathsf{sym})}, \boldsymbol{x} \right\rangle = \mathbb{E} \sum_j \sum_{\ell_1, \ell_2} M_{i,j}^{(\mathsf{sym})} M_{\ell_1, \ell_2} x_i x_j x_{\ell_1} x_{\ell_2}$$

$$= \underbrace{\mathbb{E} \sum_j \sum_{\ell_1, \ell_2} \mathbb{1}_{\ell_1 = i} \mathbb{1}_{\ell_2 = j} M_{i,j}^{(\mathsf{sym})} M_{\ell_1, \ell_2} x_i x_j x_{\ell_1} x_{\ell_2}}_{\sum_j M_{i,j}^{(\mathsf{sym})} M_{i,j}} + \underbrace{\mathbb{E} \sum_j \sum_{\ell_1, \ell_2} \mathbb{1}_{\ell_2 = i} \mathbb{1}_{\ell_1 = j} M_{i,j}^{(\mathsf{sym})} M_{\ell_1, \ell_2} x_i x_j x_{\ell_1} x_{\ell_2}}_{\sum_j M_{i,j}^{(\mathsf{sym})} M_{j,i}}$$

$$+ \underbrace{\mathbb{E} \sum_j \sum_{\ell_1, \ell_2} \mathbb{1}_{i=j} \mathbb{1}_{\ell_1 = \ell_2} M_{i,j}^{(\mathsf{sym})} M_{\ell_1, \ell_2} x_i x_j x_{\ell_1} x_{\ell_2}}_{\sum_\ell M_{i,i}^{(\mathsf{sym})} M_{\ell,\ell}} = \sum_j 2 \left[M_{i,j}^{(\mathsf{sym})}\right]^2 + M_{i,i} \operatorname{Tr}(\mathbf{M}),$$

where $\mathbf{M}^{(\mathsf{sym})}$ is defined as $\left(\mathbf{M} + \mathbf{M}^\top\right)/2$. $\square$

Then we study the properties of $\boldsymbol{\Sigma}$, which is defined as $\mathbf{X}^\top \boldsymbol{\Pi}^\natural \mathbf{X} - \boldsymbol{\Delta}$.

**Lemma 13.** *For a fixed matrix $\mathbf{M}$, we have*

$$\mathbb{E} \operatorname{Tr}\left(\boldsymbol{\Sigma} \mathbf{M} \boldsymbol{\Sigma}^\top\right) = n^2 \left[\frac{p}{n} + \left(1 - \frac{h}{n}\right)^2 + o(1)\right] \operatorname{Tr}(\mathbf{M}),$$

*where matrix $\boldsymbol{\Sigma}$ is defined in* (22).

*Proof.* We conclude the proof by showing

$$\mathbb{E} \operatorname{Tr}\left(\boldsymbol{\Sigma} \mathbf{M} \boldsymbol{\Sigma}^\top\right) \stackrel{\textcircled{1}}{=} \sum_{\ell_1, \ell_2 \in \mathcal{S}} \mathbb{E} \operatorname{Tr}\left[\mathbf{X}_{\ell_1} \mathbf{X}_{\ell_1}^\top \mathbf{M} \mathbf{X}_{\ell_2} \mathbf{X}_{\ell_2}^\top\right] + \sum_{\ell_1, \ell_2 \in \mathcal{D}} \mathbb{E} \operatorname{Tr}\left[\mathbf{X}_{\ell_1} \mathbf{X}_{\pi^\natural(\ell_1)}^\top \mathbf{M} \mathbf{X}_{\pi^\natural(\ell_2)} \mathbf{X}_{\ell_2}^\top\right]$$

$$= \sum_{\ell \in \mathcal{S}} \mathbb{E} \operatorname{Tr}\left(\mathbf{X}_\ell \mathbf{X}_\ell^\top \mathbf{M} \mathbf{X}_\ell \mathbf{X}_\ell^\top\right) + \sum_{\ell_1, \ell_2 \in \mathcal{S}, \ell_1 \neq \ell_2} \mathbb{E} \operatorname{Tr}\left[\mathbf{X}_{\ell_1} \mathbf{X}_{\ell_1}^\top \mathbf{M} \mathbf{X}_{\ell_2} \mathbf{X}_{\ell_2}^\top\right]$$

$$+ \sum_{\ell \in \mathcal{D}} \underbrace{\mathbb{E} \operatorname{Tr}\left[\mathbf{X}_\ell \mathbf{X}_{\pi^\natural(\ell)}^\top \mathbf{M} \mathbf{X}_{\pi^\natural(\ell)} \mathbf{X}_\ell^\top\right]}_{p \operatorname{Tr}(\mathbf{M})} + \sum_{(\ell_1, \ell_2) \in \mathcal{D}_{\mathrm{pair}}} \underbrace{\mathbb{E} \operatorname{Tr}\left[\mathbf{X}_{\ell_1} \mathbf{X}_{\ell_2}^\top \mathbf{M} \mathbf{X}_{\ell_1} \mathbf{X}_{\ell_2}^\top\right]}_{\operatorname{Tr}(\mathbf{M})}$$

$$= (n-h)(p+2) \operatorname{Tr}(\mathbf{M}) + (n-h)(n-h-1) \operatorname{Tr}(\mathbf{M}) + hp \operatorname{Tr}(\mathbf{M}) + |\mathcal{D}_{\mathrm{pair}}| \operatorname{Tr}(\mathbf{M})$$

$$\stackrel{\textcircled{2}}{=} n^2 \left[\frac{p}{n} + \left(1 - \frac{h}{n}\right)^2 + o(1)\right] \operatorname{Tr}(\mathbf{M}),$$

where ① is due to the definitions of index sets $\mathcal{S}$ and $\mathcal{D}$ (Equation (27) and in Equation (28)), and ② is because $|\mathcal{D}_{\mathrm{pair}}| \leq h$. $\square$

**Lemma 14.** *For a fixed matrix* $\mathbf{M}$, *we have*

$$\mathbb{E}\,\mathrm{Tr}(\mathbf{\Sigma}\mathbf{M}\mathbf{\Sigma}\mathbf{M}) = (n - h + |\mathcal{D}_{\mathrm{pair}}|)\left[\mathrm{Tr}\left(\mathbf{M}\right)\right]^2 + (n - h)^2\,\mathrm{Tr}\left(\mathbf{M}\mathbf{M}\right) + n\,\mathrm{Tr}\left(\mathbf{M}\mathbf{M}^\top\right)$$

*Proof.* Following the same strategy as in proving Lemma 13, we complete the proof by showing

$$\mathbb{E}\,\mathrm{Tr}(\mathbf{\Sigma}\mathbf{M}\mathbf{\Sigma}\mathbf{M}) = \sum_{\ell = \pi^\natural(\ell)} \mathbb{E}\,\mathrm{Tr}\left(\mathbf{X}_\ell^\top \mathbf{M}\mathbf{X}_\ell \mathbf{X}_\ell^\top \mathbf{M}\mathbf{X}_\ell\right) + \sum_{\ell_1, \ell_2 \in \mathcal{S}, \ell_1 \neq \ell_2} \mathbb{E}\,\mathrm{Tr}\left(\mathbf{X}_{\ell_1}\mathbf{X}_{\ell_1}^\top \mathbf{M}\mathbf{X}_{\ell_2}\mathbf{X}_{\ell_2}^\top \mathbf{M}\right)$$

$$+ \sum_{\ell \in \mathcal{D}} \mathbb{E}\,\mathrm{Tr}\left(\mathbf{X}_\ell \mathbf{X}_{\pi^\natural(\ell)}^\top \mathbf{M}\mathbf{X}_\ell \mathbf{X}_{\pi^\natural(\ell)}^\top \mathbf{M}\right) + \sum_{\ell \in \mathcal{D}_{\mathrm{pair}}} \mathbb{E}\,\mathrm{Tr}\left(\mathbf{X}_\ell \mathbf{X}_{\pi^\natural(\ell)}^\top \mathbf{M}\mathbf{X}_{\pi^\natural(\ell)}\mathbf{X}_\ell^\top \mathbf{M}\right)$$

$$= (n - h + |\mathcal{D}_{\mathrm{pair}}|)\left[\mathrm{Tr}\left(\mathbf{M}\right)\right]^2 + (n - h)^2\,\mathrm{Tr}\left(\mathbf{M}\mathbf{M}\right) + n\,\mathrm{Tr}\left(\mathbf{M}\mathbf{M}^\top\right).$$

$\square$

**Lemma 15.** *For a fixed matrix* $\mathbf{M}$, *we have*

$$\mathbb{E}\left[\mathrm{Tr}(\mathbf{\Sigma}\mathbf{M})\right]^2 = (n - h)^2\left[\mathrm{Tr}(\mathbf{M})\right]^2 + n\,\mathrm{Tr}\left(\mathbf{M}^\top\mathbf{M}\right) + (n - h + |\mathcal{D}_{\mathrm{pair}}|)\,\mathrm{Tr}(\mathbf{M}\mathbf{M}).$$

*Proof.* We complete the proof by showing

$$\mathbb{E}\left(\mathrm{Tr}(\mathbf{\Sigma}\mathbf{M})\right)^2 = \sum_{\ell \in \mathcal{S}} \mathbb{E}\left(\mathbf{X}_\ell^\top \mathbf{M}\mathbf{X}_\ell\right)^2 + \sum_{\ell_1, \ell_2 \in \mathcal{S}, \ell_1 \neq \ell_2} (\mathrm{Tr}(\mathbf{M}))^2 + \sum_{\ell \in \mathcal{D}} \underbrace{\mathbb{E}\mathbf{X}_{\pi^\natural(\ell)}^\top \mathbf{M}\mathbf{X}_\ell \mathbf{X}_\ell^\top \mathbf{M}^\top \mathbf{X}_{\pi^\natural(\ell)}}_{\mathrm{Tr}(\mathbf{M}^\top\mathbf{M})}$$

$$+ \sum_{\ell \in \mathcal{D}_{\mathrm{pair}}} \underbrace{\mathbb{E}\mathbf{X}_{\pi^\natural(\ell)}^\top \mathbf{M}\mathbf{X}_\ell \mathbf{X}_\ell^\top \mathbf{M}\mathbf{X}_{\pi^\natural(\ell)}}_{\mathrm{Tr}(\mathbf{M}\mathbf{M})}$$

$$= (n - h)^2\left[\mathrm{Tr}(\mathbf{M})\right]^2 + n\,\mathrm{Tr}\left(\mathbf{M}^\top\mathbf{M}\right) + (n - h + |\mathcal{D}_{\mathrm{pair}}|)\,\mathrm{Tr}(\mathbf{M}\mathbf{M}).$$

$\square$

**Lemma 16.** *For a fixed matrix* $\mathbf{M}$, *we have*

$$\mathbb{E}\sum_{\ell = \pi^\natural(\ell)} \mathbf{X}_{\pi^\natural(i)}^\top \mathbf{M}\mathbf{X}_\ell \mathbf{X}_\ell^\top \mathbf{X}_{\pi^\natural(i)} = (n - h)\,\mathrm{Tr}(\mathbf{M}) + (p + 1)\mathbb{1}_{i = \pi^\natural(i)}\,\mathrm{Tr}(\mathbf{M}).$$

*Proof.* Provided that $i = \pi^\natural(i)$, we have

$$\mathbb{E}\Xi_{1,1} = \mathbb{E}\mathbf{X}_i^\top \mathbf{M}\mathbf{X}_i \mathbf{X}_i^\top \mathbf{X}_i + \sum_{\ell \neq i,\ \ell = \pi^\natural(\ell)} \mathbb{E}\mathbf{X}_i^\top \mathbf{M}\mathbf{X}_\ell \mathbf{X}_\ell^\top \mathbf{X}_i$$

$$= (p + 2)\,\mathrm{Tr}(\mathbf{M}) + \sum_{\ell \neq i,\ \ell = \pi^\natural(\ell)} \mathrm{Tr}(\mathbf{M}) = (n - h + p + 1)\,\mathrm{Tr}(\mathbf{M})\mathbb{1}_{i = \pi^\natural(i)}. \tag{74}$$

Provided that $i \neq \pi^\natural(i)$, we have

$$\mathbb{E}\Xi_{1,1} = \sum_{\ell = \pi^\natural(\ell)} \mathbb{E}\mathbf{X}_{\pi^\natural(i)}^\top \mathbf{M}\mathbf{X}_\ell \mathbf{X}_\ell^\top \mathbf{X}_{\pi^\natural(i)} = (n - h)\,\mathrm{Tr}(\mathbf{M})\mathbb{1}_{i \neq \pi^\natural(i)}. \tag{75}$$

Combining (74) and (75) then completes the proof. $\square$

**Lemma 17.** *For a fixed* $\mathbf{M}$, *we have*

$$\mathbb{E}\sum_\ell \mathbf{X}_{\pi^\natural(i)}^\top \mathbf{M}\mathbf{X}_{\pi^\natural(\ell)}\mathbf{X}_\ell^\top \mathbf{X}_j = \left(p\mathbb{1}_{i = j}\mathbb{1}_{i \neq \pi^\natural(i)} + \mathbb{1}_{j = \pi^{\natural 2}(i)}\right)\mathrm{Tr}(\mathbf{M}).$$

We omit its proof as it is a direct application of Wick's theorem (Theorem 4).

**Lemma 18.** *We have*

$$\mathbb{E}\mathbb{1}(i = \pi^\natural(i)) = \frac{n - h}{n}(1 + o_{\mathrm{P}}(1)), \qquad \mathbb{E}\mathbb{1}_{i = j} = \frac{h}{n^2}(1 + o_{\mathrm{P}}(1)),$$

$$\mathbb{E}\mathbb{1}_{j = \pi^{\natural 2}(i)} = \frac{h}{n^2}(1 + o_{\mathrm{P}}(1)), \qquad \mathbb{E}\mathbb{1}_{i = j}\mathbb{1}_{i = \pi^{\natural 2}(i)} = \frac{|\mathcal{D}_{\mathrm{pair}}|}{n^2}(1 + o_{\mathrm{P}}(1)).$$

This lemma can be easily proved by assuming the indices $i$, $j$, $\pi^\natural(i)$, and $\pi^\natural(j)$ are uniformly sampled from the set $\{1, 2, \cdots, n\}$

# D  USEFUL FACTS

This section collects some useful facts for the sake of self-containing.

**Theorem 4** (Wick's theorem (Theorem 1.28 in Janson (1997))). *Considering the centered jointly normal variables $g_1, g_2, \cdots, g_n$, we conclude*

$$\mathbb{E}\left(g_1 g_2 \cdots g_n\right) = \sum_{\substack{\text{all possible disjoint} \\ \text{pairs } (i_k, j_k) \in \{1, 2, \cdots, n\}}} \prod_k \mathbb{E}\left(g_{i_k} g_{j_k}\right).$$

With Wick's theorem, we can reduce the computation of high-order Gaussian moments to calculating the expectations of a series of low-order Gaussian moments.

**Lemma 19** (Equation (3.2) in Neudecker & Wansbeek (1987)). *For a normally distributed random matrix $\mathbf{G} \in \mathbb{R}^{n \times p}$ which satisfies $\mathbb{E}\mathbf{G} = \mathbf{0}$ and $\mathbb{E}\text{vec}(\mathbf{G})\text{vec}(\mathbf{G})^\top = \mathbf{U} \otimes \mathbf{V}$, we have*

$$\mathbb{E}\left(\mathbf{G}^\top \mathbf{A} \mathbf{G} \mathbf{C} \mathbf{G}^\top \mathbf{B} \mathbf{G}\right) = \text{Tr}\left(\mathbf{A}\mathbf{U}\right)\text{Tr}\left(\mathbf{B}\mathbf{U}\right)\mathbf{V}\mathbf{C}\mathbf{V} + \text{Tr}\left(\mathbf{A}\mathbf{U}\mathbf{B}^\top \mathbf{U}\right)\mathbf{V}\mathbf{C}^\top \mathbf{V}$$
$$+ \text{Tr}\left(\mathbf{A}\mathbf{U}\mathbf{B}\mathbf{U}\right)\text{Tr}\left(\mathbf{C}\mathbf{V}\right)\mathbf{V},$$

*where $\text{vec}(\cdot)$ is the vector operation; $\otimes$ is the Kronecker product (Horn & Johnson, 1990); and $\mathbf{A}, \mathbf{B}$ and $\mathbf{C}$ are arbitrary fixed matrices.*

**Lemma 20** (Stein's Lemma (cf. Section 1.3 in Talagrand (2010))). *Let $g \sim \mathsf{N}(0, 1)$. Then for any differentiable function $f : \mathbb{R} \mapsto \mathbb{R}$ we have*

$$\mathbb{E}[gf(g)] = \mathbb{E}f^{'}(g),$$

*where $\lim_{\|g\| \to \infty} f(g)e^{-a\|g\|_2^2} = 0$ for any $a > 0$.*