# OpenReview forum: "The Phase Transition Phenomenon of Shuffled Regression"
_ICLR.cc/2025/Conference — Submitted to ICLR 2025_

### Official Review · Reviewer_YtGr · 2024-10-28

**Soundness:** 2
**Presentation:** 2
**Contribution:** 2
**Rating:** 3
**Confidence:** 4

**Summary:**

This paper studies high-dimensional shuffled linear regression, which is the following problem.
We are given a design matrix $X$ and a set of observations $Y$, drawn according to the following model:

$Y = Pi \cdot X \cdot B + \sigma W$

Here, $B$ is a "signal" matrix and $Pi$ is an unknown permutation matrix.
$\sigma > 0$ is a parameter and $W$ is a noise matrix with iid Gaussian entries.
In some variants of the problem, we may also be given $B$.
The goal is to recover $Pi$, and, if it is not given to us, $B$.

The paper offers a method to predict the signal-to-noise ratio above which a certain "message passing" algorithm, derived from statistical physics, can recover $Pi$ or $Pi,B$.
(The message passing algorithm is also derived in the paper.)
From what I can tell, the method is heuristic, making "physics-style" approximations along the way.
The authors corroborate their findings with some numerical experiments to show that the predictions they make are not too far numerically from the observed signal-to-noise threshold at which message passing stops working.

The problem of shuffled linear regression is a natural one, and well within the scope of ICLR.
However, I do not feel that the paper clears the bar for acceptance.

The main (major) issue is that the presentation of results is not at all clear.
The results are scattered throughout the 10 page, there is inadequate context for all of them.
For instance, I think proposition 1 is supposed to be one of the main results.
But I cannot tell how to interpret it as a mathematical proposition -- is it meant to be a rigorously-proven equation?

There are lots of heuristic physics-style "derivations" of phase transitions for message passing algorithms for high-dimensional learning problems now in the literature.
What is novel about this derivation?
If it is just a matter of turning the crank on existing technology, then even with clarified presentation I think it should go to a more specialized conference.
If there is something really novel about the way the derivation works here that I am missing, then perhaps it could be of interest to a broad ICLR audience.

**Strengths:**

see above

**Weaknesses:**

see above

**Questions:**

see above

---

> ### Author Response · Authors · 2024-11-15
> **Thank you & The Advantages of Our Approach**
>
> First, we would like to thank you for your feedback. Before we answer your concern, we would like to first discuss the advancement of our approach.
>
> 1. **Our work shows noticeable advantage over the work in the same field**. One example is [Lufkin et al. 2024], which was released this Feburary from several renowned statistists. It considers a similar but simpler setting of our paper $(m=1)$. A conjectured (c.f. the phase transition happens at $snr = n^{4/m}$ on the oracle case)  that they mention as an open problem can be easily derived by us, c.f. Line 372-374. In fact, we treat them as a warm-up example. Other advantages include 1) our work can predict the precise phase transition point; 2) our applicability to a wider class of cases; and 3) our consolidation into a cohesive framework to predict the phase transition point. More details can be found in Remark 4.
>
> 2. **Our work can predict the phase transition in a rather accurate way**. The numerical experiments show that differences between the predicted value and our predictions are under 0.001 in some cases. Meanwhile, previous works in the permuted linear regression only consider the logarithmic value (c.f.  Pananjady et al., 2018; Slawski & Ben-David, 2019; Pananjady et al., 2017; Slawski et al., 2020; Zhang & Li, 2020), that is, only checks the order of magnitude.

---

> ### Author Response · Authors · 2024-11-15
> **Addressing Concerns**
>
> Then, let us address your concerns.
>
> (1) First, we would like to clarify that **conducting our analysis within the message-passing (MP) framework does not diminish the technical novelty of our work**.
>
> MP serves as an analytical tool, much like concentration inequalities. Conducting analysis within the framework of MP is like giving analysis with concentration inequalities, which we hardly think downgrade the technical novelty. Determining the phase transition point of a specific group of message passing equations requires careful and intricate analysis.
>
> Furthermore, numerous papers adopting similar methodologies, or inherently related approaches (such as the replica method), have been published in leading machine learning conferences/journals. A few examples are listed in below.
>
> [Construction of optimal spectral methods in phase retrieval](https://proceedings.mlr.press/v145/maillard22a/maillard22a.pdf)
>
> [Stochasticity helps to navigate rough landscapes: comparing gradient-descent-based algorithms in the phase retrieval problem
> ](https://iopscience.iop.org/article/10.1088/2632-2153/ac0615/pdf)
>
> [Spectral redemption in clustering sparse networks](https://www.pnas.org/doi/epdf/10.1073/pnas.1312486110)
>
> [Probabilistic reconstruction in compressed sensing: algorithms, phase diagrams, and threshold achieving matrices](https://iopscience.iop.org/article/10.1088/1742-5468/2012/08/P08009/meta)
>
> [Optimal errors and phase transitions in high-dimensional generalized linear models
> ](https://www.pnas.org/doi/full/10.1073/pnas.1802705116)
>
> [Mutual information for symmetric rank-one matrix estimation: A proof of the replica formula
> ](https://proceedings.neurips.cc/paper/2016/file/621bf66ddb7c962aa0d22ac97d69b793-Paper.pdf)
>
> [Short-term memory in neuronal networks through dynamical compressed sensing](https://proceedings.neurips.cc/paper/2010/file/0f2c9a93eea6f38fabb3acb1c31488c6-Paper.pdf)
>
> (2) Second, we would like to confirm you that Proposition 1 is derived rigorously under the Assumption 1.
>
> (3) As for your concern regarding to the presentation clarity, we really appreciate it if you could provide more details.  For your convenience, we would like to give a brief roadmap of our results.
>
> Section 3 is about the oracle case, Section 3.1 derives the phase transition by directly analyzing the RV $\Xi$, and Section 3.2 proposes to approximate $\Xi$ with Gaussian approximation, which is easier to compute and more general.
>
> Section 4 concerns the non-oracle case, which calculates the phase transition point based on the Gaussian approximation.
> Proposition 3 concerns the position of the phase transition while Theorem 3 calculates the associated quantities.
>
> In the end, please feel free to let us know if you have any questions.

---

> ### Author Response · Authors · 2024-11-21
>
> Dear Reviewer,
>
> We are reaching out to check if you have any additional concerns, as the deadline for the discussion period is approaching. We would be happy to address any other questions you may have.
>
> Thank you.

---

> ### Author Response · Authors · 2024-11-27
>
> Dear Reviewer,
>
> As the deadline is approaching, we would like to ask whether you're satisfied with our responses. We will be happy to address the additional issues.
>
> Thank you.

---

### Official Review · Reviewer_5FFe · 2024-11-04

**Soundness:** 3
**Presentation:** 3
**Contribution:** 2
**Rating:** 6
**Confidence:** 2

**Summary:**

This paper studies the shuffled regression problem, in which unknown permutation matrix is multiplied to a linear model, with a focus on predicting phase transition. The authors leverage the message passing algorithm to find phase transition point in the limit of some parameters. In the analysis of phase transition, they categorize oracle and non-oracle cases, where the signal matrix is known and unknown respectively.

**Strengths:**

The derivation of the phase transition point using message passing and LAP is clearly presented and technically sound. This work distinguishes the difference from the recent papers having similar problem and settings, which in fact shows technical contribution over the  recent papers.

**Weaknesses:**

Some motivations and intuition in the derivation part are missing. For instance, eq(3), eq (10) and assumption 1 are given without motivation. Analysis of the approximation error (such as Gaussian approximation in eq (10) or the use of the lower bound for $\theta^*$) is not provided. As oracle case is rare in practice, I think comparison of non-oracle case with oracle case is needed at least in empirically, but it is missing in the paper.

**Questions:**

1) Is eq (3) necessary for the problem of shuffled regression? How much does prediction error arise if the prior distribution on the permutation matrix is not following eq(3)?
2) As I wrote in the weaknesses part, what are the motivation or reasoning behind using eq(3), eq(10), and assumption 1?
3) With the similar setting of Table 2, what are the predicted phase transition points by the algorithm for non-oracle case? Are they similar with the prediction of oracle case?
4) How to read the upper panel in figure 2? Is the fluctuated point of $\tau_h$ the phase transition point w.r.t. $\tau_h$? If yes, why it is?

---

> ### Author Response · Authors · 2024-11-15
> **Thank you**
>
> Dear Reviewer, first we would like to thank you for your feedback. Let us address your questions.
>
> > 1. Questions regarding Eq. (3), the underlying motivation, whether it is necessary and the prediction error.
>
> Eq. (3) is to frame the permuted linear regression under the message passing framework, which requires a probability distribution to build the probabilistic graphical model. Eq. (3) is to associate the optimization Eq (2) with a probability distribution, that allows us to study its property.  It is an equivalent transformation and thus won't bring any prediction errors.
>
>
> > 2. Reasoning behind Eq (10).
>
> It comes from the definition of random variable $\Xi$, whose definition can be found in Eq. (8).
>
> > 3. Motivation behind Assumption 1.
>
> This assumption is a common practice in analyzing the message passing algorithm such that the correlations are weak enough to be ignored. It's required by default except for some simple cases. And all previous study suggests that such assumption can be safely placed without any noticeable impact on the final prediction.
>
> A reference example can be found in Section 16.2.5 "Non-zero temperature and stability analysis" (start reading from the third paragraph) in Information, Physics, and Computation by Andrea Montanari and Marc Mézard [also cited in our paper].
>
> > 4. What are the predicted points for the non-oracle case in Table 2? Are they similar with Table 2?
>
> First, the non-oracle will be drastically different from the oracle case, whose phase transition point is with a smaller number. In addition, a direct comparison between the non-oracle case and oracle case is difficult. In the non-oracle case, we can predict and observe the appearance of another phase transition associated with the number of permuted rows (to be more specific, $\tau_h$). Predicting the phase transition induced by the SNR is more complex, as $\tau_h$ tends to trigger a phase transition earlier, complicating direct comparisons.
>
>
> > 5. Interpreting the upper panel in Fig 2.
>
> Yes. The fluctuated point is the predicted phase transition point. As you note, the SNR should be positive value. The sudden change to a negative value suggests a phase transition point happens here.
>
> In the end, we would like to thank you again for your feedback and please feel free to let us know if you have any other questions. Thank you.

---

> > ### Comment · Reviewer_5FFe · 2024-11-26
> >
> > I would like to thank the authors for the answers to my questions. I will keep my score.

---

### Official Review · Reviewer_vEMT · 2024-11-04

**Soundness:** 3
**Presentation:** 2
**Contribution:** 3
**Rating:** 8
**Confidence:** 1

**Summary:**

The paper studies the phase transition phenomenon in the shuffled regression problem. The authors transform the permutation recovery problem into a probabilistic graphical model and use message passing (MP) algorithms to derive equations tracking convergence. To address the challenge of deriving closed-form formulas for critical points, they propose a Gaussian approximation method, which provides accurate predictions of phase transition thresholds in multiple scenarios. Experiments are provided and suggest that the proposed algorithm is able to recover the structure.

**Strengths:**

- The problem being studied seems novel. I'm more familiar with phase transitions in the context of network models, and in the context of permuted LR the problem seems new.
- The authors propose a new method which leverages the MP algorithms by relating the combinatorial structure in the shuffled regression problem to a graphical model. This seems novel.
- Detailed proofs for the phase transition points are included. Specifically the Gaussian approximation analysis can be interesting when tackling more general scenarios.
- Synthetic experiments are provided, and the phase transition points from the results match the prediction from the theorem.

**Weaknesses:**

I am not familiar with this line of literature and I wouldn't be the best to provide an assessment. That being said, I think the organization of the paper can be improved. While I understand the paper is highly theoretical and statistical, it would be easier for readers to follow if the authors can provide more high level motivations or concrete examples.

**Questions:**

- Can the authors provide some motivation behind eq. (3)?

---

> ### Author Response · Authors · 2024-11-15
> **Thank you**
>
> Dear Reviewer, we would like to first thank you for your encouragement and high rating. Let us address your question, i.e., the motivation under Eq. (3).
>
> This is to frame the permuted linear regression under the message-passing framework. It requires constructing a probabilistic graphical model based on the joint probability distribution. Eq (3) transforms the optimization in Eq (2) to a probability distribution, thus enabling us to study the property of Eq (2) with the maximum prior estimate of $\mu(\Pi)$.
>
>
> In the end, we would like to thank you again for you comments. Please feel free to let us know if you have any other questions. We will be happy to answer them.

---

### Official Review · Reviewer_sF1x · 2024-11-10

**Soundness:** 1
**Presentation:** 2
**Contribution:** 1
**Rating:** 3
**Confidence:** 2

**Summary:**

The paper studies the shuffled multi-observation linear regression in two regimes: when the signal matrix is known and when it is unknown. The authors investigate the SNR at the phase transition.

**Strengths:**

The paper studies the general version of the problem that was studied in several prior works. The authors claim that they solve an open problem from a prior work [Lufkin et al., 2024]. While I'm not sure about this statement (see below), I think their result might be anyway helpful to investigate the regime that [Lufkin et al., 2024] were interested in.

**Weaknesses:**

Your assumption 1 seems to be unrealistic. You justify it by some numerical experiments, but it is not the usual way how assumptions in learning theory work. You have to assume some nice properties of B (or other paramerers of the problem), and then rigorously derive some result. Currently it is unclear to me whether there is any non-trivial matrix B that satisfies this assumption. Is it the case that you used this assumption in your formal proofs? Also, is it correct that your result formally answers the question from [Lufkin et al., 2024] only under this assumption?

In addition, your proofs seem to contain only equalities (or approximate equalities up to $o(1)$ terms). While this complaint might sound weird, it indicates that the approach may not be very sophisticated. You basically just do equivalent transformations of formulas. While potentially sometimes it might be non-trivial, practically almost always it is not the case. So far it seems to me that your assumption 1 is needed exactly for these equivalent transformations to work, and with any mathematically correct assumption the analysis has to become significantly more challenging.

Given these two observations, I recommend rejecting the paper.

**Questions:**

(see Weaknesses above)

---

> ### Author Response · Authors · 2024-11-14
>
> We thank you for your feedback. Let us answer some of your questions.
>
> First, we would like to confirm that **we only need Assumptin 1 to derive the conjecture of [Lufkin et al. 2024]**. We appreciate your suggestion regarding assumptions on some nice properties of B (or other parameters of the problem). However, our approach does not require us to impose such assumptions, allowing us to work under more general conditions.
>
> Second, Assumption 1 is to meet the i.i.d conditions in Theorem 1. Our work is not  `performing equivalent transformations`, as you commented. This can be easily seen from the depth and detail of our analysis, which spans 46 pages.

---

> ### Author Response · Authors · 2024-11-14
> **Points of Divergence: Areas of Disagreement**
>
> In addition, we respectfully disagree with some of your suggestions. A few points raised appear somewhat confusing and might even suggest some degree of bias. We hope to clarify these aspects further.
>
> First, you suggest that in learning theory community, `you have to assume some nice properties of B (or other parameters of the problem), and then rigorously derive some result.` In particular, we are very confused about your suggestion such that we should not justify our assumptions by numerical experiments.
>
> We regard **the theory needs to be aligned with the numerical experiments but not vice versa**. A theory that appears elegant but fails to align with experiments lacks any practical value.
>
> As you can see, **our numerical experiments (Table 1 and 2) align with the numerical experiments to a very good extent, with differences under 0.001 in some cases**. Meanwhile, notice that all the previous works in the permuted linear regression only considers the logarithmic value (c.f.  Pananjady et al., 2018; Slawski & Ben-David, 2019; Pananjady et al., 2017; Slawski et al., 2020; Zhang & Li, 2020), that is, **the order of magnitude**.
>
> Second, we are confused by your comment `our result only contain equalities (approximate equalities up to o(1) terms) suggest the approach are not very sophisticated`. As you noted, the goal of asymptotic analysis is to identify expressions that approximate complex behavior in the limit. Almost all the asymptotic analysis are of a similar look.
>
> While we do not view sophistication as a primary objective, we hope our analysis, which spans over 40 pages, demonstrates a thorough approach that goes beyond simple equivalence transformations.
>
> In the end, we are happy to assist you in understanding our paper and address your questions regarding the technical or whatever questions you have.

---

> ### Comment · Reviewer_sF1x · 2024-11-15
>
> Let's only focus on the issues with assumption 1. I never said that you should not justify assumptions by experiments. I said that assumptions should be mathematically correct and they should be true for some non-trivial classes of inputs, and some examples of such classes should be provided. Can you name some classes of $B$ that satisfy this assumption? (e.g. low rank, or sparse, or some other properties that can be fully described in terms of $B$, preferably with an easy description). When you do it, we can say that you proof works for these classes, but we cannot formally say anything about other $B$. You can say then that experiments suggest that for many other $B$ the result should also be true, that is fine and good for future works. The strength of you result then would mainly depend on the specific classes of $B$ for which assumption 1 is satisfied, and for some other classes of $B$ you can formulate a relevant conjecture based on the experiments.

---

> ### Author Response · Authors · 2024-11-15
> **Thank you for your response**
>
> Dear reviewer,
>
> We thank you for your response.
>
> Directly checking Assumption 1 is difficult however wet do not think that Assumption 1 is affected by the B matrix structure, as the **correlation is brought by the sensing matrix and the noise**.
>
> Numerical experiments also confirm this.  We conduct experiments on almost all the typical matrices, including both deterministic (c.f. Table 1 and 2) and Gaussian random matrices and other matrices. We do not find any matrices will lead to significant divergence between our predicted results and the numerical results. A detailed description of the B matrix can be found in our paper.
>
> We will be happy to add new experiments on the matrices you would like to try.

---

> ### Author Response · Authors · 2024-11-15
>
> Another fact we feel helpful for the discussion is that all the previous findings  (c.f.  Pananjady et al., 2018; Slawski & Ben-David, 2019; Pananjady et al., 2017; Slawski et al., 2020; Zhang & Li, 2020) suggests that **the structure of the B matrix barely has any impact on the phase transition point**, including the  [Lufkin et al., 2024] that also targets on the precise location of the phase transition point.
>
> In fact, the SNR and the stable rank of $B$ play the dominant role. This is also suggested in our result.
>
> In summary, we have already verified Assumption 1 with the following three ways:
>
> 1. Our numerical experiments on multiple matrices show good alignment between the numerical values and the predicted values.
> 2. Our prediction results agree with the previous rigorously derived result, that is, the SNR and the stable rank determines the phase transition point.
> 3. Our prediction results agrees with the conjecture in [Lufkin et al., 2024], which consider the same problem but under a much simpler setting,
>
> We hope this can address your concern over our assumption 1 and please feel free to let us know if you have questions on other issues.

---

> > ### Comment · Reviewer_sF1x · 2024-11-15
> >
> > I can only see that assumption 1 is satisfied for $B=0$. Are there other examples of $B$ that satisfy this assumption? If I understood you correctly, the correlations should be weak (it is a conclusion from the experiments), but they are almost always nonzero. These weak correlations should lead to some additional terms that should be vanishing, and showing that they are vanishing is complicated, that's why you just assumed that they are zero, is it correct?

---

> > > ### Author Response · Authors · 2024-11-21
> > >
> > > Dear Reviewer,
> > >
> > > We are writing to ask whether you have any other concerns, as the deadline for the discussion period is approaching.
> > > We will be happy to provide a more detailed explanation of Assumption 1. Also, we will be happy to answer other questions.
> > >
> > >
> > > Thank you.

---

> ### Author Response · Authors · 2024-11-15
>
> Thank you for the timely feedback.
>
> First, we agree with you that assuming the correlation is zero is inappropriate. But we should respectfully mention that **we do not make such assumption but only assume the correlation can be ignored**.  In fact, **our words is `ignore the weak correlation`** but we will make this more clear.
>
>
> Second, we would like to mention that **ignoring these correlations is quite common in analyzing the message-passing algorithm**. The most related example can be found in Section 16.2.5 "Non-zero temperature and stability analysis" (starting from the third paragraph) in Information, Physics, and Computation by Andrea Montanari and Marc Mézard [cited in our paper].
>
> They consider the linear assignment problem and give some heuristic explanations why the correlation may be ignored (because the correlation decays very fast). Although their explanation may not fully apply to our case (ours is more complex), we expect a justification of Assumption 1 can be provided following a similar argument.
>
>
> In the end, we would like to thank you again for your comments & please let us know if you have any other questions.

---

> ### Author Response · Authors · 2024-11-15
> **Follow-up Comments**
>
> Again we would like to thank you for your comments.
>
> We would like to be more specific here, that is, **all matrices $B$ should satisfy the Assumption 1**, at least our current findings do not find any $B$ that violates this assumption (c.f. the correlation is weak and can be ignored). This has already been verified by our numerical experiments.
>
> By the way, we do not assume the structure of B will have any impact on Assumption 1, as the correlation is brought by the sensing matrix $X$ and noise $W$. The sensing matrix $B$ has no role here.
>
> In the end, we really appreciate it if **you can kindly list any examples or potential examples you feel Assumption 1 is violated**. We are happy to conduct experiments and report the discrepancy between the numerical values and predicted values.
>
> Regarding your comment that only $B= 0$ will satisfy the assumption, we would like to kindly mention that **our paper contains several matrices (c.f. Table 1 and 2) that is not zero such that Assumption 1 can be safely taken, as the associated prediction accuracies are pretty high**.

---

> ### Author Response · Authors · 2024-11-27
>
> Dear Reviewer,
>
> We are writing to ask if you're satisfied with our responses, since the discussion period is about to end and you do not give any feedback since Nov 15, 2024.
>
> We will be happy to address any of your concerns. Thank you.

---

### Meta-Review · Area_Chair_XEpT · 2024-12-11

**Metareview:**

The views on this paper are fairly mixed, and some positive points are certainly there on giving new thresholds for message passing in shuffled regression.  However, there are still some strong doubts about (i) the underlying assumptions and level of rigor, and (ii) the writing clarity and presentation.

Regarding (i), since Assumption 1 was a focus of much of the discussion, I will give a summary of the latest thoughts:
- The statement in Assumption 1 is not mathematically formal, particularly the phrasing “We ignore … and view…”.  If you were to make it formal (e.g., by assuming exactly 0 correlation), then the assumption would probably be strictly false.  This raises the delicate question of whether any formal statements in the entire paper are fully justified.
- Assumption 1 is not cross-referenced after it is introduced, so it is hard to pinpoint where it is used.
- Formal theorem statements (e.g., Proposition 1) are stated as being true without mentioning Assumption 1, which is misleading.
- The derived thresholds are shown to have fairly good prediction accuracy in several scenarios (Tables 1 and 2), which is useful, but is by no means a complete justification.
- The reference to Montanari/Mezard is appreciated but it remains quite difficult to pinpoint exactly what they assumed, how it compares, and whether or not they similarly gave formal statements that rely on similar assumptions.
- I personally don't agree with the (implicit) defense that the reviewer(s) can’t specifically identify a case where the assumption fails.  That at most points to the assumption's validity being "inconclusive" rather than "justified".

Regarding (ii), perhaps this could be more dependent on the reader.  It is probably fair to say that the paper is not so accessible to outsiders, but on the other hand, that is often the case for these kinds of papers.

**Additional Comments On Reviewer Discussion:**

There was a fair bit interaction between the reviewers and authors, particularly on Assumption 1, which is why I focused on it in the meta-review.  In the private discussion, another reviewer re-iterated issues in the presentation and clarity.

---

### Decision · Program_Chairs · 2025-01-22

Reject